# Doubly Robust Uncertainty Quantification for Quantile Treatment Effects in Sequential Decision Making

**Yang Xu**                                                                      *yxu63@ncsu.edu*
*Department of Statistics*
*North Carolina State University*

**Chengchun Shi**                                                                *c.shi7@lse.ac.uk*
*Department of Statistics*
*London School of Economics and Political Science*

**Shikai Luo**                                                                   *sluo198912@163.com*
*ByteDance*

**Lan Wang**                                                                     *lanwang@mbs.miami.edu*
*Miami Herbert Business School*
*University of Miami*

**Rui Song**                                                                     *songray@gmail.com*
*Department of Statistics*
*North Carolina State University*

**Reviewed on OpenReview:** *https://openreview.net/forum?id=F0BwbieVws*

## Abstract

We consider multi-stage sequential decision making, where the treatment at any stage may depend on the subject's entire treatment and covariate history. We introduce a general framework for doubly robust uncertainty quantification for the quantiles of cumulative outcomes under a sequential treatment rule. While previous studies focused on mean effects, quantile effects offer unique insights into the distributional properties and are more robust for heavy-tailed outcomes. It is known that, doubly robust inference is significantly more challenging and largely unexplored for quantile treatment effects. More importantly, for mean effects, doubly robust estimation does not ensure doubly robust inference. Our approach first provides a doubly robust estimator for any quantile of interest based on pre-collected data, achieving semi-parametric efficiency. We then propose a novel doubly robust estimator for the asymptotic variance, enabling the construction of a doubly robust confidence interval. To overcome the challenges in parameter-dependent nuisance functions, we leverage deep conditional generative learning techniques. We demonstrate advantages of our approach via both simulation and real data from a short video platform. Additionally, we observe that our proposed approach leads to another mean effect estimator that outperforms existing estimators with heavy-tailed outcomes.

## 1 Introduction

Treatment effect evaluation plays an important role in a large variety of real-world applications. In precision medicine (see e.g., Murphy, 2003; Kosorok & Laber, 2019; Tsiatis et al., 2019), physicians wish to know the impact of implementing an individualized treatment strategy. In technological companies, decision makers conduct A/B testing to evaluate the performance of a newly developed product (see e.g., Bojinov & Shephard,

2019; Shi et al., 2023; Sun et al., 2024a; Wen et al., 2025). In economics and social science, program evaluation methods are widely applied to access the effects of certain intervention policies (see Abadie & Cattaneo, 2018, for an overview). In the aforementioned applications, a new treatment decision rule and/or policy need to be evaluated offline before online validation. Over the past decade, offline evaluation of a fixed, data-dependent or optimal policy has been widely studied in statistics (Zhang et al., 2012; Chakraborty et al., 2013; Zhang et al., 2013; Matsouaka et al., 2014; Luedtke & Van Der Laan, 2016; Luckett et al., 2020; Liao et al., 2021; 2022; Shi et al., 2022; Ramprasad et al., 2023; Wang et al., 2023; Zhou et al., 2023a; Shi et al., 2024b; Bian et al., 2025) and machine learning (Dudík et al., 2011; Jiang & Li, 2016; Thomas & Brunskill, 2016; Xie et al., 2019; Cai et al., 2020; Dai et al., 2020; Kallus & Zhou, 2020; Kallus & Uehara, 2020; Namkoong et al., 2020; Shi et al., 2020; Hao et al., 2021; Shi et al., 2021a; Chen & Qi, 2022; Xu et al., 2023; Bossens & Thomas, 2024; Yu et al., 2024; Zhou et al., 2025).

Nonetheless, all the previously mentioned works have primarily focused on the mean outcome, which may not fully capture the distributional effects of a policy, such as its impact on specific quantiles or the entire outcome distribution, are of primary interest. Take our real-data application as an example. The data is collected from a world-leading technological company with one of the largest mobile platforms for production, aggregation and distribution of short-form videos. We aim to assess the impact of specific advertising policies on user experience, measured by the length of stay on the platform. A particularly important subgroup is the passive users, who exhibit little responsiveness to any advertising policy and tend to spend minimal time on the platform. These users are concentrated in the lower tail of the engagement distribution, around the 25th percentile. Policy-makers are particularly interested in addressing the needs of this subgroup, since treatments with a greater impact on the lower quartile are more likely to attract potential customers who were previously less engaged with the platform.

In addition to short video platforms, evaluating the distributional effects of a policy is crucial in many other applications as well. For instance, in welfare programs, policy-makers may wish to identify interventions that specifically reduce the cost of living for the lowest 10% individuals, rather than focusing on the average population (Cui & Han, 2023). When studying gender discrimination, researchers often care about the entire distribution of wages, social visibility, or media coverage, rather than solely comparing mean differences between men and women (Firpo, 2007). In microbiome research, sequencing reads (Wang et al., 2025), and in e-commerce, advertisement clicks or profit (Wei et al., 2023) are typically non-negative and zero-inflated. In these cases, the mean can be a misleading summary statistic, making it more appropriate and informative to focus on quantiles.

Even when the mean outcome is of interest, traditional sample average estimators can perform poorly in the presence of skewed or heavy-tailed outcomes (Rowland et al., 2023; Zhu et al., 2024; Behnamnia et al., 2025). To illustrate the prevalence of heavy-tailed outcomes in practice, Figure 1 visualizes the distribution of the outcome variable in our real data example – user watch time – using a Q-Q plot and a log-log frequency plot. The Q-Q plot reveals a clearly right-skewed distribution, while the log-log plot displays an approximately linear decay in its probability density function, consistent with a power-law tail distribution – such as the $t$-distribution commonly used to model heavy-tailed outcomes – of the form $x^{-\alpha}$. As we will show later in Section 5.3, when outcome variables are heavy-tailed, computing quantiles first and then averaging results in a much improved estimator for the mean. This further motivates us to study quantile treatment effects.

This paper focuses on developing doubly robust and semi-parametrically efficient statistical inference framework for the quantiles of a target treatment's return using pre-collected datasets from multi-stage studies, which has been less studied in literature. In causal inference, the estimation of the causal effect often depends on two nuisance parameters, one for the propensity score and another for the outcome regression model. An estimator is doubly robust if it is consistent for the causal effect of interest when any one of the two nuisance parameters is consistently estimated. While doubly robust estimation has been extensively studied for mean-based estimators, our work addresses two key difficulties that set it apart from existing studies: (1) estimating quantiles rather than means, and (2) conducting rigorous statistical inference beyond point estimation. These additional complexities make our approach both novel and technically demanding.

The challenges of conducting treatment effect inference on quantiles are twofold. Firstly, constructing a doubly robust confidence interval requires doubly robust asymptotic linearity, which is more stringent than

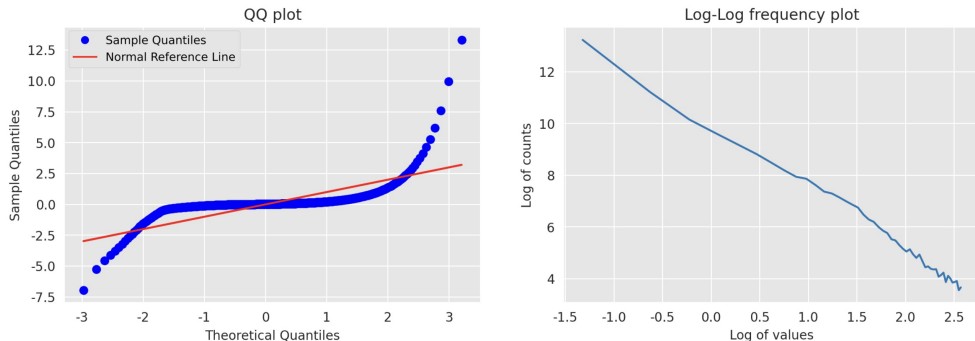

Figure 1: Left: Q-Q plot that compares the sample quantiles of user watch time in our real dataset to the theoretical quantiles of a standard normal distribution. Right: Log-log plot showing the count of data points at each user watch time value, grouped by percentiles. Both plots highlight the heavy-tailed nature of the outcome distribution in our dataset.

doubly robust consistency and has been less studied. Benkeser et al. (2017) provided a thorough discussion of the challenges associated with doubly robust inference for the average treatment effect at a single time point. In particular, they revealed from a careful theoretical analysis that when flexible, data-adaptive estimators of the nuisance parameters are used, doubly robust estimation does not readily extend to doubly robust inference. They showed that some natural alternatives are inappropriate for this purpose and further proposed a doubly robust inference procedure via targeted minimum loss-based estimation, which, however, requires additional nuisance parameters. Secondly, the efficient estimating equation for the quantile value is nonsmooth and involves parameter-dependent conditional mean function estimation which requires learning the conditional mean function for various values of the target parameter (Kallus et al., 2019). Directly applying supervised learning algorithms to construct the efficient estimating equation is very computationally intensive. This problem becomes more challenging when considering multiple-stage scenarios with history-dependent treatment.

Our contributions can be summarized as follows: (i) We propose a comprehensive doubly-robust uncertainty quantification procedure for quantile treatment effect in sequential decision making. In particular, we develop a doubly-robust quantile value estimator and a doubly-robust estimator for its asymptotic variance. Semi-parametric efficiency is established to guarantee the performance of our quantile estimator. To the best of our knowledge, this is the first work that rigorously investigates uncertainty quantification for a target policy's quantile value in multi-stage studies. (ii) We propose utilizing existing state-of-the-art deep conditional generative learning methods to handle parameter-dependent nuisance function estimation, which circumvents the error aggregation commonly encountered in standard iterative estimation methods prevalent in sequential decision making. (iii) We further develop a tail-robust doubly-robust estimator for the mean outcome by aggregating the proposed quantile estimator at multiple quantile values. Our simulation study demonstrates that the mean squared error (MSE) of the proposed estimator is around 15%-50% less than that of the standard doubly-robust mean effect estimator with heavy-tailed responses.

## 1.1 Related work

In the literature, several methods have been proposed for evaluating treatment effects beyond the mean. One line of work focuses on quantile or distribution regression (Machado & Mata, 2005; Chernozhukov et al., 2013). Another line employs inverse probability weighting, using propensity scores to reweight each outcome in order to estimate off-policy quantile treatment effects (QTE, Firpo, 2007) or cumulative distribution functions (Donald & Hsu, 2014). More recently, Kallus et al. (2019) proposed a localized debiased machine learning approach for QTE estimation, while Muandet et al. (2021) introduced a nonparametric method based on kernel mean embeddings. Huang et al. (2021) further extended prior analyses from mean outcomes to distributional metrics in contextual bandit settings. Wu et al. (2023a) proposed a non-crossing neural network architecture for learning distributional individualized treatment effects. A recent concurrent work

by Cheng et al. (2024) also investigated the quantile treatment effect under a marginal structural quantile model. Our work differs from theirs by further establishing a doubly robust estimator for the variance.

A related but different line of work focuses on policy learning rather than treatment effect estimation. For example, Wang et al. (2018a) and Qi et al. (2019) studied optimal policy learning to maximize distributional quantities other than the mean outcome. Later, Leqi & Kennedy (2021) considered optimizing the average of conditional quantiles to promote "across-group fairness". More recently, Cui & Han (2023) proposed minimax policy learning that are robust to model uncertainty. Despite these developments, both strands of research – whether focused on policy evaluation or policy learning – have largely been confined to single-stage decision making settings, limiting their applicability to real-world problems that require sequential decision making.

Some recent works have explored distributional evaluation and policy optimization in sequential decision making and reinforcement learning (see e.g., Dabney et al., 2018; Zhou et al., 2020; Huang et al., 2022; Li et al., 2022; Wu et al., 2023b; Zhang et al., 2023; Li et al., 2024; Sun et al., 2024b; Shen et al., 2025). However, most of the aforementioned work focuses on point estimation of target quantiles, with limited attention to inference procedures. Our contribution to this line of research is to provide a doubly robust inference procedure for quantile-based policy evaluation.

## 1.2 Paper Organization

The rest of the paper is organized as follows. In Section 2, we introduce the doubly-robust estimation and inference procedure for single-stage decision making. In Section 3, we formally establish the framework to the multi-stage setting with detailed algorithm and estimation procedures. In Section 4, we study the asymptotic properties and semi-parametric efficiency of the proposed procedure. Simulation studies and real data analysis are conducted in Sections 5 and 6, respectively. Finally, we conclude our paper in Section 7. All the proofs are presented in the appendix.

## 2 Single-Stage Decision Making

### 2.1 Preliminaries

We first describe the observed data. Let $X \in \mathcal{X}$ denote the $d$-dimensional vector that summarizes the baseline information (e.g., age, gender) of a given subject before treatment assignment, let $A \in \mathcal{A} = \{1, \cdots, m\}$ denote the treatment assigned to that subject where $m$ denotes the number of treatment options, and let $R \in \mathbb{R}$ denote the subject's outcome (the larger the better by convention). The observed data can thus be summarized as $N$ i.i.d. covariates-treatment-outcome triplets $\{X_i, A_i, R_i\}_{i=1}^N$ where $N$ denotes the number of subjects.

We next introduce the potential outcomes. For any $1 \le a \le m$, let $R^*(a)$ denote the potential outcome had the subject received the treatment $a$. Let $b : \mathcal{X} \times \mathcal{A} \mapsto [0, 1]$ denote the behavior policy that generates the data, i.e., $b(a|x) = \mathbb{P}(A = a|X = x)$. Likewise, let $\pi : \mathcal{X} \times \mathcal{A} \mapsto [0, 1]$ denote the target policy we wish to evaluate. Notice that $b$ is allowed to be unknown, as in observational studies.

Finally, we introduce the following standard assumptions:

(C1) Consistency: $R = R^*(A)$.

(C2) No unmeasured confounders (Rosenbaum & Rubin, 1983): $A \perp (R^*(1), \cdots, R^*(m))|X$.

(C3) Positivity: There exists some constant $\epsilon > 0$, such that $\mathbb{P}(b(A|X) \ge \epsilon) = 1$.

We remark that (C1)-(C3) are commonly imposed in causal inference and individualized treatment regimes literature (see e.g., Zhang et al., 2012; Wang et al., 2018a; Nie & Wager, 2021). They allow us to infer the potential outcome distribution from the observed data. (C2) and (C3) are automatically satisfied in randomized studies where the behavior policy is usually a strictly positive function independent of $x$.

## 2.2 Quantile Treatment Effect evaluation

Let $R^*(\pi)$ denote the potential outcome variable following the target policy $\pi$. For a given quantile level $0 < \tau < 1$, our problem of interest is to infer the $\tau$th quantile of $R^*(\pi)$ marginalized over the initial state distribution $X \in \mathbb{G}$, denoted by $\eta_\tau$. Specifically, we have $\tau = \mathbb{P}(R^*(\pi) \leq \eta_\tau) = \mathbb{E}_{X \sim \mathbb{G}}[\mathbb{P}(R^*(\pi) \leq \eta_\tau | X)]$. For a given target policy $\pi$, the $\tau$th quantile of $R^*(\pi)$ can be decomposed to a mixture of $\{R^*(a) : 1 \leq a \leq m\}$ with weights specified by $\pi$. As such,

$$\mathbb{P}(R^*(\pi) \leq r | X) = \sum_{a=1}^{m} \pi(a|X)\mathbb{P}(R^*(a) \leq r | X). \tag{1}$$

Under (C1)-(C3), the conditional potential outcome distribution $\mathbb{P}_{R^*(a)|X}$ is equal to the conditional distribution of observed outcome $\mathbb{P}_{R|X,A}$. This together with Equation 1 allows us to identify the quantile value from the observed data. In the following, we first introduce the direct method and the inverse probability weighting method, and then combine the two for efficient and robust estimation of the quantile treatment effect.

### 2.2.1 Direct Method (DM)

The first estimator is provided by a direct method, where we integrate quantile regression with Equation 1 to devise an optimization-based approach for quantile estimation. The methodology is conceptually equivalent to the g-computation formula used in causal inference (Daniel et al., 2011; Snowden et al., 2011; Chatton & Rohrer, 2024). Let $\rho_\tau(u) = u \cdot (\tau - \mathbb{I}\{u < 0\})$ denote the quantile loss function. By the law of iterated expectation, $\eta_\tau = \arg\min_\eta \mathbb{E}[\rho_\tau(R^*(\pi) - \eta)] = \arg\min_\eta \mathbb{E}[\mathbb{E}\{\rho_\tau(R^*(\pi) - \eta)|X\}]$. Under (C1)-(C3), it follows that

$$\widehat{\eta}_{\mathrm{DM}} = \arg\min_\eta \sum_{a=1}^{m} \widehat{\mathbb{E}}[\pi(a|X) \cdot \widehat{\mathbb{E}}\{\rho_\tau(R - \eta)|A = a, X\}], \tag{2}$$

where $\widehat{\mathbb{E}}$ denotes the empirical expectation calculated from the observed data. DM first estimates the conditional expectation within the square brackets for each $\eta$ and then plugs in this estimator in the above optimization problem to learn $\eta_\tau$.

As discussed in the introduction, it remains challenging to compute the conditional mean estimator for each $\eta$. We will later introduce our proposal to address this issue.

### 2.2.2 Inverse Probability Weighting (IPW) method

An inverse probability weighting estimator, akin to the off-policy mean outcome estimation in classical causal literature, can be developed by the change of measure theorem based on Equation 2:

$$\widehat{\eta}_{\mathrm{IPW}} = \arg\min_\eta \widehat{\mathbb{E}}\left[\frac{\pi(A|X)}{\widehat{b}(A|X)}\rho_\tau(R - \eta)\right]. \tag{3}$$

When the behavior policy $b$ is unknown, it can be estimated using any proper supervised learning algorithm with a mild convergence rate. The precise requirements are provided in Assumption (A1) and Remark 1.

### 2.2.3 Doubly Robust (DR) Estimator

In causal inference, an estimator is doubly robust if it is consistent for the causal effect of interest when any one of the two nuisance functions, i.e. propensity score or outcome regression function, is consistently estimated. As a result, the doubly-robust method combines the DM and IPW procedures and alleviates the risk of model misspecification by only requiring one of the nuisance functions to be correctly specified. Based on the classical DR estimator for the mean outcome, a DR quantile estimator can be formulated by

incorporating the fundamental concept of quantile regression as follows:

$$
\begin{aligned}
\widehat{\eta}_{\mathrm{DR}} &= \arg\min_{\eta} \frac{1}{N} \sum_{i=1}^{N} \Psi(\eta; X_i, A_i, R_i, \widehat{b}, \widehat{\mathbb{E}}) \\
&\equiv \arg\min_{\eta} \frac{1}{N} \sum_{i=1}^{N} \left[ \frac{\pi(A_i|X_i)}{\widehat{b}(A_i|X_i)} \rho_\tau(R_i - \eta) + \left(1 - \frac{\pi(A_i|X_i)}{\widehat{b}(A_i|X_i)}\right) \widehat{\mathbb{E}}\{\rho_\tau(R_i - \eta)|X_i, A_i \sim \pi\} \right],
\end{aligned}
\tag{4}
$$

In the above formula, $\widehat{b}$ represents the estimated behavior policy. $\widehat{\mathbb{E}}\{\cdot|X_i, A_i \sim \pi\}$ denotes the conditional mean functional that needs to be estimated later. Here, the reward $R_i$ in the conditional expectation is a random variable by executing the target policy $\pi$ under baseline information $X_i$.

The first term of the second line in Equation 4 is identical to the IPW estimator. The second part is an augmentation term which offers additional protection against the model misspecification of the behavior policy while improving the estimation efficiency. When the conditional mean function is correctly specified, the expectation of our DR estimator is equivalent to that of the DM estimator. When the behavior policy is correctly specified, the estimator is unbiased to the IPW estimator and thus is also consistent. This verifies that $\widehat{\eta}_{\mathrm{DR}}$, as a combination of DM and IPW estimators, indeed satisfies the doubly-robustness property.

Similar to DM, DR requires estimating the conditional mean function for each $\eta$. We propose employing a deep conditional generative learning method to compute $\widehat{\mathbb{E}}\{\cdot \mid \cdot\}$. The main idea is utilizing deep neural networks to learn a conditional sampler that takes the covariate-treatment pair and some random noises as input and outputs pseudo outcomes whose conditional distribution is similar to that of $\mathbb{P}_{R|A,X}$. By using the Monte Carlo method, these pseudo outcomes can be further used to approximate the conditional mean functions for any $\eta$. This addresses the challenge of parameter-dependent nuisance function estimation. As a powerful approach for generating random samples from complex and high-dimensional conditional distributions, deep conditional generative learning methods have been successfully implemented in a large variety of applications including computer vision, imaging processing, epidemiology, time series analysis and machine learning (Yan et al., 2016; Shu et al., 2017; Wang et al., 2018b; Bellot & van der Schaar, 2019; Davis et al., 2020; Jo et al., 2021; Shi et al., 2021b; Zhou et al., 2023b; Shi et al., 2024a; Zhang et al., 2024; Yang et al., 2025). We detail our estimation procedure in the next section.

## 2.3   Estimation and Inference Procedures

Our estimation procedure consists of four steps: 1) sample splitting, 2) nuisance functions estimation, 3) doubly-robust quantile value and variance estimation, 4) tail-robust mean value estimation. Pseudocode summarizing the proposed algorithm is given in Algorithm 1.

### 2.3.1   Step 1: Sample Splitting

Step 1 of the proposed algorithm is to randomly divide samples $\{1, \ldots, N\}$ into $S$ disjoint subgroups of equal size. We define $n = N/S$ and assume $n \in \mathbb{Z}$ without the loss of generality. This step incorporates Equation 4 with cross-fitting in Step 3 to reduce the bias of the nuisance function estimators obtained by the aforementioned supervised and generative learning methods. For any $1 \leq s \leq S$, let $\mathcal{I}_s$ denote the $s$th subgroup and $\mathcal{I}_s^c$ denote its complement. Sample splitting allows us to use the data in $\mathcal{I}_s^c$ to compute $\widehat{b}$ and $\widehat{\mathbb{E}}\{\cdot\}$, and those in $\mathcal{I}_s$ to construct the estimating equation. In addition, we can get full efficiency by aggregating the estimating equations over different $s$. Such a technique has been implemented for policy evaluation in recent literature (Chernozhukov et al., 2018; Shi et al., 2021a; Kallus & Uehara, 2022).

### 2.3.2   Step 2: Nuisance Functions Estimation

Step 2 of the proposed algorithm is to use each data subset in $\mathcal{I}_s^c$ to estimate the behavior policy $b$ and the conditional distribution of $R$ given $A$ and $X$.

Firstly, notice that estimating the behavior policy is essentially a regression problem. As such, we can apply any supervised learning algorithm designed for categorical response variables to compute $\widehat{b}$. In our

---

**Algorithm 1** Pseudo Code of Quantile Treatment Effect Evaluation in Single Stage

---

1: **Input:** $\{X_i, A_i, R_i\}_{i=1}^N$ triplets; quantile levels $\tau$; number of folds $S$.
2: **Output:** doubly robust quantile estimator $\widehat{\eta}_\tau^{\mathrm{DR}}$; tail-robust DR mean estimator $\hat{\mu}_{\mathrm{DR}}$.
3: **procedure** A. DOUBLY ROBUST QUANTILE ESTIMATOR
4:     Randomly split the data into $S$ folds. For any $s \in S$, denote $\mathcal{I}_s$ as the $s$th subgroup.
5:     **for all** $s \in S$ **do**
6:         Estimate $\widehat{b}$ by GBDT with the data in $\mathcal{I}_s^c$;
7:         Estimate $\widehat{R}|(X, A)$ by MDN with the data in $\mathcal{I}_s^c$;
8:         Generate $\{\widehat{R}_{x_i,a}^j|(X = x_i, A = a)\}_{j=1}^M$ for any $a \in \mathcal{A}$ and $i \in \mathcal{I}_s$;
9:     **end for**
10:     Solve the optimization problem in formula (4) to obtain $\widehat{\eta}_{\mathrm{DR}}$.
11: **end procedure**
12: **procedure** B. TAIL-ROBUST DR MEAN ESTIMATOR
13:     For all of the quantile levels of our interest, calculate $\widehat{\eta}_\tau$ by Procedure A. Averaging among all quantiles yields the tail-robust DR mean estimator $\widehat{\mu}_{\mathrm{DR}}$.
14: **end procedure**

---

implementation, we employ a Gradient Boosting Decision Tree (GBDT) algorithm and find that it works well in our numerical examples.

Secondly, we need to estimate the conditional expectation $\mathbb{E}[\rho_\tau(R - \eta)|X = x, A = a]$. As discussed earlier, one approach is to apply supervised learning to approximate the conditional expectation for any $\eta$. However, such a method is computationally intensive when the search space of $\eta$ is large, since it is unrealistic and super inefficient to posit a separated model for each possible value of $\eta$. To facilitate the computation, we leverage the state-of-the-art deep conditional generative learning methods to learn the conditional reward distribution. This type of approach includes but not limited to Generative Adversarial Networks (GAN) and Mixture Density Network (MDN), which provide promising performance in real applications.

In the subsequent sections, we employ Mixture Density Networks (MDN) as an example for illustrative purposes. It's worth noting that our framework is adaptable to the implementation of any conditional distribution approximation methodology, provided that the convergence rate satisfies a readily attainable assumption (Assumption (A1)), which will be elucidated in Section 4.

For a brief introduction to MDN, it integrates the Gaussian mixture model with deep neural networks by assuming that the conditional density of $R$ given $X$ and $A$ is a mixture of $J$ conditional Gaussian distributions. That is, the probability density function of $R|(X, A)$ satisfies

$$f(r|x, a) = \sum_{j=1}^J \alpha_j(x, a) \frac{1}{\sqrt{2\pi}\sigma_j(x, a)} \exp\left\{ -\frac{(r - \mu_j(x, a))^2}{2\sigma_j^2(x, a)} \right\},$$

with weights $\{\alpha_j(x, a)\}_{j=1}^J \in [0, 1]$, conditional means $\{\mu_j(x, a)\}_{j=1}^J$, and the corresponding variances $\{\sigma_j(x, a)\}_{j=1}^J$ parameterized via deep neural networks. The universal approximation property of deep neural nets allows to approximate any smooth or non-smooth weight, conditional mean and variance functions (Imaizumi & Fukumizu, 2019). This together with the universal approximation property of Gaussian mixture models allows $f$ to approximate any smooth conditional density functions (see e.g., Kostantinos, 2000).

Finally, we discuss how to compute $\widehat{\mathbb{E}}[\rho_\tau(R - \eta)|X = x, A = a]$. The idea is to sample i.i.d. pseudo outcomes $\{\widehat{R}_{x,a}^{(j)} : 1 \leq j \leq M\}$ based on the estimated conditional distribution generator for $R|(X, A)$ and approximate the expectation via Monte Carlo, i.e.,

$$\widehat{\mathbb{E}}[\rho_\tau(R - \eta)|X = x, A = a] \approx \frac{1}{M} \sum_{j=1}^M \rho_\tau(\widehat{R}_{x,a}^{(j)} - \eta).$$

The number of Monte Carlo samples $M$ represents a trade-off. Notice that the bias of the estimator is the same for any $M$. A large $M$ improves the accuracy of the estimator at the cost of increasing the computation

time. In theory, as shown in Section 4, we require $M \to \infty$ to ensure the resulting value estimator achieves the minimum asymptotic variance. In our numerical implementation, we set $M$ to 50 to achieve a good balance between estimation accuracy and computation cost.

### 2.3.3 Step 3: Doubly-Robust Quantile Value and Variance Estimation

Step 3 of the proposed algorithm is to employ cross-fitting to estimate the quantile value and construct the associated confidence interval. In particular, we develop two doubly-robust estimators, one for the quantile value itself, and another for the asymptotic variance of the quantile estimator.

First, let $\widehat{b}_s$ and $\widehat{\mathbb{E}}_s$ denote the estimated behavior policy and conditional mean function, respectively. We use cross-fitting to construct the following estimating equation to compute the doubly-robust quantile estimator, i.e.

$$\widehat{\eta}_\tau^{\mathrm{DR}} = \arg\min_\eta \sum_{s=1}^{S} \sum_{i \in \mathcal{I}_s} \Psi(\eta; W_i, \widehat{b}_s, \widehat{\mathbb{E}}_s),$$

where $\Psi$ is defined in Equation 4 and $W_i$ is a shorthand for the triplet $(X_i, A_i, R_i)$.

Second, as shown in Theorem 1 (see Section 4), the asymptotic variance of $\widehat{\eta}_\tau^{\mathrm{DR}}$ is given by the sandwich formula $J_0^{-2}\mathrm{Var}(\partial_\eta \Psi(\eta_\tau; W_i, b, \mathbb{E}))$ where $J_0 := f_{R^*(\pi)}(\eta_\tau)$ equals the probability density function (pdf) of the potential outcome $R^*(\pi)$ evaluated at $\eta_\tau$. While the variance term $\mathrm{Var}(\partial_\eta \Psi(\eta_\tau; W_i, b, \mathbb{E}))$ can be consistently estimated via the sampling variance estimator, the pdf is more difficult to learn from the data.

A key observation is that, $f_{R^*(\pi)}(\eta_\tau)$ is not only the marginal pdf of the reward following $\pi$ and the baseline distribution $\mathbb{G}$, but the expectation of a conditional pdf for $f_{R^*(\pi)|X}(\eta_\tau|X)$ as well. This motivates us to consider the following two estimators for $f_{R^*(\pi)}(\eta_\tau)$:

**(i) IPW estimator for $J_0$:** for those subjects following the target policy $\pi$, a natural estimator for $J_0$ can be obtained by applying kernel density estimation (KDE). This idea, combined with the change of measure theorem, yields the following IPW estimator for $J_0$:

$$\widehat{J}_0^{\mathrm{IPW}} = \frac{1}{N} \sum_{s=1}^{S} \sum_{i \in \mathcal{I}_s} \frac{\pi(A_i|X_i)}{\widehat{b}_s(A_i|X_i)} \frac{1}{h} K\left(\frac{R_i - \widehat{\eta}_{\mathrm{DR}}}{h}\right),$$

where $K$ is some kernel function and $h$ is a pre-specified bandwidth.

**(ii) IPW estimator for $J_0$:** In Step 2, we have obtained the conditional pdf of $R|X, A$ via some deep conditional generative model. Let $\widehat{f}_s$ denote the conditional probability density function obtained from the data in $\mathcal{I}_s^c$. This yields the following direct method estimator for $f_{R^*(\pi)}(\eta_\tau)$,

$$\widehat{J}_0^{\mathrm{DM}} = \widehat{f}_{R^*(\pi)}(\eta_\tau) = \frac{1}{N} \sum_{s=1}^{S} \sum_{i \in \mathcal{I}_s} \sum_a \pi(a|X_i)\widehat{f}_s(\widehat{\eta}_{\mathrm{DR}}|a, X_i) := \frac{1}{N} \sum_{s=1}^{S} \sum_{i \in \mathcal{I}_s} \widehat{f}_{R^*(\pi),s}(\widehat{\eta}_{\mathrm{DR}}|X_i).$$

Combining both (i) and (ii) yields a doubly-robust variance estimator, given by

**(iii) DR estimator for $J_0$:**

$$\widehat{J}_0^{\mathrm{DR}} = \frac{1}{N} \sum_{s=1}^{S} \sum_{i \in \mathcal{I}_s} \left[ \frac{\pi(A_i|X_i)}{\widehat{b}_s(A_i|X_i)} \frac{1}{h} K\left(\frac{\widehat{\eta}_{\mathrm{DR}} - R_i}{h}\right) + \left(1 - \frac{\pi(A_i|X_i)}{\widehat{b}_s(A_i|X_i)}\right) \widehat{f}_{R^*(\pi)}(\widehat{\eta}_{\mathrm{DR}}|X_i) \right]. \tag{5}$$

As such, a Wald-type confidence interval can thus be derived for $\eta_\tau$, which is detailed in Equation 18 of the theory section as a multi-stage general form.

### 2.3.4 Step 4: Tail-Robust Mean Value Estimation

For some studies where the mean outcome is still of interest, Step 4 of the proposed algorithm provides a tail-robust mean value estimator based on the quantile values we estimated from Step 3. Since the target policy's mean value can be represented as an average of the quantile values, i.e., $\int_0^1 \eta_\tau d\tau$, this motivates us to consider the following plug-in estimator $\int_0^1 \widehat{\eta}_\tau^{\mathrm{DR}} d\tau$ for the mean value. The integral can be approximated numerically based on midpoint rule, trapezoidal rule or Simpson's rule.

Compared to the standard doubly-robust estimator, the proposed estimator is tail robust in that its consistency requires very mild moment conditions. Specifically, it only requires a well-defined mean value, i,e, $\mathbb{E}|R| < \infty$. On the contrary, the standard doubly-robust estimator requires the reward to have at least a finite second-order moment. In our numerical studies, we find that the proposed tail-robust estimator achieves a much smaller MSE when reward is heavy-tailed or has a quasi-infinite variance.

## 3 Sequential Decision Making

### 3.1 Preliminaries

In this section, we extend our framework to more general sequential decision making problems with $K$ stages. The data trajectory for a single subject is given by

$$(X_1, A_1, R_1, X_2, A_2, R_2, \ldots, X_K, A_K, R_K), \tag{6}$$

where $(X_k, A_k, R_k)$ denotes the covariates-treatment-outcome triplet observed at the $k$th decision stage and $K$ denotes the horizon (e.g., number of decision stages).

A (history-dependent) policy $\pi = (\pi_1, \cdots, \pi_K)$ maps the observed data history to a probability distribution over the action space at each time. Specifically, let $H_k$ denote the historical data collected prior to the assignment of $A_k$, i.e., $H_k = \{X_1, A_1, R_1, \cdots, X_k\}$.

Following $\pi$, the agent will set $A_k$ according to the probability mass function $\pi_k(\cdot|H_k)$ at each time $k$. In addition, we use $b = (b_1, \cdots, b_K)$ to denote the behavior policy that generates the data, which consists of $N$ i.i.d. realizations of equation 6, denoted by

$$\left\{ (X_{1,i}, A_{1,i}, R_{1,i}, X_{2,i}, A_{2,i}, R_{2,i}, \ldots, X_{K,i}, A_{K,i}, R_{K,i}) : 1 \le i \le N \right\}. \tag{7}$$

Note that the i.i.d. assumption is imposed across trajectories (i.e., over index $i$), rather than across time steps within a trajectory. That is, we do not assume $(X_k, A_k, R_k)$ are i.i.d. over $k$, which would be unrealistic in practice.

Usually, the final outcome can be defined as a function of rewards at all stages, i.e. $f(R_1, \ldots, R_K)$. However, in most cases when we periodically evaluate a specific kind of outcome of interest, it is natural to infer the cumulative reward $\sum_{k=1}^K R_k$ following the target policy $\pi$. Note that in the following sections, although we focus on the estimation and inference of $\sum_{k=1}^K R_k$ at the $\tau$th quantile, the whole framework can be easily extended to any reward function $f(R_1, \ldots, R_K)$.

Similar to Section 2, we assume versions of (C1$'$) consistency, (C2$'$) no unmeasured confounders, which is better known as the sequential randomization assumption, and (C3$'$) positive assumptions hold in multiple stage settings. An explicit version of these assumptions are provided in Appendix A.1. (see e.g., Zhang et al., 2013, for the detailed definitions of these assumptions).

### 3.2 Quantile Treatment Effect Evaluation

In multi-stage settings, our objective is to infer the $\tau$th quantile of the sum of potential outcomes at all stages under a fixed target policy $\pi$. Let $R_k^*(\bar{a}_k)$ denote the potential outcome one would observe under policy sequence $\bar{a}_k = (a_1, \ldots, a_k)$. Similarly, define $R_k^*(\pi)$ as the potential outcome under policy $\pi$, where

$R^*(\pi) = \sum_k R_k^*(\pi)$ is the cumulative outcome of interest. It then follows that

$$\mathbb{P}\Big(\sum_{k=1}^{K} R_k^*(\pi) \le r \Big| X_1\Big) = \sum_{\{a_k\}_{k=1}^{K}} \mathbb{E}\Big\{\prod_{k=1}^{K} \pi_k(a_k|H_k) \cdot \mathbb{P}\Big(\sum_{k=1}^{K} R_k^*(\bar{a}_k) \le r \Big| X_1\Big)\Big\}, \tag{8}$$

where the summation over $\{a_k\}_{k=1}^{K}$ can take all possible treatment combinations in $|\mathcal{A}|^K$, and the expectation $\mathbb{E}$ is taken w.r.t. possible sequences of $\{H_k, R_k\}_{1 \le k \le K}$. Under (C1')-(C3'), formula (8) can be represented using the observed data distribution. For any given quantile $\tau$, we solve the optimization problem

$$\eta_\tau = \arg\min_{\eta} \mathbb{E}_{X_1 \sim \mathbb{G}}\left[\rho_\tau\Big(\sum_{k=1}^{K} R_k - \eta\Big)\Big| A \sim \pi\right], \tag{9}$$

where $\mathbb{E}_{X_1 \sim \mathbb{G}}[\cdot | A \sim \pi]$ denotes the expectation with respect to the history data $(H_K, A_K, R_K)$, assuming the initial covariates $X_1$ were generated according to $\mathbb{G}$ and all subsequent observations were to follow the target policy $\pi$. In the following sections, we will first introduce the direct method and inverse probability weighting estimator, and then combine the above two to derive the doubly robust estimator.

### 3.2.1 The DM estimator

Under (C1')-(C3'), it follows from (8) that

$$\widehat{\eta}_{\text{DM}} = \arg\min_{\eta} \sum_{\{a\}_{k=1}^{K}} \left\{\widehat{\mathbb{E}}\Big[\prod_{k=1}^{K} \pi_k(a_k|H_k) \cdot \widehat{\mathbb{E}}\Big\{\rho_\tau\Big(\sum_{k=1}^{K} R_k - \eta\Big)\Big| \{A_k\}_{k=1}^{K} = \{a_k\}_{k=1}^{K}, X_1\Big\}\Big]\right\}. \tag{10}$$

Similar to the single-stage setting, we first estimate the conditional probability function for $\sum_{k=1}^{K} R_k$ given $(H_K, A_K)$ and then leverage MC approximation to calculate the conditional expectation in Equation 10. Details will be provided in Section 3.3.

### 3.2.2 The IPW estimator

According to the the change of measure theorem,

$$\widehat{\eta}_{\text{IPW}} = \arg\min_{\eta} \widehat{\mathbb{E}}\left[\prod_{k=1}^{K} \frac{\pi(A_k|H_k)}{\widehat{b}_k(A_k|H_k)} \cdot \rho_\tau\Big(\sum_{k=1}^{K} R_k - \eta\Big)\right].$$

Notice that when the number of stages increases, the proportion of data we observe that aligns with target policy will decrease drastically. This may lead to a poor IPW estimator. Akin to the single-stage setting, one can implement any supervised learning algorithm to learn $\widehat{b}_k$ that satisfies some mild convergence rate assumptions specified in Assumption (A1) of Section 4.

### 3.2.3 The DR estimator

By incorporating the DM and IPW estimators, one can derive a doubly robust estimator for $K$-stage quantile treatment effect:

$$\begin{aligned}
\widehat{\eta}_{\text{DR}} &= \arg\min_{\eta} \frac{1}{N}\sum_{i=1}^{N} \Psi(\eta; W_i, \widehat{b}, \widehat{\mathbb{E}}) \\
&\equiv \arg\min_{\eta} \frac{1}{N}\sum_{s=1}^{S}\sum_{i\in\mathcal{I}_s}\left[\frac{\prod_{k'=1}^{K}\pi_{k'}(A_{k',i}|H_{k',i})}{\prod_{k'=1}^{K}\widehat{b}_{k'}(A_{k',i}|H_{k',i})}\rho_\tau\Big(\sum_{k=1}^{K} R_{i,k} - \eta\Big) \right. \\
&\qquad \left. + \sum_{k=1}^{K}\left(\prod_{k'=1}^{k-1}\frac{\pi_{k'}(A_{k',i}|H_{k',i})}{\widehat{b}_{k'}(A_{k',i}|H_{k',i})}\right)\left(1 - \frac{\pi_k(A_{k,i}|H_{k,i})}{\widehat{b}_k(A_{k,i}|H_{k,i})}\right)\widehat{L}_k(H_{k,i})\right],
\end{aligned} \tag{11}$$

where $W_i$ is a shorthand for the observed data trajectory

$$(X_{1,i}, A_{1,i}, R_{1,i}, X_{2,i}, A_{2,i}, R_{2,i}, \ldots, X_{K,i}, A_{K,i}, R_{K,i}),$$

and $\widehat{L}_k(H_k) = \widehat{\mathbb{E}}\left[\rho_\tau(R_1 + \cdots + R_{k-1} + \widehat{R}_k + \cdots + \widehat{R}_K - \eta)|\{A_{k'}\}_{k'=1}^{k-1}, \{A_{k'}\}_{k'=k}^K \sim \pi, X_1\right]$. It is worth noting that in $L_k(H_k)$, the expectation is conditioning on $H_k$, in which $R_1, \ldots, R_{k-1}$ are directly obtained from the observed data, and $\widehat{R}_k, \ldots, \widehat{R}_K$ are the rewards estimated from some data generating model under treatment $(\pi_k, \ldots, \pi_K)$.

Equation 11 is the average of the sum of $K + 1$ terms. The first term serves as an IPW estimator, where the coefficient $\frac{\prod_{k'=1}^K \pi_{k'}}{\prod_{k'=1}^K \widehat{b}_{k'}}$ is the probability of a subject's actions at all $K$ stages being consistent with policy $\pi$. The last $K$ terms, shown in the second line, denote the augmentation terms estimating the quantile loss originated from the $k$th level of missing data. The $k$th augmentation term is the multiplication of two main components: the product of the two policy ratios in parenthesis, and a direct estimator $\widehat{L}_k(H_k)$. It aims to leverage subjects whose observed actions are consistent with $\pi$ during the first $k-1$ stages, but inconsistent starting from the $k$th stage. The product of the two policy ratios in this case roughly describes the probability of this inconsistency occurring. $\widehat{L}_k(H_k)$ is the expected quantile loss estimated by combining the observed reward before stage $k$ with the estimated rewards from stage $k$ onwards. Using this DR estimator, one can maximize the utilization of the observed information related to policy $\pi$. The estimation details are described in Section 3.3.

The proposed estimator $\widehat{\eta}_{\text{DR}}$ is doubly robust in the sense that if the propensity score functions $\{\widehat{b}_k(A_k|H_k)\}_{k=1}^K$ or the outcome regression models $\{\widehat{L}_k(H_k)\}_{k=1}^K$ are consistently estimated at each stage, then $\widehat{\eta}_{\text{DR}}$ is a consistent estimator of $\eta_\tau$. Compared with the DM and IPW estimators, the DR estimator $\widehat{\eta}_{\text{DR}}$ tends to be more robust w.r.t. model misspecification. This also enables us to implement more flexible machine learning methods and leverage their advantages in estimating propensity score and outcome models. Later in Section 4, we will also prove the semi-parametric efficiency of $\widehat{\eta}_{\text{DR}}$, which provides further theoretical guarantee for the performance of our estimator.

### 3.3 Estimation details

In this subsection, we focus on the estimation procedure under general $K$-stage settings. Similar to single stage, our approach contains four steps in sequential decision making process: (1) sample splitting, (2) nuisance functions estimation, (3) doubly-robust quantile value and (4) variance estimation, tail-robust mean value estimation. The pseudo code for multi-stage quantile estimation is summarized in Algorithm 2.

Steps 1 and 4 are essentially the same as the single-stage case described in Sections 2.3.1 and 2.3.4, and are therefore omitted here. Below, we focus on the differences in Steps 2 and 3.

In Step 2, the estimation involves two sets of functions: propensity score functions $\{b_k(H_k)\}_{k=1}^K$, and the conditional expectation functions $\{\widehat{L}_k(H_k)\}_{k=1}^K$. The estimation of $\{\widehat{L}_k(H_k)\}_{k=1}^K$ is more involved. First, the conditional distribution functions for $\{\widehat{R}_k|(H_k, A_k)\}_{k=1}^K$ can be obtained by some conditional generative modeling process. After getting these reward-generators at each stage, $\widehat{L}_k(H_k)$ can be estimated through a Monte Carlo approximation. As long as the number of replicates $M$ is large enough, $\widehat{L}_k(H_k)$ will converge to the true value if the model is correctly specified. Specifically,

$$\widehat{L}_k(H_k) = \widehat{\mathbb{E}}\left[\rho_\tau(R_1 + \cdots + R_{k-1} + \widehat{R}_k + \cdots + \widehat{R}_K - \eta_\tau^\pi)|\{A_{k'}\}_{k'=1}^{k-1}, \{A_{k'}\}_{k'=k}^K \sim \pi, X_1\right]$$

$$= \frac{1}{M}\sum_{j=1}^M \rho_\tau(R_1 + \cdots + R_{k-1} + \widehat{R}_k^{(j)} + \cdots + \widehat{R}_K^{(j)} - \eta_\tau^\pi),$$

where $\{\widehat{R}_k^{(j)}, \ldots, \widehat{R}_K^{(j)}\}_{j=1}^M$ are obtained from the reward generator for $\{\widehat{R}_{k'}|(H_{k'}, A_{k'})\}_{k'=k}^K$ estimated from deep conditional generative learning, and $\{A_{k'}\}_{k'=k}^K$ in this case is randomly generated from target policy $\pi$. Note that the whole procedure for estimating $\widehat{L}_k(H_k)$ only involves the information in the observed $H_k$.

---

**Algorithm 2** Pseudo Code of Quantile Treatment Effect Evaluation in Multiple Stage

---

**Input** Data trajectories $\{(X_{1,i}, A_{1,i}, R_{1,i}, X_{2,i}, A_{2,i}, R_{2,i}, \ldots, X_{K,i}, A_{K,i}, R_{K,i}) : 1 \le i \le N\}$;
 quantile levels $\tau$; number of folds $S$; a large integer $M$.

**Output** doubly robust quantile estimator $\widehat{\eta}_\tau^{\mathrm{DR}}$; tail-robust DR mean estimator $\hat\mu_{\mathrm{DR}}$.

1: **procedure** A. DOUBLY ROBUST QUANTILE ESTIMATOR
2:   Randomly split the data into $S$ folds. For any $s \in S$, denotes $\mathcal{I}_s$ as the $s$th subgroup.
3:   **for all** $s \in S$ **do**
4:     Estimate $\{\widehat{b}_k\}_{k=1}^K$ by GBDT with the data in $\mathcal{I}_s^c$;
5:     Estimate $\{\widehat{R}_k|(H_k, A_k)\}_{k=1}^K$ by MDN with the data in $\mathcal{I}_s^c$;
6:     Generate $\{\widehat{R}_{h,a}^j|(H_k = H_{k,i}, A_k = a)\}_{j=1}^M$ for any $a \in \mathcal{A}$ and $i \in \mathcal{I}_s$;
7:   **end for**
8:   Solve the optimization problem in formula (11) to obtain $\widehat{\eta}_{\mathrm{DR}}$.
9: **end procedure**
10: **procedure** B. TAIL-ROBUST DR MEAN ESTIMATOR
11:   For all of the quantile levels of our interest, calculate $\widehat{\eta}_\tau$ by procedure A. Averaging among all quantiles yields the tail-robust DR mean Estimator $\widehat{\mu}_{\mathrm{DR}}$.
12: **end procedure**

---

The level of information we utilize in estimating $R^*(\pi)$ depends on how well the observed actions are in line with the target policy, which aligns with the core idea of double robustness.

In Step 3, the main difference from the single-stage case lies in the extended form of the estimators for $J_0$ in the multi-stage. Define $f_{R^*(\pi),s}$ as the conditional pdf of $R^*(\pi)|H_1$ obtained by some conditional generative learning approach from the data in $\mathcal{I}_s^c$. Then $J_0$ can be estimated by

**(i) DM estimator for $J_0$:**

$$\widehat{J}_0^{\mathrm{DM}} = \widehat{f}_{R^*(\pi)}(\eta) = \frac{1}{N} \sum_{s=1}^S \sum_{i \in \mathcal{I}_s} \widehat{f}_{R^*(\pi),s}(\widehat{\eta}_{\mathrm{DR}}|H_{1,i}).$$

**(i) IPW estimator for $J_0$:** Following the same logic in Equation 3, we have:

$$\widehat{J}_0^{\mathrm{IPW}} = \frac{1}{N} \sum_{s=1}^S \sum_{i \in \mathcal{I}_s} \prod_{k=1}^K \frac{\pi_k(A_{k,i}|H_{k,i})}{\widehat{b}_k(A_{k,i}|H_{k,i})} \frac{1}{h} K\left(\frac{\sum_{k=1}^K R_{i,k} - \widehat{\eta}_{\mathrm{DR}}}{h}\right)$$

**(i) DR estimator for $J_0$:**

$$\widehat{J}_0^{\mathrm{DR}} = \frac{1}{N} \sum_{s=1}^S \sum_{i \in \mathcal{I}_s} \left[ \frac{\prod_{k'=1}^K \pi_{k'}(A_{k',i}|H_{k',i})}{\prod_{k'=1}^K \widehat{b}_{k'}(A_{k',i}|H_{k',i})} \frac{1}{h} K\left(\frac{\sum_{k=1}^K R_{i,k} - \widehat{\eta}_{\mathrm{DR}}}{h}\right) \right.$$
$$\left. + \sum_{k=1}^K \left(\prod_{k'=1}^{k-1} \frac{\pi_{k'}(A_{k',i}|H_{k',i})}{\widehat{b}_{k'}(A_{k',i}|H_{k',i})}\right) \left(1 - \frac{\pi_k(A_{k,i}|H_{k,i})}{\widehat{b}_k(A_{k,i}|H_{k,i})}\right) \widehat{f}_{\bar{R}_k^*(\pi)}(\widehat{\eta}_{\mathrm{DR}}|H_{k,i}) \right] \tag{12}$$

where $\widehat{f}_{\bar{R}_k^*(\pi)}(\bullet|H_{k,i})$ is the estimated conditional pdf of $\bar{R}_k^*(\pi) := \sum_{k'=k}^K R_{k'}^*(\pi)$ given $H_k$ and $A_k \sim \pi$ obtained by MDN.

A Wald-type confidence interval can thus be constructed based on our DR quantile estimator $\widehat{\eta}_\tau^{\mathrm{DR}}$ and its DR variance estimator obtained from formula (12). We will detail this in the corollary of Section 4.

# 4   Theory

In this section, we provide asymptotic guarantees for the proposed doubly robust quantile estimator $\widehat{\eta}_\tau^{\mathrm{DR}}$.

Define $\psi(W_i; \eta, \hat{\alpha}) \equiv \partial_\eta \Psi(W_i; \eta, \hat{b}, \widehat{\mathbb{E}})$, which is the derivative of the objective function we are trying to optimize. Similarly, we define $\psi^*(W_i; \eta) = \partial_\eta \Psi(W_i; \eta, b, \mathbb{E})$ where the nuisance functions for $\hat{b}$ and $\widehat{\mathbb{E}}$ are replaced by their population functions[1].

Solving the optimization problem in (11) is thus equivalent to obtaining the solution $\hat{\eta}_\tau^{\mathrm{DR}}$ to the following estimating equation:

$$\frac{1}{N} \sum_{s=1}^{S} \sum_{i \in \mathcal{I}_s} \psi(W_i; \eta, \hat{\alpha}_s) = 0, \tag{13}$$

where $\hat{\alpha}_s$ is the collection of nuisance parameters in estimating $\hat{b}$ and $\widehat{\mathbb{E}}$ using data in $\mathcal{I}_s^c$.

## 4.1 The asymptotic properties of $\hat{\eta}_\tau^{\mathrm{DR}}$

Before we proceed, let's introduce the assumptions that will be used in this section.

**Assumption 1.** *We assume the following conditions hold:*

*(A1)* $\left\| \hat{b}_k(A_k|H_k) - b_k(A_k|H_k) \right\|_{P,2} \left\| \delta\left( \widehat{F}_{\bar{R}_k^*(\pi)|H_1}, F_{\bar{R}_k^*(\pi)|H_1} \right) \right\|_{P,2} = o(n^{-1/2}), \forall\ k \in \{1, \ldots, K\}.$

*(A2)* $f_{\bar{R}_k^*(\pi)|H_k}(r|H_k)$ *is uniformly bounded in* $r$ *and* $H_k$, $\forall\ k \in \{1, \ldots, K\}.$

*(A3)* $\exists$ *a constant* $C_1 > 0$, *such that* $f_{R^*(\pi)}(\eta) \geq C_1$ *holds for all* $\eta$ *in a neighbor of* $\eta_\tau.$

*(A4)* $\exists\ \epsilon > 0$, *s.t.* $\mathbb{P}(\epsilon \leq \hat{b}_k(A_k|H_k) \leq 1 - \epsilon) = 1$, $\forall\ H_k$ *and* $k \in \{1, \ldots, K\}.$

*(A5)* $\partial_r f_{R^*(\pi)}(r)\big|_{r=\eta_\tau}$ *is bounded.*

**Remark 1.** Assumptions (A1) measures the accuracy in estimating propensity score and outcome at each stage. Specifically, we require the product of the convergence rates of both the propensity score model and the outcome regression model to be at least $o(n^{-1/2})$, implying that each one of them must be $o(n^{-1/4})$ if both share the same convergence rate. This assumption is mild and can be achieved by various methods. For instance, estimators constructed via lookup tables typically attain the standard $o(n^{-1/2})$ convergence rate individually, yielding a product rate of $o(n^{-1})$, which satisfies Assumption (A1). Additionally, several machine-learning-based approaches, such as the "lassoed" gradient boosted tree algorithm (Schuler et al., 2022) for estimating the behavior policy, and mixture density networks (MDNs, see Zhou et al., 2022, Theorem 3) for estimating conditional distribution functions, also meet this requirement. Similar assumptions are commonly imposed in the literature (see, e.g., Chernozhukov et al., 2018; Farrell et al., 2021; Kallus & Uehara, 2022). In Assumption (A2), we require the conditional pdfs to be uniformly bounded, which aims to guarantee the continuity of the cdf of the corresponding reward functions. Assumption (A3) ensures the identifiability of the quantile $\eta_\tau$ of our interest, and Assumption (A4) is an adjunctive condition on the estimated propensity score, which is commonly assumed in related literature (Chernozhukov et al., 2018). Assumption (A5) requires the derivative of $f_{R^*(\pi)}(r)$ to be bounded only at the true value $\eta_\tau$. Since the true value is unknown to us, we may need a stronger condition that requires uniform boundedness on the support of $\eta$.

We first introduce the two lemmas that guarantees the double robustness of both the quantile estimator and its variance estimator.

**Lemma 1.** *(Double Robustness of the Quantile Estimator)*

*Suppose that Assumptions (A3)-(A4) are satisfied.* $\hat{\eta}_\tau^{DR}$ *is a consistent estimator of* $\eta_\tau$, *as long as one of the following two parts of models is consistently estimated:*

*(1) The propensity score functions at each stage, i.e.* $\{\hat{b}_k(H_k)\}_{k=1}^{K}.$

---

[1]The explicit expressions for $\psi(W_i; \eta, \hat{\alpha})$ and $\psi^*(W_i; \eta)$ are given in Appendix A.

(2) *The conditional expectation functions at each stage, i.e.* $\{\widehat{L}_k(H_k)\}_{k=1}^K$,
*or equivalently the data generating models for* $\{\widehat{R}_k|(H_k, A_k)\}_{k=1}^K$.

**Lemma 2.** *(Double Robustness of the Variance Estimator)*

*In kernel density estimation, suppose the kernel function $K$ is a real-valued, Borel-measurable function in $L^\infty(\mathbb{R})$ which satisfies $\lim_{x\to\infty} |xK(x)| = \lim_{x\to\infty} |xK^2(x)| = 0$. Define the bandwidth as $h_N$, which can be a function of the sample size. We assume $\lim_{N\to\infty} h_N = 0$, and $\lim_{N\to\infty} Nh_N = \infty$.*

*We claim that $\widehat{\sigma}_{DR}^2$ is a consistent estimator of $\sigma^2$, as long as one of the following two parts of models is consistently estimated:*

(1) *The propensity score functions at each stage, i.e.* $\{\widehat{b}_k(H_k)\}_{k=1}^K$.

(2) *The conditional expectation functions at each stage, i.e.* $\{\widehat{L}_k(H_k)\}_{k=1}^K$,
*or equivalently the data generating models for* $\{\widehat{R}_k|(H_k, A_k)\}_{k=1}^K$.

The proofs of Lemma 1 and Lemma 2 along with other auxiliary theoretical results are provided in Appendix C. Next, we will derive the asymptotic properties of our DR quantile estimator $\widehat{\eta}_\tau^{\text{DR}}$.

Define $\|\cdot\|_{P,q}$ as the $L^q(P)$ norm which satisfies $\|f(W)\|_{P,q} = \left(\int |f(\omega)|^q dP(\omega)\right)^{1/q}$. Let $\delta(F_1, F_2)$ denote the total variation distance between two probability measures, where $F_1$ and $F_2$ are the CDF of the two distributions. Define $f_{\bar{R}_k^*(\pi)|H_k}(r|H_k)$ and $F_{\bar{R}_k^*(\pi)|H_k}(r|H_k)$ as the true conditional pdf and cdf of $\bar{R}_k^*(\pi) = \sum_{k'=k}^K R_{k'}^*(\pi)$ given the historical data $H_k$, respectively. Similarly, define $\widehat{f}_{\bar{R}_k^*(\pi)|H_k}(r|H_k)$ and $\widehat{F}_{\bar{R}_k^*(\pi)|H_k}(r|H_k)$ as the conditional pdf and cdf of $\bar{R}_k^*(\pi)$ estimated by conditional distribution generator, respectively.

**Theorem 1.** *(Asymptotic Normality)*

*Suppose that Assumptions (A1)-(A5) are satisfied. When $M \to \infty$, $\widehat{\eta}_\tau^{DR}$ is asymptotically normal. Specifically,*

$$\sigma^{-1}\sqrt{N}(\widehat{\eta}_\tau^{DR} - \eta_\tau) = -\sigma^{-1}J_0^{-1}\left(\frac{1}{\sqrt{N}}\sum_{i=1}^N \psi^*(W_i; \eta_\tau)\right) + o_p(1) \xrightarrow{\mathcal{D}} \mathcal{N}(0, 1), \tag{14}$$

*where*

$$\sigma^2 = J_0^{-1}\mathbb{E}[\psi^{*2}(W; \eta_\tau)](J_0^{-1})', \quad and \quad J_0 = \partial_\eta\{\mathbb{E}_W[\psi^*(W; \eta)]\}|_{\eta=\eta_\tau}. \tag{15}$$

*Furthermore,*

$$J_0 = \partial_\eta\{\mathbb{E}_W[\psi^*(W; \eta)]\}|_{\eta=\eta_\tau} = \mathbb{E}_{H_1 \sim \mathbb{G}}\left[f_{R^*(\pi)|H_1}(\eta_\tau|H_1)\right] = f_{R^*(\pi)}(\eta_\tau). \tag{16}$$

*The results still hold when $\sigma^2$ is replaced by its doubly robust estimator $\widehat{\sigma}_{DR}^2$, given by*

$$\widehat{\sigma}_{DR}^2 = \left(\widehat{J}_0^{DR}\right)^{-1} \frac{1}{S}\sum_{s=1}^S \widehat{\mathbb{E}}_{n,s}[\psi^2(W; \widehat{\eta}_s, \hat{\alpha}_s)]\left(\widehat{J}_0^{DR}\right)^{-1}, \tag{17}$$

*where $\mathbb{E}_{n,s}$ denotes the empirical expectation calculated with the data in fold $s$, and $\widehat{J}_0^{DR}$ is estimated under the doubly robust framework, as we've discussed in formula (5) and (12).*

Estimation details are provided in Section 2.3.3 and Section 3.3 for both single-stage and multiple-stage settings.

**Corollary** *According to Theorem 1, a Wald-type $\alpha$-level confidence interval of $\eta_\tau$ can be constructed by*

$$CI = \left[\widehat{\eta}_\tau^{DR} \pm \frac{1}{\sqrt{N}}\Phi^{-1}(1-\alpha/2) \cdot \widehat{\sigma}_{DR}\right], \tag{18}$$

*where $\Phi$ is the cdf of the standard normal distribution. The confidence interval constructed in (18) achieves an asymptotic coverage rate $1-\alpha$.*

As illustrated in Lemma 1 and Theorem 1, our quantile estimator $\widehat{\eta}_\tau^{\mathrm{DR}}$ satisfies the double robustness property and is semi-parametrically efficient in both single stage and sequential decision making problems. The proof of Theorem 1 is summarized in Appendix C.4.

Theorem 2 below ensures that our DR quantile estimator is constructed from the efficient influence function (EIF) and thus enjoys semi-parametric efficiency.

**Theorem 2.** *(Semi-Parametric Efficiency)*

*Define $C(\eta) = -f_{R^*(\pi)}(\eta)$. Under the same conditions as proposed in Theorem 1, the EIF of $\eta_\tau$ is given by*

$$
\begin{aligned}
EIF_\eta = C(\eta_\tau) \cdot \mathbb{E} &\left[ \frac{\prod_{k'=1}^{K} \pi_{k'}(A_{k'}|H_{k'})}{\prod_{k'=1}^{K} b_{k'}(A_{k'}|H_{k'})} \left( \mathbb{I}\Big\{ \sum_{k=1}^{K} R_k < \eta \Big\} - \tau \right) \right. \\
&\left. + \sum_{k=1}^{K} \left( \prod_{k'=1}^{k-1} \frac{\pi_{k'}(A_{k'}|H_{k'})}{b_{k'}(A_{k'}|H_{k'})} \right) \left( 1 - \frac{\pi_k(A_k|H_k)}{b_k(A_k|H_k)} \right) F_{R^*(\pi)|H_k, \bar{A}_k}(\eta) \right],
\end{aligned}
\tag{19}
$$

*which alignes with the DR estimator we proposed in Equation 11. The efficiency bound for DR variance is given by Equation 17.*

The proof of semi-parametric efficiency is summarized in Appendix C.5.

# 5 Numerical Results

In this section, we will justify the performance of our DR quantile estimator in both single-stage and multi-stage settings under different levels of heavy-tailness of the reward distribution. Throughout this paper, we use student-t distributions with different degrees of freedom to measure the level of heavy tail. All source code and supplementary experimental results are available on our Github page.

In the following sections, we will first illustrate how close our estimated quantile is to the true reward distribution, and then report the empirical coverage probability of our estimator. To show the performance of quantile aggregation in estimating the mean outcome in the heavy-tailed cases, we compare our quantile-based DR mean estimator *Rquantile*, with the DR mean estimator *Rmean* used in common literature[2]. The details of the data-generating mechanism and additional numerical results are given in Appendix B.1.1.

Before we proceed, let's discuss how to choose the number of folds in cross fitting. Although a larger value of $S$ may yield intuitively better performance in quantile estimation, according to some comparison results in existing work (Chernozhukov et al., 2018), there is no strong relationship between the number of folds and estimation accuracy in the context of double machine learning. To balance the computational complexity and the potential advantage of a large number of $S$, we set $S = 5$ throughout the simulation and real data analysis. For ease of calculation, we first compute the DR quantile estimator under each fold $s$ and average over all folds to obtain our final estimator.

## 5.1 Quantile Estimation Performance

Figure 2 shows the estimation accuracy of our DR quantile estimator. As we can see, our DR quantile estimator performs quite well in evaluating the true reward distributions under the target policy of our interest.

## 5.2 Coverage Probability

According to the estimation details elaborated in Section 2.3 and 3.3, we calculate the quantile $\widehat{\eta}_\tau^{\mathrm{DR}}$ and its doubly robust variance estimator $\widehat{\sigma}_{\mathrm{DR}}^2$ by formula (17). For each quantile level $\tau$, we repeat the estimation procedure 500 times to calculate the empirical coverage probability, which is shown in Figure 3.

---

[2]The expression of the DR mean estimator is summarized in Appendix A.

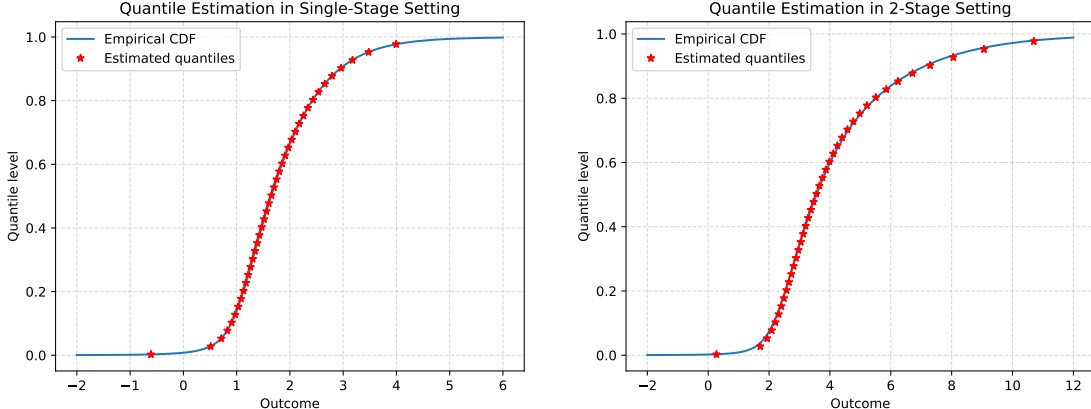

Figure 2: Quantile estimation result in single-stage (left) and two-stage (right) settings. The blue curve denotes the true reward distribution (the CDF under target policy $\pi$), and red stars denote the estimated quantiles at 20 equally spaced quantile levels. Details about the data generating processes are introduced in Appendix B.1.1.

Notice that when calculating the asymptotic variance, it is required to select a proper bandwidth in the kernel function. After comparing several commonly used bandwidth selection methods, we finally choose fixed bandwidth under this heavy-tailed circumstance due to its robustness in preventing the over-smoothing issue in kernel density estimation.[3]

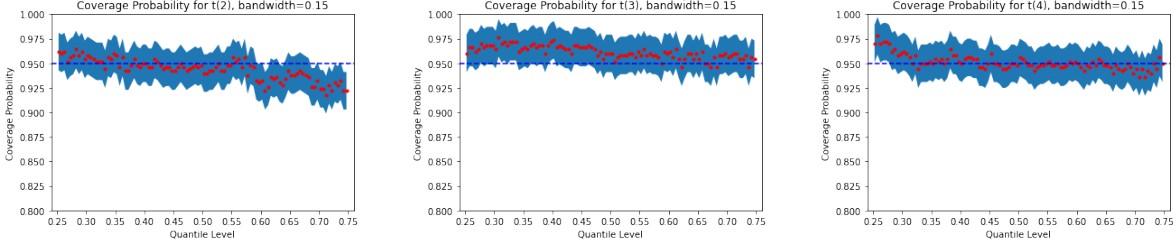

Figure 3: Coverage Probability under noise level $t(2)$, $t(3)$, and $t(4)$. The red dots correspond to the empirical coverage probability estimated from 500 times of replicates, and the blue area denotes the confidence band at each quantile level. It is clear from the plot that 0.95 falls into almost all of the confidence bands, which means that our CI achieves nominal coverage in most cases.

## 5.3   Comparison between *Rquantile* and *Rmean*

To compare our quantile-based mean DR estimator *Rquantile* with the classical DR mean estimator *Rmean*, we calculated the MSE for both estimators under different heavy-tailed levels of the reward distributions. Detailed comparisons are summarized in Table 1 and Table 2. As we can see from both tables, *Rquantile* always yields smaller MSE and standard deviation than *Rmean*. The more heavy-tailed the reward distribution is, the more powerful our method tends to be.

---

[3]See Appendix B.1.3 for details about bandwidth selection.

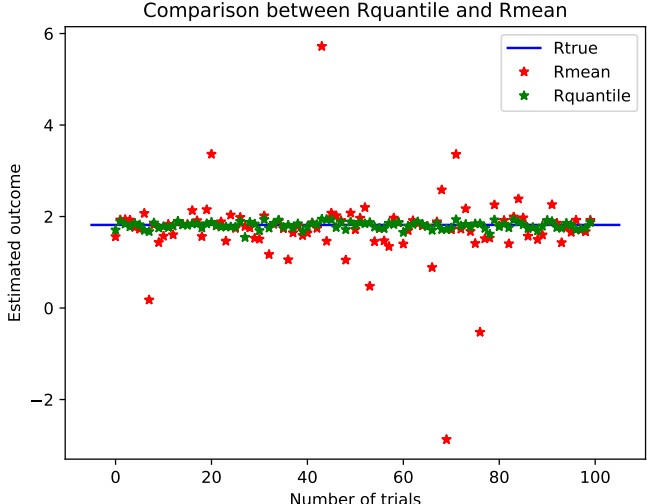

Figure 4: Comparison when the noise follows t(1.2) distribution. The blue line denotes the true value of mean reward, red stars denote the estimated value of *Rmean*, and green stars denote the estimated value of *Rquantile*. For comparisons across all levels of reward heavy-tailedness, please refer to this Github page.

Table 1: Single-stage comparison between *Rquantile* and *Rmean*. Each cell reports the MSE (with standard deviation in parentheses). We use 5-folds cross-fitting with $N = 2500$. All the results are obtained from 100 times of replicates.

| Heavy-tailed level | t(1.2) | t(1.5) | t(1.8) | t(2) | t(2.5) | t(3) | t(3.5) | t(4) |
|---|---|---|---|---|---|---|---|---|
| MSE of Rquantile | 0.00600 (0.009771) | 0.00169 (0.002250) | 0.00090 (0.001190) | 0.00099 (0.010338) | 0.00055 (0.000724) | 0.00043 (0.000557) | 0.00031 (0.000355) | 0.00037 (0.000547) |
| MSE of Rmean | 0.59478 (32.11465) | 0.03105 (2.852596) | 0.00203 (0.044305) | 0.00186 (0.043518) | 0.00062 (0.004603) | 0.00045 (0.003373) | 0.00032 (0.003202) | 0.00038 (0.002741) |

Table 2: Two-stage comparison between *Rquantile* and *Rmean*. Each cell reports the MSE (with standard deviation in parentheses). We use 5-folds cross-fitting with $N = 2500$. All the results are obtained from 100 times of replicates.

| Heavy-tailed level | t(2) | t(4) | t(6) | t(8) | $\mathcal{N}(0, 1)$ |
|---|---|---|---|---|---|
| MSE of Rquantile | 0.006708 (0.007954) | 0.002729 (0.003865) | 0.002447 (0.003007) | 0.002427 (0.002709) | 0.001549 (0.001541) |
| MSE of Rmean | 0.027780 (1.648467) | 0.003945 (0.027229) | 0.003558 (0.024007) | 0.003710 (0.023832) | 0.002062 (0.002024) |

To illustrate the advantage of our estimator more clearly, we visualize the estimation results of 100 times of replicates for both *Rquantile* and *Rmean* when the reward follows t(1.2) distribution, which is shown in Figure 4.

It is obvious from Figure 4 that under a super heavy-tailed setting, *Rquantile* outperforms *Rmean* in estimating the true reward expectation. There are some "outliers" when estimating *Rmean*, which performs much worse than other points. This is because the estimated propensity scores of some subjects are quite close to 0, leading to extremely unstable estimators for *Rmean*. On the contrary, our approach divides the mean estimation procedure into many quantile levels, and solving each estimating equation would not affect as much as what we may have in obtaining *Rmean*. This intuition explains the tail robustness of our estimator.

Since each quantile we estimate also enjoys the double robustness property, it's reasonable that *Rquantile* achieves better results than *Rmean.*

## 6    Real Data Analysis

In this section, we use a dataset collected from an advertisement experiment conducted at a world-leading tech company. The company plans to add an advertising position with three candidate choices. This experiment serves as an exploration applied to three groups of customers (180,000 users per group) randomly selected from daily active users. The company conduct causal inference on this data, and come up with four different personalized intervention policies. The key response of interest is the user watch time (the larger, the better), which has a heavy-tailed distribution as shown by the histogram, Q-Q plot and log-log plot in Figure 1.

We demonstrate that our method can (i) reliably infer the quantiles of revenues under the four policies, and (ii) accurately estimate their expected revenues using only a small subset of the full dataset. Toward that end, we repeat the following procedure 1,000 times: in each iteration, we randomly sample 1% of users from each group and apply our method to estimate 999 quantiles – from 0.001 to 0.999 – of the outcome distribution for each policy. Based on these estimators, we construct confidence intervals for the first, second, and third quartiles, and use the mean of the 999 quantiles as the estimated expected value. The ground-truth values of the four policies are obtained from online A/B tests. The mixture density network consists of two hidden layers with eight neurons each, and four mixture components.

Table 3 shows that the coverage probabilities of the 25%, 50% and 75% quantiles of the reward distribution of the four personalized policies are all close to the 95% nominal level. The bias, standard deviation and root MSE are presented in Table 4. Our doubly robust quantile-based mean estimator outperforms the traditional doubly robust mean estimator on this heavy-tailed real dataset. Furthermore, we also compare the performance among DM, IPW and DR estimators. Details are deferred to Appendix B.2 to save space.

Table 3: Coverage probabilities of the first, second, third quartiles of the proposed confidence intervals under the four personalized policies.

|      | $S_1$ | $S_2$ | $S_3$ | $S_4$ |
|------|-------|-------|-------|-------|
| 25%  | 0.926 | 0.944 | 0.933 | 0.911 |
| 50%  | 0.961 | 0.958 | 0.957 | 0.948 |
| 75%  | 0.949 | 0.926 | 0.942 | 0.915 |

Table 4: Comparison of biases, standard deviations and root MSEs of the expected revenue estimators between mean-based and the proposed quantile-based approaches. $S_1$, $S_2$, $S_3$, and $S_4$ denote the four personalized policies.

|       | Bias | | STD | | RMSE | |
|-------|-------|-----------|-------|-----------|-------|-----------|
|       | Rmean | Rquantile | Rmean | Rquantile | Rmean | Rquantile |
| $S_1$ | 0.006 | -0.013    | 0.076 | 0.069     | 0.076 | 0.071     |
| $S_2$ | -0.008 | -0.026   | 0.060 | 0.055     | 0.060 | 0.061     |
| $S_3$ | 0.007 | -0.014    | 0.066 | 0.059     | 0.067 | 0.061     |
| $S_4$ | 0.001 | -0.017    | 0.077 | 0.068     | 0.077 | 0.070     |

## 7    Conclusion and Discussions

In this paper, we conducted comprehensive research on the doubly robust off policy evaluation procedure for the entire reward distribution in sequential decision making problems. We constructed a doubly robust quantile estimator with theoretical guarantees, and provided an algorithm based on deep conditional generative learning, allowing a broad class of machine-learning-based tools to be utilized in estimating propensity scores and reward functions at each stage. Furthermore, a doubly robust variance estimator was also proposed to im-

prove the stability of the estimated confidence interval. Our quantile estimator enjoys asymptotic normality and semi-pamametric efficiency, which is well-illustrated from both simulation and real data analysis.

Based on our estimated DR quantiles, we proposed a tail-robust doubly-robust mean estimator which significantly outperforms the classical DR mean estimator used in the earlier literature. Both simulation and real data analysis illustrate the power of our estimator in decreasing the MSE when the reward is heavy-tailed.

The quantile-based mean estimator has demonstrated significant advantages over classical mean estimators in various contexts, including single-stage (Wang & Zhou, 2010) and infinite-horizon scenarios such as reinforcement learning (RL, Rowland et al., 2023). Given the promising results observed in both simulated and real-world data, further investigation into the theoretical properties of the proposed tail-robust doubly robust (DR) mean estimator would be valuable for future research. Additionally, it's quite interesting to explore the potential of developing optimal treatment regimes using the doubly robust quantile estimator introduced in this study. Another direction is to extend the binary actions to a discrete or continous action space, which would allow the entire framework to be applied to a wider range of real-life scenarios.

## Acknowledgement

Shi's research was partly supported by an EPSRC grant EP/W014971/1.

## Data Availability

Due to the sensitive nature of the dataset, it cannot be made publicly available.

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

## Appendix

## A    Assumptions and Notations

In this section, we aim to elaborate more on the explicit expressions of some assumptions and formulas mentioned in the main paper.

### A.1   Assumption (C1′)-(C3′)

In multi-stage setting, we also accept the potential outcome framework in causal inference.

(C1′) Consistency (or SUTVA) (Rubin, 1990): For any $k \in \{1, \ldots, K\}$,

$$R_k = R_k^*(\bar{A}_k) = \sum_{\bar{a}_k \in \bar{\mathcal{A}}_k} R^*(\bar{a}_k)\mathbb{I}\{\bar{A}_k = \bar{a}_k\};$$

When $k \geq 2$,

$$X_k = X_k^*(\bar{A}_{k-1}) = \sum_{\bar{a}_{k-1} \in \bar{\mathcal{A}}_{k-1}} X_k^*(\bar{a}_{k-1})\mathbb{I}\{\bar{A}_{k-1} = \bar{a}_{k-1}\}.$$

(C2′) Sequential Randomization Assumption (SRA) (Rosenbaum & Rubin, 1983): Given any treatment sequence $\bar{b}_K$, we have

$$A_k \perp \{R_k(\bar{b}_k), X_{k+1}(\bar{b}_k), R_{k+1}(\bar{b}_{k+1}), \ldots, R_K(\bar{b}_K)\}|H_k, \quad \forall k \in \{1, \ldots, K\}.$$

(C3′) Positivity Assumption: For any $k \in \{1, \ldots, K\}$, there exists a constant $\epsilon > 0$, s.t. $\mathbb{P}(\epsilon \leq b_k(A_k|H_k) \leq 1 - \epsilon) = 1$ for any $A_k \in \{0, 1\}$.

Note that (C1′)-(C3′) are just natural extensions of (C1)-(C3) in multi-stage setting, which is also widely assumed in causal inference literature.

### A.2   $\psi(W_i; \eta, \hat{\alpha})$ and $\psi^*(W_i; \eta)$

The estimating equation of our interest is defined as

$$
\begin{aligned}
\psi(W_i; \eta, \hat{\alpha}) &\equiv \partial_\eta \Psi(W_i; \eta, \widehat{b}, \widehat{\mathbb{E}}) \\
&= \frac{1}{N} \sum_{s=1}^{S} \sum_{i \in \mathcal{I}_s} \left[ \frac{\prod_{k'=1}^{K} \pi_{k'}(A_{k',i}|H_{k',i})}{\prod_{k'=1}^{K} \widehat{b}_{k'}(A_{k',i}|H_{k',i})} \left( \mathbb{I}\Big\{ \sum_{k=1}^{K} R_{i,k} < \eta \Big\} - \tau \right) \right. \\
&\qquad\qquad \left. + \sum_{k=1}^{K} \left( \prod_{k'=1}^{k-1} \frac{\pi_{k'}(A_{k',i}|H_{k',i})}{\widehat{b}_{k'}(A_{k',i}|H_{k',i})} \right) \left( 1 - \frac{\pi_k(A_{k,i}|H_{k,i})}{\widehat{b}_k(A_{k,i}|H_{k,i})} \right) \widehat{l}_k(H_{k,i}) \right],
\end{aligned}
\tag{20}
$$

where $\widehat{l}_k(H_k) = \widehat{\mathbb{E}}\left[ \mathbb{I}\{R_1 + \cdots + R_{k-1} + \widehat{R}_k + \cdots + \widehat{R}_K < \eta\} - \tau | \{A_{k'}\}_{k'=1}^{k-1}, \{A_{k'}\}_{k'=k}^{K} \sim \pi, X_1 \right]$, in which $\widehat{R}_k, \ldots, \widehat{R}_K$ are the estimated rewards obtained by the data generating models following treatment $(\pi_k, \ldots, \pi_K)$ starting from stage $k$.

When the nuisance functions are replaced by the true models, we have

$$
\begin{aligned}
\psi^*(W_i; \eta) = \frac{1}{N} \sum_{s=1}^{S} \sum_{i \in \mathcal{I}_s} \left[ \frac{\prod_{k'=1}^{K} \pi_{k'}(A_{k',i}|H_{k',i})}{\prod_{k'=1}^{K} b_{k'}(A_{k',i}|H_{k',i})} \left( \mathbb{I}\Big\{ \sum_{k=1}^{K} R_{i,k} < \eta \Big\} - \tau \right) \right. \\
\left. + \sum_{k=1}^{K} \left( \prod_{k'=1}^{k-1} \frac{\pi_{k'}(A_{k',i}|H_{k',i})}{b_{k'}(A_{k',i}|H_{k',i})} \right) \left( 1 - \frac{\pi_k(A_{k,i}|H_{k,i})}{b_k(A_{k,i}|H_{k,i})} \right) l_k(H_{k,i}) \right].
\end{aligned}
\tag{21}
$$

In the rest of this appendix, we will focus on the case $K = 2$. Therefore, the expressions become

$$
\begin{aligned}
\psi(W_i; \eta, \hat{\alpha}) &\equiv \partial_\eta \Psi(W_i; \eta, \hat{b}, \widehat{\mathbb{E}}) \\
&= \left[ \frac{\pi_1(A_{1,i}|H_{1,i})\pi_2(A_{2,i}|H_{2,i})}{\hat{b}_1(A_{1,i}|H_{1,i}, \hat{\alpha})\hat{b}_2(A_{2,i}|H_{2,i}, \hat{\alpha})} (\mathbb{I}\{R_{1,i} + R_{2,i} < \eta\} - \tau) \right. \\
&\quad + \left(1 - \frac{\pi_1(A_{1,i}|H_{1,i})}{\hat{b}_1(A_{1,i}|H_{1,i})}\right) \widehat{\mathbb{E}}[(\mathbb{I}\{\widehat{R}_1 + \widehat{R}_2 < \eta\} - \tau)|X_{1,i}, (A_{1,i}, A_{2,i}) \sim \pi] \\
&\quad \left. + \left(\frac{\pi_1(A_{1,i}|H_{1,i})}{\hat{b}_1(A_{1,i}|H_{1,i})}\right) \left(1 - \frac{\pi_2(A_{2,i}|H_{2,i})}{\hat{b}_2(A_{2,i}|H_{2,i})}\right) \widehat{\mathbb{E}}[(\mathbb{I}\{R_{1,i} + \widehat{R}_2 < \eta\} - \tau)|H_{2,i}, A_{2,i} \sim \pi] \right],
\end{aligned}
\tag{22}
$$

and

$$
\begin{aligned}
\psi^*(W_i; \eta) &= \left[ \frac{\pi_1(A_{1,i}|H_{1,i})\pi_2(A_{2,i}|H_{2,i})}{b_1(A_{1,i}|H_{1,i})b_2(A_{2,i}|H_{2,i})} (\mathbb{I}\{R_{1,i} + R_{2,i} < \eta\} - \tau) + a_1^* \widehat{\mathbb{E}}[(\mathbb{I}\{R_1 + R_2 < \eta\} \right. \\
&\quad \left. -\tau)|X_{1,i}, (A_{1,i}, A_{2,i}) \sim \pi] + a_2^* \widehat{\mathbb{E}}[(\mathbb{I}\{R_{1,i} + R_2 < \eta\} - \tau)|H_{2,i}, A_{2,i} \sim \pi] \right],
\end{aligned}
\tag{23}
$$

where we define $a_1^*$ and $a_2^*$ as

$$
a_1^* = \left(1 - \frac{\pi_1(A_{1,i}|H_{1,i})}{b_1(A_{1,i}|H_{1,i})}\right), \quad a_2^* = \left(\frac{\pi_1(A_{1,i}|H_{1,i})}{b_1(A_{1,i}|H_{1,i})}\right) \left(1 - \frac{\pi_2(A_{2,i}|H_{2,i})}{b_2(A_{2,i}|H_{2,i})}\right).
\tag{24}
$$

### A.3 *Rmean*

In simulation studies, we compare our quantile-based DR mean estimator with the traditional DR mean estimator used in common literature, named as *Rmean*. In single-stage setting, *Rmean* is estimated from

$$
Rmean = \frac{1}{n} \sum_{i=1}^n \left[ \frac{\pi_1(A_{1,i}|H_{1,i})}{\hat{b}_1(A_{1,i}|H_{1,i})} R_{1,i} + \left(1 - \frac{\pi_1(A_{1,i}|H_{1,i})}{\hat{b}_1(A_{1,i}|H_{1,i})}\right) \widehat{\mathbb{E}}[\widehat{R}_1|X_1 = X_{1,i}, A_1 \sim \pi] \right].
\tag{25}
$$

In two-stage settings,

$$
\begin{aligned}
Rmean = \frac{1}{n} \sum_{i=1}^n &\left[ \frac{\pi_1(A_{1,i}|H_{1,i})\pi_2(A_{2,i}|H_{2,i})}{\hat{b}_1(A_{1,i}|H_{1,i})\hat{b}_2(A_{2,i}|H_{2,i})} (R_{1,i} + R_{2,i}) \right. \\
&\quad + \left(1 - \frac{\pi_1(A_{1,i}|H_{1,i})}{\hat{b}_1(A_{1,i}|H_{1,i})}\right) \widehat{\mathbb{E}}[(\widehat{R}_1 + \widehat{R}_2)|H_1 = H_{1,i}, (A_1, A_2) \sim \pi] \\
&\quad \left. + \left(\frac{\pi_1(A_{1,i}|H_{1,i})}{\hat{b}_1(A_{1,i}|H_{1,i})}\right) \left(1 - \frac{\pi_2(A_{2,i}|H_{2,i})}{\hat{b}_2(A_{2,i}|H_{2,i})}\right) \widehat{\mathbb{E}}[(R_{1,i} + \widehat{R}_2)|H_2 = H_{2,i}, A_2 \sim \pi] \right].
\end{aligned}
\tag{26}
$$

## B More on Simulation and Real Data Analysis

### B.1 Simulation

### B.1.1 Data Generating Process

We first introduce the data generating processes in both single stage and multiple stage settings.

In single stage quantile estimation problem, we generate the data as follows:

$$
\begin{aligned}
X &\sim \mathbb{G} = \mathcal{N}(0, 1); \\
A &= b(X), \text{ where } b(X) = \mathbb{I}\{X + \frac{1}{4}\epsilon > 0\}; \\
R &= (1 - X + 2AX)(1 + \frac{1}{4}\epsilon'),
\end{aligned}
\tag{27}
$$

where $\epsilon$ and $\epsilon'$ are noise terms with different levels of heavy tail. The more heavy-tailed $\epsilon$ and $\epsilon'$ are, the harder it would be to estimate the reward distribution. As mentioned in the main paper, we use the student-t distribution with different degrees of freedom to generate heavy-tailed distributions. To illustrate the performance of our estimator under different levels of heavy tail, we let $\epsilon$ and $\epsilon'$ follow t distribution t(1.5), t(1.8), t(2), t(2.5), t(3), t(3.5), and t(4) respectively. The target policy here is defined as $\pi(A|X) = \mathbb{I}\{X > 0\}$.

Let's take t(3) as an example. Figure 5 shows the PDF of reward $R^*(\pi)$ when $\epsilon$ and $\epsilon'$ follows t distribution with $df = 3$. From the probability plot shown in Figure 6, the rewards generated in this setting, shown by the blue dots, aligns very well with the baseline distribution t(3) which is shown by the red line. Therefore, the heavy tail level of $\epsilon$ and $\epsilon'$ indeed represents the heavy tail level of the reward distribution.

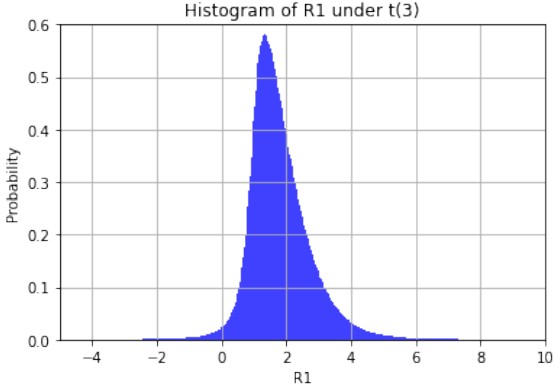
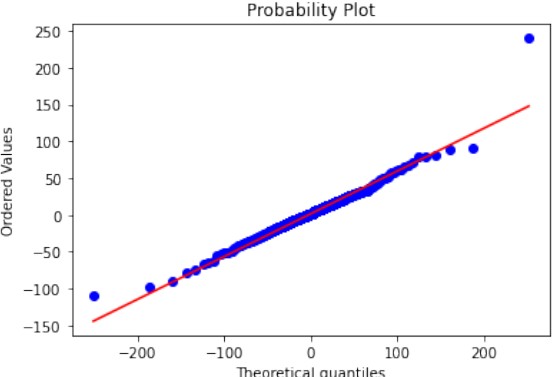

Figure 5: PDF of $R^*(\pi)$ when $\epsilon \sim$ t(3)

Figure 6: Probability plot of $R^*(\pi)$ compared with baseline distribution t(3), shown by the red line. The reward distribution is indeed heavy-tailed.

In multiple stage settings, we will only consider the case when $K = 2$ for illustration purpose. The observational covariates-action-reward triplet $(X_1, A_1, R_1, X_2, A_2, R_2)$ is generated as follows:

$$
\begin{aligned}
&X_1 \sim \mathbb{G} = \mathcal{N}(0, 1); \\
&A_1 | X_1 = b_1(X_1), \text{ where } b_1(X_1) = \mathbb{I}\{X_1 + \frac{1}{4}\epsilon_1 > 0\}; \\
&R_1 | (X_1, A_1) = (1 - X_1 + 2A_1 X_1)(1 + \frac{1}{4}\epsilon_2); \\
&X_2 | (X_1, A_1) \sim \frac{1}{2}X_1 + \frac{1}{2}\epsilon_3; \\
&A_2 | (X_1, A_1, X_2) = b_2(H_2), \text{ where } b_2(H_2) = \mathbb{I}\{X_2 + \frac{1}{4}\epsilon_4 > 0\}; \\
&R_2 | (X_1, A_1, X_2, A_2) = (1 + 0.5X_1 + A_1 X_1 - X_2 + 3A_2 X_2)(1 + \frac{1}{4}\epsilon_5),
\end{aligned}
\tag{28}
$$

where we define $\epsilon = (\epsilon_1, \epsilon_2, \epsilon_3, \epsilon_4, \epsilon_5)$ as the noise terms which denote student-t distributions with different degrees of freedom to control the heavy tail level of the cumulative reward distribution. The target policy sequence at each stage is defined as

$$
\begin{aligned}
\pi_1(A_1|X_1) &= \mathbb{I}\{X_1 > 0\}, \\
\pi_2(A_2|X_1, A_1, X_2) &= \mathbb{I}\{X_2 > 0\}.
\end{aligned}
\tag{29}
$$

Consider the cases when $\epsilon$ follows t(2), t(4), t(6), t(8) and $\mathcal{N}(0, 1)$. Unlike the single-stage setting, here we use relatively larger degrees of freedom in t distribution since the estimation is harder when the number of decision stages increases.

The PDF of $R^*(\pi) = R_1^*(\pi) + R_2^*(\pi)$ is shown in Figure 7, and the probability plot of $R^*(\pi)$ comparing with t(2) distribution is shown in Figure 8. The true reward distribution $R^*(\pi)$ is indeed quite heavy-tailed.

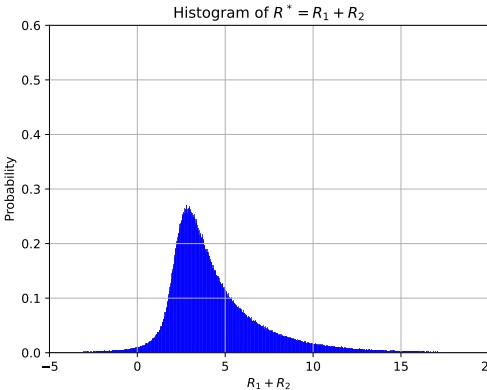
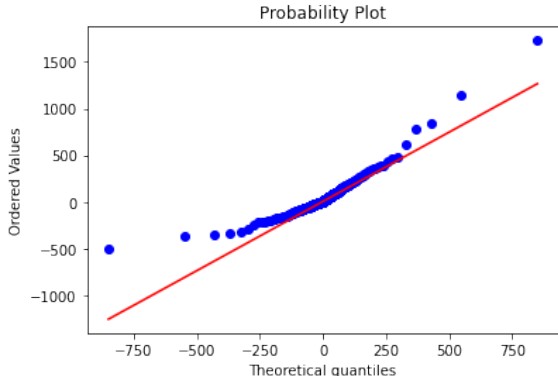

Figure 7: PDF of $R^*(\pi)$ when $\epsilon \sim \mathrm{t}(2)$

Figure 8: Probability plot of $R^*(\pi)$ compared with baseline distribution $\mathrm{t}(2)$, shown by the red line.

### B.1.2 Quantile Estimation Bias and Variance

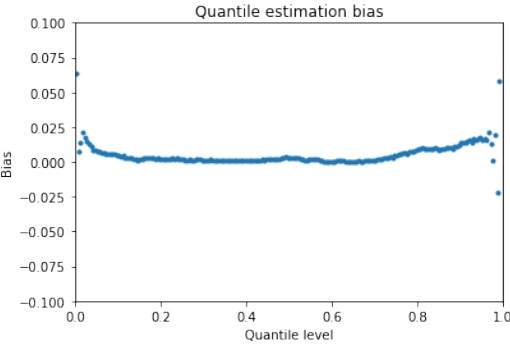
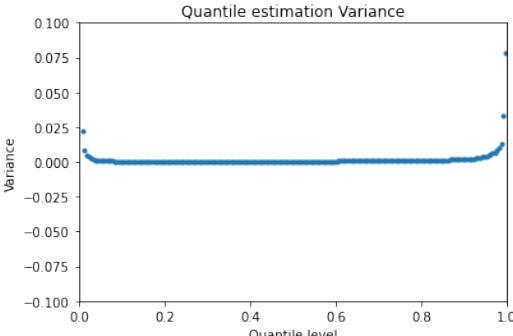

Figure 9: Quantile estimation bias and variance at different quantile levels. The estimation bias and variance are quite close to 0 at most of the quantile levels in the middle, and the performance tends to become unstable when $\tau$ approaches 0 or 1. The poor performance at extreme quantile levels is reasonable because of the lack of data points, which makes the statistical inference even harder to handle.

### B.1.3 Bandwidth Selection

Since the estimation of $\widehat{J}_0^{\mathrm{DR}}$ involves KDE, the choice of bandwidth $h$ is also an important problem to deal with. In Figure 10, we tried three commonly used methods: cross-validation, Scott's method, and fixed bandwidths where $h = 0.10$, $0.15$ and $0.20$, to compare their performances in estimating the standard error of $\widehat{\eta}_\tau^{\mathrm{DR}}$. Surprisingly, both cross-validation and Scott's method encountered some over-smoothing issues, which is potentially caused by the heavy tail of the reward distribution. On the contrary, a reasonable fixed bandwidth tends to stabilize the estimation of $\widehat{J}_0^{\mathrm{DR}}$, yielding a smaller MSE of the standard error estimated at all quantile levels. Since the performances under different fixed bandwidths are all pretty well, we will fix $h = 0.15$ in simulation studies.

### B.1.4 Comparison between DM, IPW and DR

As we can see from the comparisons in Figure 11, the performance of DM is clearly worse than IPW and DR. The performance of IPW estimator was not badly influenced due to the similarity of behavior policy and target policy under this specific simulation setting. As is expected to us, our DR estimator always yields the best result.

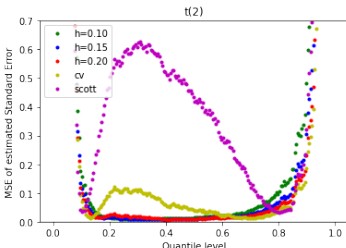 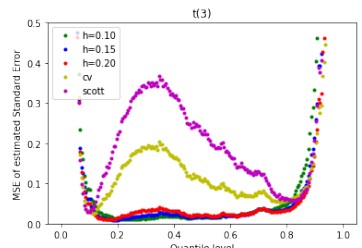 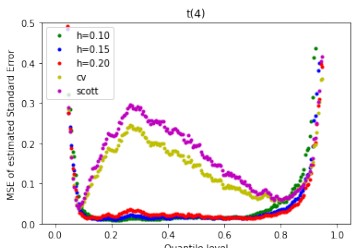

Figure 10: The comparison between the MSE of the standard error at each quantile level $\tau$ with noise distribution t(2), t(3), and t(4).

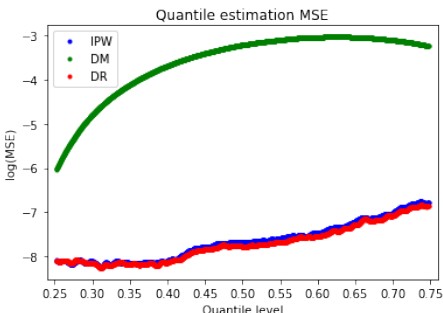 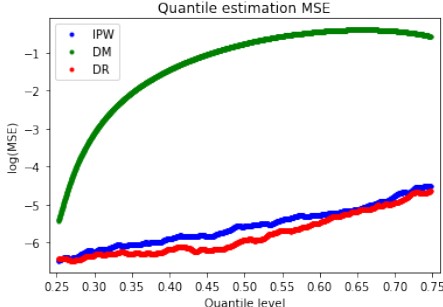

Figure 11: The comparison between DM, IPW, and DR estimators with heavy tail level t(4). Logarithm transformation was taken to better distinguish the performance of the three methods.

### B.2 Real Data Analysis

In this section, we add the comparison between DM, IPW and DR in our real dataset.

As show in Figure 12, our doubly robust approach performs the best in terms of bias, standard deviation and root mean squared errors, for all the four policies we considered.

## C More on Theory Section

In this section, we provide several lemmas and proofs that are omitted in the main paper.

### C.1 Lemma 3: Some Basic Results

**Lemma 3.** *Suppose* $W = \{(X_k, A_k, R_k)\}_{k=1}^K$ *is the full data with baseline information* $X_1 \sim \mathbb{G}$, *and* $\eta_\tau$ *is the $\tau$th quantile of the potential cumulative reward function* $R^*(\pi)$. *Then we have*

$$\mathbb{E}_W[\psi^*(W; \eta_\tau)] = 0. \tag{30}$$

*Proof.* For the brevity of content, we'll only show the proof when $K = 2$. The results in $K$ stage settings can be derived using the same logic.

To prove Lemma 3, we first separate $\psi^*(W; \eta_\tau)$ into three parts. Let $\psi(W^*; \eta_\tau) = a + b + c$, where

$$
\begin{aligned}
a &= \frac{\pi_1(A_1|H_1)\pi_2(A_2|H_2)}{b_1(A_1|H_1)b_2(A_2|H_2)}(\mathbb{I}\{R_1 + R_2 < \eta_\tau\} - \tau), \\
b &= a_1^* \cdot \widehat{\mathbb{E}}[(\mathbb{I}\{R_1 + R_2 < \eta_\tau\} - \tau)|X_1, (A_1, A_2) \sim \pi], \\
c &= a_2^* \cdot \widehat{\mathbb{E}}[(\mathbb{I}\{R_1 + \widehat{R}_2 < \eta_\tau\} - \tau)|H_2, A_2 \sim \pi],
\end{aligned}
\tag{31}
$$

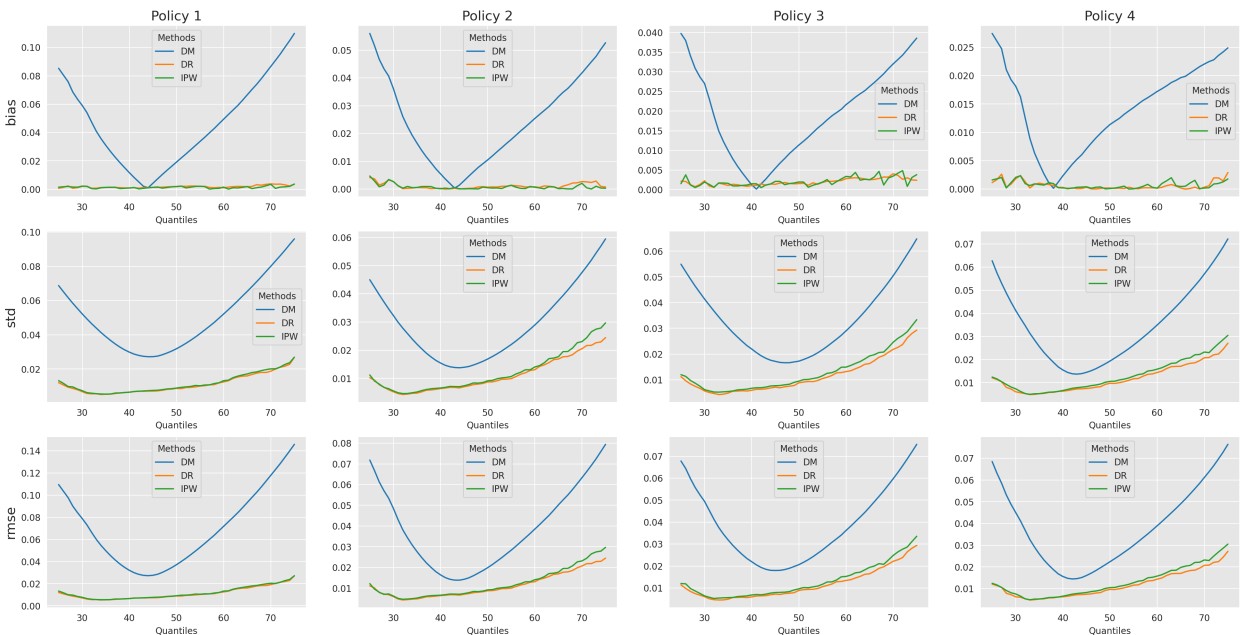

Figure 12: Bias, standard deviation, and root mean squared error of quantiles over 100 replicates. The four columns correspond to four target policies, and each plot shows the comparison of three methods: DM-direct method, IPW-inverse probability weighting method and DR-doubly robust method.

where $a_1^*$ and $a_2^*$ are defined in Formula (24). Now it suffices to show $\mathbb{E}[a] = \mathbb{E}[b] = \mathbb{E}[c] = 0$.

**1. $\mathbb{E}[a] = 0$.**

By change of measure theorem,

$$
\begin{aligned}
\mathbb{E}[a] &= \mathbb{E}\left[\frac{\pi_1(A_1|H_1)\pi_2(A_2|H_2)}{b_1(A_1|H_1)b_2(A_2|H_2)}(\mathbb{I}\{R_1 + R_2 < \eta_\tau\} - \tau)\right] \\
&= \mathbb{E}\left[\mathbb{I}\{R_1^*(\pi) + R_2^*(\pi) < \eta_\tau\}\right] - \tau = \tau - \tau = 0.
\end{aligned}
$$

**2. $\mathbb{E}[b] = 0$.**

$$
\begin{aligned}
\mathbb{E}[b] &= \mathbb{E}\left[\left(1 - \frac{\pi_1(A_1|H_1)}{b_1(A_1|H_1)}\right)\widehat{\mathbb{E}}[(\mathbb{I}\{R_1 + R_2 < \eta_\tau\} - \tau)|X_1, (A_1, A_2) \sim \pi]\right] \\
&= \mathbb{E}\left[\mathbb{E}\left[\left(1 - \frac{\pi_1(A_1|H_1)}{b_1(A_1|H_1)}\right)\Big|H_1\right] \cdot \mathbb{E}\left[(\mathbb{I}\{R_1^*(\pi_1) + R_2^*(\pi) < \eta_\tau\} - \tau)\right]\right] \\
&= \mathbb{E}\left[0 \cdot \mathbb{E}\left[(\mathbb{I}\{R_1^*(\pi_1) + R_2^*(\pi) < \eta_\tau\} - \tau)\right]\right] = 0.
\end{aligned}
\tag{32}
$$

**3. $\mathbb{E}[c] = 0$.**

Similarly, since

$$
\begin{aligned}
\mathbb{E}[a_2^*|H_2] &= \mathbb{E}\left[\left(\frac{\pi_1(A_1|H_1)}{b_1(A_1|H_1)}\right) \cdot \left(1 - \frac{\pi_2(A_2|H_2)}{b_2(A_2|H_2)}\right)\Big|H_2\right] \\
&= \mathbb{E}\left[\left(\frac{\pi_1(A_1|H_1)}{b_1(A_1|H_1)}\right) \cdot \mathbb{E}\left[\left(1 - \frac{\pi_2(A_2|H_2)}{b_2(A_2|H_2)}\right)\Big|H_2\right]\right] = \mathbb{E}\left[\left(\frac{\pi_1(A_1|H_1)}{b_1(A_1|H_1)}\right) \cdot 0\right] = 0,
\end{aligned}
\tag{33}
$$

then we have

$$
\begin{aligned}
\mathbb{E}[c] &= \mathbb{E}\left[a_2^* \cdot \widehat{\mathbb{E}}[(\mathbb{I}\{R_1 + \widehat{R}_2 < \eta\} - \tau)|H_2, A_2 \sim \pi]\right] \\
&= \mathbb{E}\left[\mathbb{E}\left[a_2^*|H_2\right] \cdot \widehat{\mathbb{E}}[(\mathbb{I}\{R_1^*(\pi_1) + R_2^*(\pi) < \eta\} - \tau)|H_2, A_2 \sim \pi]\right] \\
&= \mathbb{E}[0 \cdot \widehat{\mathbb{E}}[(\mathbb{I}\{R_1^*(\pi_1) + R_2^*(\pi) < \eta\} - \tau)|H_2, A_2 \sim \pi]] = 0.
\end{aligned}
\tag{34}
$$

The proof is thus complete. □

## C.2 Proof of Lemma 1: Double Robustness of our quantile Estimator

For illustration purpose, we will focus on the case where $K = 2$ for simplicity.

*Proof.* Suppose without the loss of generality that the sample size According to our estimating procedure, the quantile estimator $\widehat{\eta}_\tau^{\mathrm{DR}}$ is the solution to the following estimating equation:

$$
\frac{1}{S} \sum_{s=1}^{S} \mathbb{E}_{n,s}\left[\psi(W_s, \eta, \hat{\alpha}_s)\right] = 0,
\tag{35}
$$

where $W_s$ denotes the data in $\mathcal{I}_s$, and $\mathbb{E}_{n,s}$ is the empirical expectation over $s$th subgroup.

Suppose that $\Delta_n$ is a sequence of positive numbers that converge to 0. We define $\mathcal{N}_{\Delta_n}(\eta_\tau) = \{\eta : \|\eta - \eta_\tau\| \leq \Delta_n\}$ as a shrinking neighborhood of $\eta_\tau$. According to Uniform Law of Large Numbers (ULLN), it suffices to show the following three conditions hold:

1. $\sup_{\eta \in \mathcal{N}_{\Delta_n}(\eta_\tau)} \left|\frac{1}{S}\sum_{s=1}^{S} \mathbb{E}_{n,s}\left[\psi(W; \eta, \hat{\alpha})\right] - \mathbb{E}\left[\psi^*(W; \eta)\right]\right| = o_p(1)$;

2. $\forall \epsilon > 0, \inf\{|\mathbb{E}\left[\psi^*(W; \eta)\right]| : \|\eta - \eta_\tau\| \geq \epsilon\} > 0 = \mathbb{E}\left[\psi^*(W; \eta_\tau)\right]$;

3. $\frac{1}{S}\sum_{s=1}^{S} \mathbb{E}_{n,s}\left[\psi(W; \widehat{\eta}_\tau^{\mathrm{DR}}, \hat{\alpha})\right] = o_p(1)$.

Condition 2 can be easily proved by the identifiability of $\eta_\tau$. According to Assumption (A3), there exists a positive constant $C_1$, such that

$$
\partial_\eta\{\mathbb{E}_W[\psi^*(W; \eta)]\} = f_{R^*(\pi)}(\eta) \geq C_1
\tag{36}
$$

for all $\eta \in \mathcal{N}_{\Delta_n}(\eta_\tau)$. According to Lemma 1, $\eta_\tau$ is the solution to $\mathbb{E}_W[\psi^*(W; \eta)] = 0$. Therefore, the claim of condition 2 follows.

Condition 3 guarantees the estimated $\widehat{\eta}_\tau^{\mathrm{DR}}$ to be close to the solution to the empirical estimating equation, which naturally holds under a valid estimating procedure. To complete the consistency proof, it remains to establish Condition 1.

Define the empirical process $\mathbb{G}_{n,s}\left[\psi(W_s; \eta, \hat{\alpha}_s)\right]$ as a linear operator on measurable functions $\psi$ via

$$
\mathbb{G}_{n,s}\left[\psi(W_s; \eta, \hat{\alpha}_s)\right] = \sqrt{n}\left[\frac{1}{n}\sum_{i \in \mathcal{I}_s} \psi(W_i; \eta, \hat{\alpha}_s) - \mathbb{E}\left[\psi(W; \eta, \hat{\alpha}_s)\right]\right].
$$

Since

$$
\begin{aligned}
&\sup_{\eta \in \mathcal{N}_{\Delta_n}(\eta_\tau)} \left|\frac{1}{S}\sum_{s=1}^{S} \mathbb{E}_{n,s}\left[\psi(W; \eta, \hat{\alpha})\right] - \mathbb{E}\left[\psi^*(W; \eta)\right]\right| \\
&\leq \frac{1}{S}\sum_{s=1}^{S}\left\{\frac{1}{\sqrt{n}} \cdot \sup_{\eta \in \mathcal{N}_{\Delta_n}(\eta_\tau)} |\mathbb{G}_{n,s}\left[\psi(W; \eta, \hat{\alpha}_s)\right]| + \sup_{\eta \in \mathcal{N}_{\Delta_n}(\eta_\tau)} |\mathbb{E}\left[\psi(W; \eta, \hat{\alpha}_s)\right] - \mathbb{E}\left[\psi^*(W; \eta)\right]|\right\},
\end{aligned}
$$

it suffices to show that for any $s \in \{1, \ldots, S\}$, the two terms on the RHS of the above inequality are $o_p(1)$. For the brevity of content, we omit the subscript $s$ in $\mathbb{G}_{n,s}$ and $\hat{\alpha}_s$ in the following proof to illustrate the general results that hold for any $s$.

**Step 1**: Prove that $\sup_{\eta \in N_{\Delta_n}(\eta_\tau)} |\mathbb{G}_n[\psi(W; \eta, \hat{\alpha})]| = o_p(\sqrt{n})$.

Define $\mathcal{F} = \{\psi(W; \eta, \hat{\alpha}) : \hat{\eta}_\tau^{\mathrm{DR}} \in \mathcal{N}_{\Delta_n}(\eta_\tau)\}$. $\mathcal{F}$ is a VC Class. By Conditioning on $(W_i)_{i \in \mathcal{I}^c}$, $\hat{\alpha}$ can be regarded as a fixed value. Therefore, according to the Maximal Inequality in VC type classes (Chernozhukov et al., 2014; 2018), there exist a sufficiently large constant $C$, such that with probability $1 - o(1)$,

$$\sup_{f \in \mathcal{F}} |\mathbb{G}_n(f)| \leq C \cdot \left( r_n \log^{1/2}(1/r_n) + n^{-1/2 + 1/q} \log n \right), \tag{37}$$

where $r_n = \sup_{\eta \in N_{\Delta_n}(\eta_\tau)} \|\psi(W; \eta, \hat{\alpha})\|_{P,2}$, and $q \geq 2$ is an integer that can be arbitrarily large. To be more specific,

$$
\begin{aligned}
r_n =& sup_{\eta \in N_{\Delta_n}(\eta_\tau)} \|\psi(W; \eta, \hat{\alpha})\|_{P,2} \\
=& sup_{\eta \in N_{\Delta_n}(\eta_\tau)} \left\| \frac{\pi_1(A_{1,i}|H_{1,i})\pi_2(A_{2,i}|H_{2,i})}{\hat{b}_1(A_{1,i}|H_{1,i},\hat{\alpha})\hat{b}_2(A_{2,i}|H_{2,i},\hat{\alpha})} (\mathbb{I}\{R_{1,i} + R_{2,i} < \eta\} - \tau) \right. \\
& + \left(1 - \frac{\pi_1(A_{1,i}|H_{1,i})}{\hat{b}_1(A_{1,i}|H_{1,i})}\right) \widehat{\mathbb{E}}[(\mathbb{I}\{\widehat{R}_1 + \widehat{R}_2 < \eta\} - \tau)|X_{1,i}, (A_{1,i}, A_{2,i}) \sim \pi] \\
& + \left. \left(\frac{\pi_1(A_{1,i}|H_{1,i})}{\hat{b}_1(A_{1,i}|H_{1,i})}\right)\left(1 - \frac{\pi_2(A_{2,i}|H_{2,i})}{\hat{b}_2(A_{2,i}|H_{2,i})}\right) \widehat{\mathbb{E}}[(\mathbb{I}\{R_{1,i} + \widehat{R}_2 < \eta\} - \tau)|H_{2,i}, A_{2,i} \sim \pi] \right\|_{P,2} \\
\leq& sup_{\eta \in N_{\Delta_n}(\eta_\tau)} \left\| \frac{\pi_1(A_{1,i}|H_{1,i})\pi_2(A_{2,i}|H_{2,i})}{\hat{b}_1(A_{1,i}|H_{1,i},\hat{\alpha})\hat{b}_2(A_{2,i}|H_{2,i},\hat{\alpha})} \right\|_{P,2} \cdot \|\mathbb{I}\{R_{1,i} + R_{2,i} < \eta\} - \tau\|_{P,2} \\
& + \left\|1 - \frac{\pi_1(A_{1,i}|H_{1,i})}{\hat{b}_1(A_{1,i}|H_{1,i})}\right\|_{P,2} \cdot \left\|\widehat{\mathbb{E}}[(\mathbb{I}\{\widehat{R}_1 + \widehat{R}_2 < \eta\} - \tau)|X_{1,i}, (A_{1,i}, A_{2,i}) \sim \pi]\right\|_{P,2} \\
& + \left\|\left(\frac{\pi_1(A_{1,i}|H_{1,i})}{\hat{b}_1(A_{1,i}|H_{1,i})}\right)\left(1 - \frac{\pi_2(A_{2,i}|H_{2,i})}{\hat{b}_2(A_{2,i}|H_{2,i})}\right)\right\|_{P,2} \cdot \left\|\widehat{\mathbb{E}}[(\mathbb{I}\{R_{1,i} + \widehat{R}_2 < \eta\} - \tau)|H_{2,i}, A_{2,i} \sim \pi]\right\|_{P,2} \\
\leq& \frac{1}{\epsilon^2} \cdot 2 + \frac{1}{\epsilon} \cdot 2 + \frac{1}{\epsilon^2} \cdot 2 \leq \frac{6}{\epsilon^2},
\end{aligned}
$$

where the second last inequality is obtained from the Positivity Assumption (C3) and Assumption (A4). Therefore, $r_n = O(1)$, and we have $\sup_{f \in \mathcal{F}} |\mathbb{G}_n(f)| \leq O_p(1) = o(\sqrt{n})$.

**Step 2**: Prove that $\sup_{\eta \in N_{\Delta_n}(\eta_\tau)} |\mathbb{E}[\psi(W; \eta, \hat{\alpha})] - \mathbb{E}[\psi^*(W; \eta)]| = o_p(1)$.

In the following proof, we use $\widehat{\mathbb{E}}$ to denote the error aggregated by Monte Carlo Method, and use $\widehat{R}_k$ to denote the estimated reward obtained by MDN. Likewise, $\widehat{R}_k^*(\pi)$ is to denote the estimated potential outcome under policy $\pi$ at the $k$th stage. When $M \to \infty$, no error comes from the Monte Carlo method, and thus $\widehat{\mathbb{E}} = \mathbb{E}$.

Let's first start with $\mathbb{E}[\psi(W; \eta, \hat{\alpha})]$.

$$
\begin{aligned}
& \mathbb{E}[\psi(W; \eta, \hat{\alpha})] \\
=& \mathbb{E}\left[\left(\frac{\pi_1(A_{1,i}|H_{1,i})}{\hat{b}_1(A_{1,i}|H_{1,i})}\right)\left(\frac{\pi_2(A_{2,i}|H_{2,i})}{\hat{b}_2(A_{2,i}|H_{2,i})}\right) \cdot (\mathbb{I}\{R_{1,i} + R_{2,i} < \eta\} - \tau) \right. \\
& + \left(1 - \frac{\pi_1(A_{1,i}|H_{1,i})}{\hat{b}_1(A_{1,i}|H_{1,i})}\right) \cdot \widehat{\mathbb{E}}\left[\mathbb{I}\{\widehat{R}_1 + \widehat{R}_2 < \eta\} - \tau \big| X_{1,i}, (A_{1,i}, A_{2,i}) \sim \pi\right] \\
& + \left. \left(\frac{\pi_1(A_{1,i}|H_{1,i})}{\hat{b}_1(A_{1,i}|H_{1,i})}\right)\left(1 - \frac{\pi_2(A_{2,i}|H_{2,i})}{\hat{b}_2(A_{2,i}|H_{2,i})}\right) \cdot \widehat{\mathbb{E}}\left[\mathbb{I}\{R_{1,i} + \widehat{R}_2 < \eta\} - \tau \big| H_{2,i}, A_{2,i} \sim \pi\right]\right],
\end{aligned}
$$

which is equivalent to

$$
\begin{aligned}
=&\mathbb{E}\Bigg[\left(\frac{\pi_1(A_{1,i}|H_{1,i})}{\widehat{b}_1(A_{1,i}|H_{1,i})}\right)\left(\frac{\pi_2(A_{2,i}|H_{2,i})}{\widehat{b}_2(A_{2,i}|H_{2,i})}\right)\cdot\left(\mathbb{E}[\mathbb{I}\{R_{1,i}+R_2^*(\pi)<\eta\}|H_{2,i}]-\widehat{\mathbb{E}}[\mathbb{I}\{R_{1,i}+\widehat{R}_2^*(\pi)<\eta\}|H_{2,i}]\right)\\
&+\left(\frac{\pi_1(A_{1,i}|H_{1,i})}{\widehat{b}_1(A_{1,i}|H_{1,i})}\right)\widehat{\mathbb{E}}[\mathbb{I}\{R_{1,i}+\widehat{R}_2^*(\pi)<\eta\}-\tau|H_{2,i}]\\
&+\left(1-\frac{\pi_1(A_{1,i}|H_{1,i})}{\widehat{b}_1(A_{1,i}|H_{1,i})}\right)\widehat{\mathbb{E}}[\mathbb{I}\{\widehat{R}_1^*(\pi)+\widehat{R}_2^*(\pi)<\eta\}-\tau|H_{1,i}]\Bigg]:=L1+L2+L3,
\end{aligned}\tag{38}
$$

where we define the last three lines in (38) as $L1$, $L2$ and $L3$. Then

$$
\begin{aligned}
L2=&\mathbb{E}\left[\left(\frac{\pi_1(A_{1,i}|H_{1,i})}{\widehat{b}_1(A_{1,i}|H_{1,i})}\right)\cdot\widehat{\mathbb{E}}[\mathbb{I}\{R_{1,i}+\widehat{R}_2^*(\pi)<\eta\}-\tau|H_{2,i}]\right]\\
=&\mathbb{E}\left[\left(\frac{\pi_1(A_{1,i}|H_{1,i})}{\widehat{b}_1(A_{1,i}|H_{1,i})}\right)\left(\frac{\pi_2(A_{2,i}|H_{2,i})}{b_2(A_{2,i}|H_{2,i})}\right)\cdot\widehat{\mathbb{E}}[\mathbb{I}\{R_{1,i}+\widehat{R}_2^*(\pi)<\eta\}-\tau|H_{2,i}]\right]\\
=&\mathbb{E}\left[\left(\frac{\pi_1(A_{1,i}|H_{1,i})}{\widehat{b}_1(A_{1,i}|H_{1,i})}\right)\left(\frac{\pi_2(A_{2,i}|H_{2,i})}{b_2(A_{2,i}|H_{2,i})}\right)\cdot\left(\widehat{\mathbb{E}}[\mathbb{I}\{R_{1,i}+\widehat{R}_2^*(\pi)<\eta\}|H_{2,i}]-\widehat{\mathbb{E}}[\mathbb{I}\{R_{1,i}+R_2^*(\pi)<\eta\}|H_{2,i}]\right)\right.\\
&\qquad\qquad\left.+\left(\frac{\pi_1(A_{1,i}|H_{1,i})}{\widehat{b}_1(A_{1,i}|H_{1,i})}\right)\widehat{\mathbb{E}}[\mathbb{I}\{R_{1,i}+R_2^*(\pi)<\eta\}-\tau|H_{2,i}]\right],
\end{aligned}
$$

where we add and subtract a term in the last equality to maintain the equivalence of the formula.

Plug in $L2$ to Formula (38), we have

$$
\begin{aligned}
&\mathbb{E}\left[\psi(W;\eta,\hat{\alpha})\right]\\
=&\mathbb{E}\Bigg[\left(\frac{\pi_1(A_{1,i}|H_{1,i})}{\widehat{b}_1(A_{1,i}|H_{1,i})}\right)\left(\frac{\pi_2(A_{2,i}|H_{2,i})}{\widehat{b}_2(A_{2,i}|H_{2,i})}\right)\cdot\left(\mathbb{E}[\mathbb{I}\{R_{1,i}+R_2^*(\pi)<\eta\}|H_{2,i}]-\widehat{\mathbb{E}}[\mathbb{I}\{R_{1,i}+\widehat{R}_2^*(\pi)<\eta\}|H_{2,i}]\right)\\
&+\left(\frac{\pi_1(A_{1,i}|H_{1,i})}{\widehat{b}_1(A_{1,i}|H_{1,i})}\right)\left(\frac{\pi_2(A_{2,i}|H_{2,i})}{b_2(A_{2,i}|H_{2,i})}\right)\cdot\left(\widehat{\mathbb{E}}[\mathbb{I}\{R_{1,i}+\widehat{R}_2^*(\pi)<\eta\}|H_{2,i}]-\widehat{\mathbb{E}}[\mathbb{I}\{R_{1,i}+R_2^*(\pi)<\eta\}|H_{2,i}]\right)\\
&+\left(\frac{\pi_1(A_{1,i}|H_{1,i})}{\widehat{b}_1(A_{1,i}|H_{1,i})}\right)\widehat{\mathbb{E}}[\mathbb{I}\{R_{1,i}+R_2^*(\pi)<\eta\}-\tau|H_{2,i}]\\
&+\left(1-\frac{\pi_1(A_{1,i}|H_{1,i})}{\widehat{b}_1(A_{1,i}|H_{1,i})}\right)\widehat{\mathbb{E}}[\mathbb{I}\{\widehat{R}_1^*(\pi)+\widehat{R}_2^*(\pi)<\eta\}-\tau|H_{1,i}]\Bigg]\\
=&\mathbb{E}\Bigg[\left(\frac{\pi_1(A_{1,i}|H_{1,i})}{\widehat{b}_1(A_{1,i}|H_{1,i})}\right)\left(\frac{\pi_2(A_{2,i}|H_{2,i})}{\widehat{b}_2(A_{2,i}|H_{2,i})}-\frac{\pi_2(A_{2,i}|H_{2,i})}{b_2(A_{2,i}|H_{2,i})}\right)\\
&\quad\cdot\left(\mathbb{E}[\mathbb{I}\{R_{1,i}+R_2^*(\pi)<\eta\}|H_{2,i}]-\widehat{\mathbb{E}}[\mathbb{I}\{R_{1,i}+\widehat{R}_2^*(\pi)<\eta\}|H_{2,i}]\right)\\
&+\left(\frac{\pi_1(A_{1,i}|H_{1,i})}{\widehat{b}_1(A_{1,i}|H_{1,i})}\right)\widehat{\mathbb{E}}[\mathbb{I}\{R_{1,i}+R_2^*(\pi)<\eta\}-\tau|H_{2,i}]+\left(1-\frac{\pi_1(A_{1,i}|H_{1,i})}{\widehat{b}_1(A_{1,i}|H_{1,i})}\right)\widehat{\mathbb{E}}[\mathbb{I}\{\widehat{R}_1^*(\pi)+\widehat{R}_2^*(\pi)<\eta\}-\tau|H_{1,i}]\Bigg]
\end{aligned}
$$

Also, similar to the logic in Lemma 1, it's easy to show that the augmentation terms in $\psi^*(W;\eta)$ are 0. Therefore,

$$
\begin{aligned}
\mathbb{E}\left[\psi^*(W;\eta)\right]&=\mathbb{E}\left[\frac{\pi_1(A_{1,i}|H_{1,i})\pi_2(A_{2,i}|H_{2,i})}{b_1(A_{1,i}|H_{1,i})b_2(A_{2,i}|H_{2,i})}\cdot\left(\mathbb{I}\{(R_{1,i}+R_{2,i}<\eta\}-\tau\right)\right]\\
&=\mathbb{E}\left[\mathbb{I}\{(R_1^*(\pi)+R_2^*(\pi)<\eta\}-\tau\right].
\end{aligned}
$$

Then we have

$$
\begin{aligned}
&\mathbb{E}\left[\psi(W;\eta,\hat{\alpha})\right] - \mathbb{E}\left[\psi^*(W;\eta)\right] = \mathbb{E}\left[\psi(W;\eta,\hat{\alpha})\right] - \mathbb{E}\left[\mathbb{I}\{(R_1^*(\pi)+R_2^*(\pi)<\eta\} - \tau\right] \\
=&\mathbb{E}\left[\left(\frac{\pi_1(A_{1,i}|H_{1,i})}{\widehat{b}_1(A_{1,i}|H_{1,i})}\right)\left(\frac{\pi_2(A_{2,i}|H_{2,i})}{\widehat{b}_2(A_{2,i}|H_{2,i})} - \frac{\pi_2(A_{2,i}|H_{2,i})}{b_2(A_{2,i}|H_{2,i})}\right)\right. \\
&\qquad \cdot \left(\mathbb{E}[\mathbb{I}\{R_{1,i}+R_2^*(\pi)<\eta\}\big|H_{2,i}] - \widehat{\mathbb{E}}[\mathbb{I}\{R_{1,i}+\widehat{R}_2^*(\pi)<\eta\}\big|H_{2,i}]\right) \\
&\quad + \left(\frac{\pi_1(A_{1,i}|H_{1,i})}{\widehat{b}_1(A_{1,i}|H_{1,i})}\right)\widehat{\mathbb{E}}[\mathbb{I}\{R_{1,i}+R_2^*(\pi)<\eta\} - \tau\big|H_{2,i}] \\
&\quad + \left.\left(1 - \frac{\pi_1(A_{1,i}|H_{1,i})}{\widehat{b}_1(A_{1,i}|H_{1,i})}\right)\widehat{\mathbb{E}}[\mathbb{I}\{\widehat{R}_1^*(\pi)+\widehat{R}_2^*(\pi)<\eta\} - \tau\big|H_{1,i}] - \mathbb{E}\left[\mathbb{I}\{(R_1^*(\pi)+R_2^*(\pi)<\eta\} - \tau\big|H_{1,i}\right]\right].
\end{aligned}
\tag{39}
$$

Let's then consider the last two lines in formula (39).

$$
\begin{aligned}
&\mathbb{E}\left[\left(\frac{\pi_1(A_{1,i}|H_{1,i})}{\widehat{b}_1(A_{1,i}|H_{1,i})}\right)\widehat{\mathbb{E}}[\mathbb{I}\{R_{1,i}+R_2^*(\pi)<\eta\} - \tau\big|H_{2,i}]\right. \\
&\quad + \left.\left(1 - \frac{\pi_1(A_{1,i}|H_{1,i})}{\widehat{b}_1(A_{1,i}|H_{1,i})}\right)\widehat{\mathbb{E}}[\mathbb{I}\{\widehat{R}_1^*(\pi)+\widehat{R}_2^*(\pi)<\eta\} - \tau\big|H_{1,i}] - \mathbb{E}\left[\mathbb{I}\{(R_1^*(\pi)+R_2^*(\pi)<\eta\} - \tau|H_{1,i}\right]\right] \\
=&\mathbb{E}\left[\left(\frac{\pi_1(A_{1,i}|H_{1,i})}{\widehat{b}_1(A_{1,i}|H_{1,i})}\right)\left(\widehat{\mathbb{E}}[\mathbb{I}\{(R_1^*(\pi)+R_2^*(\pi)<\eta\}|H_{1,i}] - \widehat{\mathbb{E}}[\mathbb{I}\{\widehat{R}_1^*(\pi)+\widehat{R}_2^*(\pi)<\eta\}|H_{1,i}]\right)\right. \\
&\quad + \left.\widehat{\mathbb{E}}[\mathbb{I}\{\widehat{R}_1^*(\pi)+\widehat{R}_2^*(\pi)<\eta\}|H_{1,i}] - \widehat{\mathbb{E}}[\mathbb{I}\{R_1^*(\pi)+R_2^*(\pi)<\eta\}|H_{1,i}]\right] \\[6pt]
=&\mathbb{E}\left[\left(\frac{\pi_1(A_{1,i}|H_{1,i})}{\widehat{b}_1(A_{1,i}|H_{1,i})}\right)\left(\widehat{\mathbb{E}}[\mathbb{I}\{R_1^*(\pi)+R_2^*(\pi)<\eta\}|H_{1,i}] - \widehat{\mathbb{E}}[\mathbb{I}\{\widehat{R}_1^*(\pi)+\widehat{R}_2^*(\pi)<\eta\}|H_{1,i}]\right)\right. \\
&\quad - \left.\left(\frac{\pi_1(A_{1,i}|H_{1,i})}{b_1(A_{1,i}|H_{1,i})}\right)\left(\widehat{\mathbb{E}}[\mathbb{I}\{R_1^*(\pi)+R_2^*(\pi)<\eta\}|H_{1,i}] - \widehat{\mathbb{E}}[\mathbb{I}\{\widehat{R}_1^*(\pi)+\widehat{R}_2^*(\pi)<\eta\}|H_{1,i}]\right)\right] \\
=&\mathbb{E}\left[\left(\frac{\pi_1(A_{1,i}|H_{1,i})}{\widehat{b}_1(A_{1,i}|H_{1,i})} - \frac{\pi_1(A_{1,i}|H_{1,i})}{b_1(A_{1,i}|H_{1,i})}\right)\left(\widehat{\mathbb{E}}[\mathbb{I}\{R_1^*(\pi)+R_2^*(\pi)<\eta\}|H_{1,i}] - \widehat{\mathbb{E}}[\mathbb{I}\{\widehat{R}_1^*(\pi)+\widehat{R}_2^*(\pi)<\eta\}|H_{1,i}]\right)\right].
\end{aligned}
\tag{40}
$$

Combining the result of Formula (39) and Formula (40), we have

$$
\begin{aligned}
&\left|\mathbb{E}\left[\psi(W;\eta,\hat{\alpha})\right] - \mathbb{E}\left[\psi^*(W;\eta)\right]\right| \\
=&\left|\mathbb{E}\left[\left(\frac{\pi_1(A_{1,i}|H_{1,i})}{\widehat{b}_1(A_{1,i}|H_{1,i})}\right)\left(\frac{\pi_2(A_{2,i}|H_{2,i})}{\widehat{b}_2(A_{2,i}|H_{2,i})} - \frac{\pi_2(A_{2,i}|H_{2,i})}{b_2(A_{2,i}|H_{2,i})}\right)\right.\right. \\
&\qquad \cdot \left(\mathbb{E}[\mathbb{I}\{R_{1,i}+R_2^*(\pi)<\eta\}\big|H_{2,i}] - \widehat{\mathbb{E}}[\mathbb{I}\{R_{1,i}+\widehat{R}_2^*(\pi)<\eta\}\big|H_{2,i}]\right)\Big] \\
&\quad + \left.\mathbb{E}\left[\left(\frac{\pi_1(A_{1,i}|H_{1,i})}{\widehat{b}_1(A_{1,i}|H_{1,i})} - \frac{\pi_1(A_{1,i}|H_{1,i})}{b_1(A_{1,i}|H_{1,i})}\right)\cdot\left(\widehat{\mathbb{E}}[\mathbb{I}\{R_1^*(\pi)+R_2^*(\pi)<\eta\}\big|H_{1,i}] - \widehat{\mathbb{E}}[\mathbb{I}\{\widehat{R}_1^*(\pi)+\widehat{R}_2^*(\pi)<\eta\}\big|H_{1,i}]\right)\right]\right| \\
\leq&\frac{1}{\epsilon^3}\cdot\left\|\widehat{b}_2(A_{2,i}|H_{2,i}) - b_2(A_{2,i}|H_{2,i})\right\|_{P,2}\cdot\left\|\delta\left(\widehat{F}_{R_2^*(\pi)|H_2}, F_{R_2^*(\pi)|H_2}\right)\right\|_{P,2} \\
&\quad + \frac{1}{\epsilon^2}\cdot\left\|\widehat{b}_1(A_{1,i}|H_{1,i}) - b_1(A_{1,i}|H_{1,i})\right\|_{P,2}\cdot\left\|\delta\left(\widehat{F}_{R^*(\pi)|H_1}, F_{R^*(\pi)|H_1}\right)\right\|_{P,2}
\end{aligned}
\tag{41}
$$

where the last inequality holds by condition (C3$'$) and Assumption (A4).

Therefore, as long as the propensity score models are consistently estimated, or the outcome regression models for $\widehat{R}_1$ and $\widehat{R}_2$ obtained by MDN are consistently estimated, we have

$$|\mathbb{E}\left[\psi(W; \eta, \hat{\alpha})\right] - \mathbb{E}\left[\psi^*(W; \eta)\right]| = o(1),$$

according to formula (41).

To be more specific, we only need one of the following conditions to be satisfied:

1. $\left\|\widehat{b}_1(A_1|H_1) - b_1(A_1|H_1)\right\|_{P,2} = o(1)$, and $\left\|\widehat{b}_2(A_2|H_2) - b_2(A_2|H_2)\right\|_{P,2} = o(1)$;

2. $\left\|\delta\left(\widehat{F}_{R^*(\pi)|H_1}, F_{R^*(\pi)|H_1}\right)\right\|_{P,2} = o(1)$, and $\left\|\delta\left(\widehat{F}_{R_2^*(\pi)|H_2}, F_{R_2^*(\pi)|H_2}\right)\right\|_{P,2} = o(1)$.

The proof of Lemma 2 is thus complete.

Notice that Assumption (A1) is not necessary to prove the double robustness. This assumption is imposed to further prove the asymptotic normality of our quantile estimator in Appendix C.4. □

### C.3  Proof of Lemma 2: Double Robustness of our Variance Estimator

*Proof.* Firstly, we show that when $\lim_{N\to\infty} h_N = 0$, and $\lim_{N\to\infty} Nh_N = \infty$, the kernel density estimator for $f(r) := f_{R^*(\pi)}(r)$ attains weak consistency.

Let's first define $f_N(r)$ as the KDE of $f(r)$. By Chebyshev's Inequality, at each point of continuity $r$ of $f$ and for any $\epsilon > 0$,

$$\mathbb{P}(|f_N(r) - f(r)| > \epsilon) \leq \frac{\mathbb{E}[(f_N(r) - f(r))^2]}{\epsilon^2} = \frac{\text{Var}[f_N(r)] + \text{Bias}^2[f_N(r)]}{\epsilon^2}.$$

According to the result in Parzen (1962), $lim_{N\to\infty}\text{Bias}^2[f_N(r)] = 0$ when $\lim_{N\to\infty} h_N = 0$. That is, $f_N(r)$ is asymptotically unbiased when $\lim_{N\to\infty} h_N = 0$.

According to the mild assumptions on $K$ stated in Lemma 2, the asymptotic variance of the density estimator is given by

$$lim_{N\to\infty} Nh_N \text{Var}[f_N(r)] = f(r) \int_{-\infty}^{\infty} K^2(u)dy < \infty.$$

Therefore, both the asymptotic bias and variance go to 0 as $N \to \infty$. This illustrates the asymptotic consistency of the kernel density estimator in $\widehat{\sigma}^2_{\text{DR}}$.

The rest of the proof is trivial. One can follow the idea in Lemma 2 to similarly prove the double robustness of our variance estimator. For the brevity of content, we omit the details for this part. □

### C.4  Proof of Theorem 1: Asymptotic Normality of our quantile estimator

*Proof.* same as the proof in Lemma 1-3, we will only illustrate the case when $K = 2$ for the brevity of content. We will give a brief sketch in Step 1, and leave the rest of the details in Step 2-5 accordingly.

Based on our estimating procedure in theory section, we have

$$\sum_{s=1}^{S} \mathbb{E}_{n,s}[\psi(W_s; \widehat{\eta}_\tau^{\text{DR}}, \hat{\alpha}_s)] = \frac{1}{n} \sum_{s=1}^{S} \sum_{i\in\mathcal{I}_s} \psi(W_{i,s}; \widehat{\eta}_\tau^{\text{DR}}, \hat{\alpha}_s) = 0;$$

$$\mathbb{E}[\psi^*(W_s; \eta_\tau)] = \mathbb{E}[a + b + c] = 0 \quad \text{for any } s \in \{1, \ldots, S\}.$$

$$(42)$$

**Step 1**:

First, we apply Taylor series expansion on $\mathbb{E}[\psi^*(W;\eta)]$ around $\eta_\tau$.

$$
\begin{aligned}
\mathbb{E}[\psi^*(W;\eta)] &= \mathbb{E}[\psi^*(W;\eta_\tau)] + J_0(\eta - \eta_\tau) + \frac{1}{2}H_0(\eta - \eta_\tau)^2 + o(\|\eta - \eta_\tau\|^2) \\
&= J_0(\eta - \eta_\tau) + \frac{1}{2}H_0(\eta - \eta_\tau)^2 + o(\|\eta - \eta_\tau\|^2),
\end{aligned}
\tag{43}
$$

where $J_0 = \partial_\eta\{\mathbb{E}_W[\psi^*(W;\eta)]\}|_{\eta=\eta_\tau}$ and $H_0 = \partial_\eta^2\{\mathbb{E}_W[\psi^*(W;\eta)]\}|_{\eta=\eta_\tau}$.

Suppose that $\mathbb{E}[\sum_{s=1}^{S}\psi(W_s;\widehat{\eta}_\tau^{\mathrm{DR}},\widehat{\alpha}_s)] = \mathbb{E}[\sum_{s=1}^{S}\psi^*(W_s;\widehat{\eta}_\tau^{\mathrm{DR}})] + o(n^{-1/2})$ (We will prove it in Step 4). Since $\mathbb{E}[\psi^*(W_s;\eta_\tau)] = 0$, by plugging in the definition of empirical operator, we got

$$
\begin{aligned}
&\sqrt{n}\sum_{s=1}^{S}\mathbb{E}_{n,s}[\psi(W_s;\widehat{\eta}_\tau^{\mathrm{DR}},\widehat{\alpha}_s) - \psi^*(W_s;\eta_\tau)] \\
&= \sum_{s=1}^{S}\mathbb{G}_{n,s}[\psi(W_s;\widehat{\eta}_\tau^{\mathrm{DR}},\widehat{\alpha}_s) - \psi^*(W_s;\eta_\tau)] + \sqrt{n}\sum_{s=1}^{S}\mathbb{E}[\psi(W_s;\widehat{\eta}_\tau^{\mathrm{DR}},\widehat{\alpha}_s)] \\
&= \sum_{s=1}^{S}\mathbb{G}_{n,s}[\psi(W_s;\widehat{\eta}_\tau^{\mathrm{DR}},\widehat{\alpha}_s) - \psi^*(W_s;\eta_\tau)] + \sqrt{n}\sum_{s=1}^{S}\mathbb{E}[\psi^*(W_s;\widehat{\eta}_\tau^{\mathrm{DR}})] + o_p(1).
\end{aligned}
\tag{44}
$$

Let $\eta = \widehat{\eta}_\tau^{\mathrm{DR}}$ in formula (43) and combine it with formula (44). Then

$$
\begin{aligned}
\sqrt{n}\sum_{s=1}^{S}J_0(\widehat{\eta}_\tau^{\mathrm{DR}} - \eta_\tau) + \sqrt{n}\sum_{s=1}^{S}\mathbb{E}_{n,s}[\psi^*(W_s;\eta_\tau)] &= -\sum_{s=1}^{S}\mathbb{G}_{n,s}[\psi(W_s;\widehat{\eta}_\tau^{\mathrm{DR}},\widehat{\alpha}_s) - \psi^*(W_s;\eta_\tau)] \\
&\quad - \frac{1}{2}\sqrt{n}\sum_{s=1}^{S}H_0(\widehat{\eta}_\tau^{\mathrm{DR}} - \eta_\tau)^2 - \sqrt{n}\cdot o(\|\widehat{\eta}_\tau^{\mathrm{DR}} - \eta_\tau\|^2) + o_p(1).
\end{aligned}
\tag{45}
$$

Therefore,

$$
\begin{aligned}
&\sqrt{S}\cdot\sqrt{N}\left\|J_0(\widehat{\eta}_\tau^{\mathrm{DR}} - \eta_\tau) + \frac{1}{S}\sum_{s=1}^{S}\mathbb{E}_{n,s}[\psi^*(W_s;\eta_\tau)]\right\| \\
&= \sqrt{n}\left\|\sum_{s=1}^{S}J_0(\widehat{\eta}_\tau^{\mathrm{DR}} - \eta_\tau) + \sum_{s=1}^{S}\mathbb{E}_{n,s}[\psi^*(W_s;\eta_\tau)]\right\| \leq \left\|\sum_{s=1}^{S}\mathbb{G}_{n,s}[\psi(W_s;\widehat{\eta}_\tau^{\mathrm{DR}},\widehat{\alpha}_s) - \psi^*(W_s;\eta_\tau)]\right\| \\
&\quad + \frac{1}{2}\sqrt{n}\sum_{s=1}^{S}H_0(\widehat{\eta}_\tau^{\mathrm{DR}} - \eta_\tau)^2 + \sqrt{n}\cdot o(\|\widehat{\eta}_\tau^{\mathrm{DR}} - \eta_\tau\|^2) + o_p(1).
\end{aligned}
$$

To finish the proof of this theorem, it suffices to show that

$$
\sqrt{N}\left\|J_0(\widehat{\eta}_\tau^{\mathrm{DR}} - \eta_\tau) + \frac{1}{S}\sum_{s=1}^{S}\mathbb{E}_{n,s}[\psi^*(W_s;\eta_\tau)]\right\| = o_p(1),
\tag{46}
$$

which is satisfied if we can prove that all terms on the RHS of the inequality in formula (46) are $o_p(1)$. That is, we only need to show

1. $\left\|\sum_{s=1}^{S}\mathbb{G}_{n,s}[\psi(W_s;\widehat{\eta}_\tau^{\mathrm{DR}},\widehat{\alpha}_s) - \psi^*(W_s;\eta_\tau)]\right\| = o_p(1)$, which follows directly from triangle inequality if we can prove for any $s \in \{1,\ldots,S\}$,

$$
\left\|\mathbb{G}_{n,s}[\psi(W_s;\widehat{\eta}_\tau^{\mathrm{DR}},\widehat{\alpha}_s) - \psi^*(W_s;\eta_\tau)]\right\| = o_p(1).
$$

2. $\left\|\widehat{\eta}_\tau^{\mathrm{DR}} - \eta_\tau\right\| = O_p(n^{-1/2})$.

3. $\mathbb{E}[\sum_{s=1}^{S} \psi(W_s; \widehat{\eta}_\tau^{\mathrm{DR}}, \hat{\alpha}_s)] = \mathbb{E}[\sum_{s=1}^{S} \psi^*(W_s; \widehat{\eta}_\tau^{\mathrm{DR}})] + o(n^{-1/2})$, which follows naturally if we can prove for any $s \in \{1, \ldots, S\}$,

$$\mathbb{E}[\psi(W_s; \widehat{\eta}_\tau^{\mathrm{DR}}, \hat{\alpha}_s)] = \mathbb{E}[\psi^*(W_s; \widehat{\eta}_\tau^{\mathrm{DR}})] + o(n^{-1/2}).$$

4. $H_0$ is bounded.

Detailed proofs of these four conditions are provided in Step 2-5. Assuming condition 1-4 hold, we have

$$\sqrt{N} \left\| J_0(\widehat{\eta}_\tau^{\mathrm{DR}} - \eta_\tau) + \frac{1}{S} \sum_{s=1}^{S} \mathbb{E}_{n,s}[\psi^*(W_s; \eta_\tau)] \right\| = o_p(1), \tag{47}$$

then

$$\sqrt{N}(\widehat{\eta}_\tau^{\mathrm{DR}} - \eta_\tau) = -J_0^{-1} \frac{1}{S} \sum_{s=1}^{S} \mathbb{E}_{n,s}[\psi^*(W_s; \eta_\tau)] + o_p(1) = -J_0^{-1} \frac{1}{\sqrt{N}} \sum_{i=1}^{N} \psi^*(W_i; \eta_\tau) + o_p(1).$$

The claim of this theorem thus follows.

Next, we will prove that all of these four conditions are satisfied in Step 2, Step 3, Step 4 and Step 5 respectively.

In the following proof, when the result holds for any fold $s \in \{1, \ldots, S\}$, the subscript $s$ in $W_s$, $\hat{\alpha}_s$, $\mathbb{G}_{n,s}$ and $\mathbb{E}_{n,s}$ will be omitted to avoid the redundancy of notations.

**Step 2**: Prove that $\left\| \mathbb{G}_n[\psi(W; \widehat{\eta}_\tau^{\mathrm{DR}}, \hat{\alpha}) - \psi^*(W; \eta_\tau)] \right\| = o_p(1)$.

Define $\mathcal{N}_{\tau_n}(\eta_\tau) = \{\eta : \|\eta - \eta_\tau\| \leq \tau_n\}$ as a neighborhood of $\eta_\tau$ with size $\tau_n$, and define $\mathcal{F} = \{\psi(W; \eta, \hat{\alpha}) - \psi(W; \eta_\tau, \hat{\alpha}) : \eta \in \mathcal{N}_{\tau_n}(\eta_\tau)\}$, where $\tau_n$ is a sequence of positive constants converging to zero. (According to Step 3, $\tau_n$ can be any sequence with order $\tau_n \geq O(n^{-1/2})$).

Since

$$\left\| \mathbb{G}_n[\psi(W; \widehat{\eta}_\tau^{\mathrm{DR}}, \hat{\alpha}) - \psi^*(W; \eta_\tau)] \right\| \leq \sup_{\eta \in \mathcal{N}_{\tau_n}(\eta_\tau)} \left\| \mathbb{G}_n[\psi(W; \eta, \hat{\alpha}) - \psi(W; \eta_\tau, \hat{\alpha})] \right\| + \left\| \mathbb{G}_n[\psi(W; \eta_\tau, \hat{\alpha}) - \psi^*(W; \eta_\tau)] \right\|, \tag{48}$$

It suffices to show both terms on the RHS of Formula (48) is $o_p(1)$.

**Step 2.1**: Prove that $\sup_{\eta \in \mathcal{N}_{\tau_n}(\eta_\tau)} \left\| \mathbb{G}_n[\psi(W; \eta, \hat{\alpha}) - \psi(W; \eta_\tau, \hat{\alpha})] \right\| = o_p(1)$.

First, it can be shown that $\mathcal{F}_0^{\hat{\alpha}} = \{\psi(W; \eta, \hat{\alpha}) : \eta \in \mathcal{N}_{\tau_n}(\eta_\tau)\}$ is a VC Class with VC-index $V(\mathcal{F}_0^{\hat{\alpha}})$. Also, for some $q > 2$, the entropy is bounded by some constant $C_2$, i.e. $\left\| F_0^{\hat{\alpha}} \right\|_{P,q} \leq C_2$. According to Lemma 2.6.18 in Van Der Vaart et al. (1996), $\mathcal{F}$ is also a VC Class. By Conditioning on $(W_i)_{i \in \mathcal{I}^c}$, $\hat{\alpha}$ can be regarded as a fixed value. By applying the Maximal Inequality specialized to VC type classes (Chernozhukov et al., 2014), there exists a sufficiently large constant $C$, such that with probability $1 - o(1)$,

$$\sup_{f \in \mathcal{F}} |\mathbb{G}_n(f)| \leq C \cdot \left( r_N \log^{1/2}(1/r_N) + n^{-1/2+1/q} \log n \right), \tag{49}$$

where

$$r_N = \sup_{\|\eta - \eta_\tau\| \leq \tau_N} \|\psi(W; \eta, \hat{\alpha}) - \psi(W; \eta_\tau, \hat{\alpha})\|_{P,2}. \tag{50}$$

If we can show that $r_N = o(1)$, then $\sup_{f \in \mathcal{F}} |\mathbb{G}_n(f)| = o(1)$, and the claim of Step 2.1 follows.

To prove this, we split $\psi$ into three parts and show that each part is indeed $o(1)$. Let $\psi(W; \eta, \hat{\alpha}) = a(\eta, \hat{\alpha}) + b(\eta, \hat{\alpha}) + c(\eta, \hat{\alpha})$, where

$$
\begin{aligned}
a(\eta, \hat{\alpha}) &= \frac{\pi_1(A_1|H_1)\pi_2(A_2|H_2)}{\widehat{b}_1(A_1|H_1)\widehat{b}_2(A_2|H_2)}(\mathbb{I}\{R_1 + R_2 < \eta\} - \tau), \\
b(\eta, \hat{\alpha}) &= a_1(\hat{\alpha}) \cdot \widehat{\mathbb{E}}[(\mathbb{I}\{\widehat{R}_1 + \widehat{R}_2 < \eta\} - \tau)|X_1, (A_1, A_2) \sim \pi], \\
c(\eta, \hat{\alpha}) &= a_2(\hat{\alpha}) \cdot \widehat{\mathbb{E}}[(\mathbb{I}\{R_1 + \widehat{R}_2 < \eta\} - \tau)|H_2, A_2 \sim \pi],
\end{aligned}
\tag{51}
$$

and $a_1(\hat{\alpha})$, $a_2(\hat{\alpha})$ are defined as

$$a_1(\hat{\alpha}) = \left(1 - \frac{\pi_1(A_{1,i}|H_{1,i})}{\widehat{b}_1(A_{1,i}|H_{1,i})}\right), \quad a_2(\hat{\alpha}) = \left(\frac{\pi_1(A_{1,i}|H_{1,i})}{\widehat{b}_1(A_{1,i}|H_{1,i})}\right)\left(1 - \frac{\pi_2(A_{2,i}|H_{2,i})}{\widehat{b}_2(A_{2,i}|H_{2,i})}\right). \tag{52}$$

Specifically, when the estimating equation is $\psi^*(W;\eta)$, we let $\psi^*(W;\eta) := a^*(\eta) + b^*(\eta) + c^*(\eta)$.

**Step 2.1.a**: Prove that $\sup_{\eta \in \mathcal{N}_{\tau_n}(\eta_\tau)} \|a(\eta,\hat{\alpha}) - a(\eta_\tau,\hat{\alpha})\|_{P,2} = o(1)$.

$$\sup_{\eta \in \mathcal{N}_{\tau_n}(\eta_\tau)} \|a(\eta,\hat{\alpha}) - a(\eta_\tau,\hat{\alpha})\|_{P,2}$$

$$= \sup_{\eta \in \mathcal{N}_{\tau_n}(\eta_\tau)} \left\| \frac{\pi_1(A_1|H_1)\pi_2(A_2|H_2)}{\widehat{b}_1(A_1|H_1)\widehat{b}_2(A_2|H_2)}\left(\mathbb{I}\{R_1 + R_2 < \eta\} - \mathbb{I}\{R_1 + R_2 < \eta_\tau\}\right) \right\|_{P,2}$$

$$= \sup_{\eta \in \mathcal{N}_{\tau_n}(\eta_\tau)} \left\| \frac{\pi_1(A_1|H_1)\pi_2(A_2|H_2)}{\widehat{b}_1(A_1|H_1)\widehat{b}_2(A_2|H_2)}\mathbb{I}\{R_1 + R_2 \in [\min(\eta_\tau,\eta), \max(\eta_\tau,\eta))\} \right\|_{P,2}$$

$$\leq \left\| \frac{\pi_1(A_1|H_1)\pi_2(A_2|H_2)}{\widehat{b}_1(A_1|H_1)\widehat{b}_2(A_2|H_2)} \right\|_{P,2} \cdot \sup_{\eta \in \mathcal{N}_{\tau_n}(\eta_\tau)} \left( \mathbb{E}\left|\mathbb{I}\{R^*(\pi) \in [\min(\eta_\tau,\eta), \max(\eta_\tau,\eta)]\}\right|^2 \right)^{\frac{1}{2}} \tag{53}$$

$$= \left\| \frac{\pi_1(A_1|H_1)\pi_2(A_2|H_2)}{\widehat{b}_1(A_1|H_1)\widehat{b}_2(A_2|H_2)} \right\|_{P,2} \cdot \sup_{\eta \in \mathcal{N}_{\tau_n}(\eta_\tau)} \left( \mathbb{E}\left[\mathbb{I}\{R^*(\pi) \in [\min(\eta_\tau,\eta), \max(\eta_\tau,\eta)]\}\right] \right)^{\frac{1}{2}}$$

$$\leq \frac{1}{\epsilon^2} \cdot \sup_{\eta \in \mathcal{N}_{\tau_n}(\eta_\tau)} \left|F_{R^*(\pi)}(\eta) - F_{R^*(\pi)}(\eta_\tau)\right|^{\frac{1}{2}}$$

$$= \frac{1}{\epsilon^2} \cdot \sup_{\eta \in \mathcal{N}_{\tau_n}(\eta_\tau)} \left|\mathbb{E}_{H_1 \sim \mathbb{G}}\left[F_{R^*(\pi)|H_1}(\eta|H_1) - F_{R^*(\pi)|H_1}(\eta_\tau|H_1)\right]\right|^{\frac{1}{2}},$$

where $F_{R^*(\pi)}(\eta)$ is the marginal cumulative density function of random variable $R^*(\pi) = R_1^*(\pi) + R_2^*(\pi)$. Notice that the first inequality in (53) holds because of Cauchy-Schwarz inequality, and the second inequality holds by Assumption (A4).

According to Assumption (A2), there exists a constant $C_3$, such that for all $r \in \mathbb{R}$ and $H_1$, $\left|f_{R^*(\pi)|H_1}(r|H_1)\right| \leq C_3$. Thus,

$$\left|F_{R^*(\pi)|H_1}(\eta|H_1) - F_{R^*(\pi)|H_1}(\eta_\tau|H_1)\right| \leq C_3 \cdot \|\eta - \eta_\tau\|. \tag{54}$$

Then we have

$$\sup_{\eta \in \mathcal{N}_{\tau_n}(\eta_\tau)} \|a(\eta,\hat{\alpha}) - a(\eta_\tau,\hat{\alpha})\|_{P,2}$$

$$\leq \frac{1}{\epsilon^2} \cdot \sup_{\eta \in \mathcal{N}_{\tau_n}(\eta_\tau)} \left|\mathbb{E}_{H_1 \sim \mathbb{G}}\left[F_{R^*(\pi)|H_1}(\eta|H_1) - F_{R^*(\pi)|H_1}(\eta_\tau|H_1)\right]\right|^{\frac{1}{2}} \tag{55}$$

$$\leq \frac{1}{\epsilon^2} \cdot \sup_{\eta \in \mathcal{N}_{\tau_n}(\eta_\tau)} (C_3 \cdot \|\eta - \eta_\tau\|)^{\frac{1}{2}} = \frac{1}{\epsilon^2} \cdot o(1) = o(1),$$

which finishes the proof of this Step 2.1.a.

**Step 2.1.b**: Prove that $\sup_{\eta\in\mathcal{N}_{\tau_n}(\eta_\tau)} \|b(\eta,\hat\alpha) - b(\eta_\tau,\hat\alpha)\|_{P,2} = o(1)$.

$$
\sup_{\eta\in\mathcal{N}_{\tau_n}(\eta_\tau)} \|b(\eta,\hat\alpha) - b(\eta_\tau,\hat\alpha)\|_{P,2}
$$

$$
= \sup_{\eta\in\mathcal{N}_{\tau_n}(\eta_\tau)} \left\| \left(1 - \frac{\pi_1(A_{1,i}|H_{1,i})}{\hat b_1(A_{1,i}|H_{1,i})}\right) \widehat{\mathbb{E}}[(\mathbb{I}\{\hat R_1^*(\pi) + \hat R_2^*(\pi) < \eta\} - \mathbb{I}\{\hat R_1^*(\pi) + \hat R_2^*(\pi) < \eta_\tau\})|H_1] \right\|_{P,2}
$$

$$
= \sup_{\eta\in\mathcal{N}_{\tau_n}(\eta_\tau)} \left\| \left(1 - \frac{\pi_1(A_{1,i}|H_{1,i})}{\hat b_1(A_{1,i}|H_{1,i})}\right) \cdot \left(\widehat F_{R^*(\pi)|H_1}(\eta;\hat\alpha|H_1) - \widehat F_{R^*(\pi)|H_1}(\eta_\tau;\hat\alpha|H_1)\right) \right\|_{P,2} \tag{56}
$$

$$
\leq \left\| 1 - \frac{\pi_1(A_{1,i}|H_{1,i})}{\hat b_1(A_{1,i}|H_{1,i})} \right\|_{P,2} \cdot \sup_{\eta\in\mathcal{N}_{\tau_n}(\eta_\tau)} \left\| \widehat F_{R^*(\pi)|H_1}(\eta;\hat\alpha|H_1) - \widehat F_{R^*(\pi)|H_1}(\eta_\tau;\hat\alpha|H_1) \right\|_{P,2}
$$

$$
\leq \frac{2}{\epsilon} \cdot \sup_{\eta\in\mathcal{N}_{\tau_n}(\eta_\tau)} \left\| \widehat F_{R^*(\pi)|H_1}(\eta;\hat\alpha|H_1) - \widehat F_{R^*(\pi)|H_1}(\eta_\tau;\hat\alpha|H_1) \right\|_{P,2},
$$

where $\widehat F_{R^*(\pi)}(r;\hat\alpha|H_1)$ is the estimated cdf of $(\hat R_1 + \hat R_2)$ given baseline information $H_1$ and policy $\pi$. Also, we have

$$
\left\| \widehat F_{R^*(\pi)|H_1}(\eta;\hat\alpha|H_1) - \widehat F_{R^*(\pi)|H_1}(\eta_\tau;\hat\alpha|H_1) \right\|_{P,2} \leq \left\| \widehat F_{R^*(\pi)|H_1}(\eta;\hat\alpha|H_1) - F_{R^*(\pi)|H_1}(\eta|H_1) \right\|_{P,2}
$$

$$
+ \left\| \widehat F_{R^*(\pi)|H_1}(\eta_\tau;\hat\alpha|H_1) - F_{R^*(\pi)|H_1}(\eta_\tau|H_1) \right\|_{P,2} + \left\| F_{R^*(\pi)|H_1}(\eta|H_1) - F_{R^*(\pi)|H_1}(\eta_\tau|H_1) \right\|_{P,2}. \tag{57}
$$

Now let's analyze the three terms on the RHS of formula (57).

On the one hand, according to Assumption (A1), for any $\eta$,

$$
\left\| \widehat F_{R^*(\pi)|H_1}(\eta;\hat\alpha|H_1) - F_{R^*(\pi)|H_1}(\eta|H_1) \right\|_{P,2} \leq \left\| \delta\left(\widehat F_{R^*(\pi)|H_1}, F_{R^*(\pi)|H_1}\right) \right\|_{P,2} = o(1). \tag{58}
$$

The first and second terms in (57) can thus be bounded.

On the other hand, by assuming (A2) and utilizing the same proof as in formula (54),

$$
\sup_{\eta\in\mathcal{N}_{\tau_n}(\eta_\tau)} \left\| F_{R^*(\pi)|H_1}(\eta|H_1) - F_{R^*(\pi)|H_1}(\eta_\tau|H_1) \right\|_{P,2} = o(1). \tag{59}
$$

Combining the result of (58), (59) with formula (57), we have

$$
\sup_{\eta\in\mathcal{N}_{\tau_n}(\eta_\tau)} \|b(\eta,\hat\alpha) - b(\eta_\tau,\hat\alpha)\|_{P,2}
$$

$$
\leq \frac{2}{\epsilon} \cdot \sup_{\eta\in\mathcal{N}_{\tau_n}(\eta_\tau)} \left\| \widehat F_{R^*(\pi)|H_1}(\eta;\hat\alpha|H_1) - \widehat F_{R^*(\pi)|H_1}(\eta_\tau;\hat\alpha|H_1) \right\|_{P,2} = \frac{2}{\epsilon} \cdot o(1) = o(1).
$$

The proof of Step 2.1.b is thus complete.

**Step 2.1.c**: Prove that $\sup_{\eta\in\mathcal{N}_{\tau_n}(\eta_\tau)} \|c(\eta,\hat\alpha) - c(\eta_\tau,\hat\alpha)\|_{P,2} = o(1)$.

Similar to Step 2.1.b, it follows from Condition (C3$'$) that

$$
\sup_{\eta\in\mathcal{N}_{\tau_n}(\eta_\tau)} \|c(\eta,\hat\alpha) - c(\eta_\tau,\hat\alpha)\|_{P,2}
$$

$$
= \sup_{\eta\in\mathcal{N}_{\tau_n}(\eta_\tau)} \left\| a_2(\hat\alpha) \cdot \widehat{\mathbb{E}}[(\mathbb{I}\{R_1 + \hat R_2^*(\pi) < \eta\} - \mathbb{I}\{R_1 + \hat R_2^*(\pi) < \eta_\tau\})|H_{2,i}] \right\|_{P,2}
$$

$$
= \sup_{\eta\in\mathcal{N}_{\tau_n}(\eta_\tau)} \left\| a_2(\hat\alpha) \cdot \widehat{\mathbb{E}}\left[\mathbb{I}\{R_1 + \hat R_2^*(\pi) \in [\min(\eta_\tau,\eta), \max(\eta_\tau,\eta)]\}|H_{2,i}\right] \right\|_{P,2} \tag{60}
$$

$$
\leq \|a_2(\hat\alpha)\|_{P,2} \cdot \sup_{\eta\in\mathcal{N}_{\tau_n}(\eta_\tau)} \left\| \widehat F_{R_2^*(\pi)|H_2}(\eta - R_1;\hat\alpha|H_{2,i}) - \widehat F_{R_2^*(\pi)|H_2}(\eta_\tau - R_1;\hat\alpha|H_{2,i}) \right\|_{P,2},
$$

where $\widehat{F}_{R_2^*(\pi)|H_2}(r; \hat{\alpha}|H_{2,i})$ is the estimated cdf of $\widehat{R}_2$ given historical data $H_2$ and policy $\pi$. We know that

$$\|a_2(\hat{\alpha})\|_{P,2} = \left\| \left( \frac{\pi_1(A_{1,i}|H_{1,i})}{\widehat{b}_1(A_{1,i}|H_{1,i})} \right) \left( 1 - \frac{\pi_2(A_{2,i}|H_{2,i})}{\widehat{b}_2(A_{2,i}|H_{2,i})} \right) \right\|_{P,2} \leq \frac{2}{\epsilon^2}. \tag{61}$$

Also,

$$\begin{aligned}
&\left\| \widehat{F}_{R_2^*(\pi)|H_2}(\eta - R_1; \hat{\alpha}|H_{2,i}) - \widehat{F}_{R_2^*(\pi)|H_2}(\eta_\tau - R_1; \hat{\alpha}|H_{2,i}) \right\|_{P,2} \\
&\leq \left\| \widehat{F}_{R_2^*(\pi)|H_2}(\eta - R_1; \hat{\alpha}|H_{2,i}) - F_{R_2^*(\pi)|H_2}(\eta - R_1|H_{2,i}) \right\|_{P,2} \\
&+ \left\| \widehat{F}_{R_2^*(\pi)|H_2}(\eta_\tau - R_1; \hat{\alpha}|H_{2,i}) - F_{R_2^*(\pi)|H_2}(\eta_\tau - R_1|H_{2,i}) \right\|_{P,2} \\
&+ \left\| F_{R_2^*(\pi)|H_2}(\eta - R_1|H_{2,i}) - F_{R_2^*(\pi)|H_2}(\eta_\tau - R_1|H_{2,i}) \right\|_{P,2},
\end{aligned} \tag{62}$$

where according to Assumption (A1), it holds for any $\eta$ that

$$\left\| \widehat{F}_{R_2^*(\pi)|H_2}(\eta - R_1; \hat{\alpha}|H_{2,i}) - F_{R_2^*(\pi)|H_2}(\eta - R_1|H_{2,i}) \right\|_{P,2} \leq \left\| \delta \left( \widehat{F}_{R_2^*(\pi)|H_2}, F_{R_2^*(\pi)|H_2} \right) \right\|_{P,2} = o_p(1), \tag{63}$$

and by using Assumption (A2) and the proof in formula (54),

$$\sup_{\eta \in \mathcal{N}_{\tau_n}(\eta_\tau)} \left\| F_{R_2^*(\pi)|H_2}(\eta - R_1|H_{2,i}) - F_{R_2^*(\pi)|H_2}(\eta_\tau - R_1|H_{2,i}) \right\|_{P,2} = o(1). \tag{64}$$

By summarizing the result of formula (63) and (64), the LHS of formula (62) is thus $o_p(1)$. Therefore, we have

$$\begin{aligned}
&\sup_{\eta \in \mathcal{N}_{\tau_n}(\eta_\tau)} \|c(\eta, \hat{\alpha}) - c(\eta_\tau, \hat{\alpha})\|_{P,2} \\
&\leq \|a_2(\hat{\alpha})\|_{P,2} \cdot \sup_{\eta \in \mathcal{N}_{\tau_n}(\eta_\tau)} \left\| \widehat{F}_{R_2^*(\pi)|H_2}(\eta - R_1; \hat{\alpha}|H_{2,i}) - \widehat{F}_{R_2^*(\pi)|H_2}(\eta_\tau - R_1; \hat{\alpha}|H_{2,i}) \right\|_{P,2} \\
&\leq \frac{2}{\epsilon^2} \cdot o(1) = o(1).
\end{aligned} \tag{65}$$

Combining the results in Step 2.1.a-2.1.c, we have

$$\sup_{\eta \in \mathcal{N}_{\tau_n}(\eta_\tau)} \|\mathbb{G}_n[\psi(W; \eta, \hat{\alpha}) - \psi(W; \eta_\tau, \hat{\alpha})]\| = o_p(1).$$

The claim of step 2.1 thus follows.

**Step 2.2**: Prove that $\|\mathbb{G}_n[\psi(W; \eta_\tau, \hat{\alpha}) - \psi^*(W; \eta_\tau)]\| = o_p(1)$.

Notice that under each Fold $s$, $\hat{\alpha}$ is independent with $W$. By Chebyshev's Inequality, it suffices to show that

$$\mathrm{Var}[\mathbb{G}_n[\psi(W; \eta_\tau, \hat{\alpha}) - \psi^*(W; \eta_\tau)]] = \mathrm{Var}[\psi(W; \eta_\tau, \hat{\alpha}) - \psi^*(W; \eta_\tau)] = o(1). \tag{66}$$

According to our definition in formula (51), it follows from triangular inequality that

$$\begin{aligned}
(\mathrm{Var}[\psi(W; \eta_\tau, \hat{\alpha}) - \psi^*(W; \eta_\tau)])^{1/2} &\leq (\mathrm{Var}[a(\eta_\tau, \hat{\alpha}) - a^*(\eta_\tau)])^{1/2} \\
&+ (\mathrm{Var}[b(\eta_\tau, \hat{\alpha}) - b^*(\eta_\tau)])^{1/2} + (\mathrm{Var}[c(\eta_\tau, \hat{\alpha}) - c^*(\eta_\tau)])^{1/2},
\end{aligned} \tag{67}$$

in order to show the LHS is $o_p(1)$, we only need to prove that the three parts on the RHS are all $o_p(1)$.

$$
\begin{aligned}
(\mathrm{Var}&[a(\eta_\tau, \hat{\alpha}) - a^*(\eta_\tau)])^{1/2} \leq \|a(\eta_\tau, \hat{\alpha}) - a^*(\eta_\tau)\|_{P,2} \\
&= \left\| \left( \frac{\pi_1(A_1|H_1)\pi_2(A_2|H_2)}{\widehat{b}_1(A_1|H_1)\widehat{b}_2(A_2|H_2)} - \frac{\pi_1(A_1|H_1)\pi_2(A_2|H_2)}{b_1(A_1|H_1)b_2(A_2|H_2)} \right) (\mathbb{I}\{R_1 + R_2 < \eta_\tau\} - \tau) \right\|_{P,2} \\
&\leq \frac{1}{\epsilon^4} \left\| \widehat{b}_1(A_1|H_1)\widehat{b}_2(A_2|H_2) - b_1(A_1|H_1)b_2(A_2|H_2) \right\|_{P,2} \cdot \|\mathbb{I}\{R_1 + R_2 < \eta_\tau\} - \tau\|_{P,2} \\
&\leq \frac{2}{\epsilon^4} \left\| \widehat{b}_1(A_1|H_1)\widehat{b}_2(A_2|H_2) - b_1(A_1|H_1)b_2(A_2|H_2) \right\|_{P,2} \\
&\leq \frac{2}{\epsilon^4} \left[ \left\| \widehat{b}_1(A_1|H_1)\left\{\widehat{b}_2(A_2|H_2) - b_2(A_2|H_2)\right\} \right\|_{P,2} + \left\| b_2(A_2|H_2)\left\{\widehat{b}_1(A_1|H_1) - b_1(A_1|H_1)\right\} \right\|_{P,2} \right] \\
&\leq \frac{2}{\epsilon^5} \left[ \left\| \widehat{b}_1(A_1|H_1) - b_1(A_1|H_1) \right\|_{P,2} + \left\| \widehat{b}_2(A_2|H_2) - b_2(A_2|H_2) \right\|_{P,2} \right] = o(1),
\end{aligned}
\tag{68}
$$

where we used Condition (C3′), Assumption (A4), triangle inequality, and last equality holds by Assumption (A1).

Likewise,

$$
\begin{aligned}
(\mathrm{Var}&[b(\eta_\tau, \hat{\alpha}) - b^*(\eta_\tau)])^{1/2} \leq \|b(\eta_\tau, \hat{\alpha}) - b^*(\eta_\tau)\|_{P,2} \\
&= \left\| a_1(\hat{\alpha}) \cdot \widehat{\mathbb{E}}[(\mathbb{I}\{\widehat{R}_1^*(\pi) + \widehat{R}_2^*(\pi) < \eta_\tau\} - \tau)|H_1] - a_1^* \cdot \widehat{\mathbb{E}}[(\mathbb{I}\{R_1^*(\pi) + R_2^*(\pi) < \eta_\tau\} - \tau)|H_1] \right\|_{P,2} \\
&\leq \left\| a_1(\hat{\alpha}) \cdot \left\{ \widehat{\mathbb{E}}[\mathbb{I}\{\widehat{R}_1^*(\pi) + \widehat{R}_2^*(\pi) < \eta_\tau\} - \mathbb{I}\{R_1^*(\pi) + R_2^*(\pi) < \eta_\tau\}|H_1] \right\} \right\|_{P,2} \\
&\quad + \left\| \left\{ a_1(\hat{\alpha}) - a_1^* \right\} \cdot \widehat{\mathbb{E}}[(\mathbb{I}\{R_1^*(\pi) + R_2^*(\pi) < \eta_\tau\} - \tau)|H_1] \right\|_{P,2} \\
&\leq \left\| \frac{\pi_1(A_{1,i}|H_{1,i})}{\widehat{b}_1(A_{1,i}|H_{1,i})} \right\|_{P,2} \cdot \left\| \widehat{\mathbb{E}}[\mathbb{I}\{\widehat{R}_1^*(\pi) + \widehat{R}_2^*(\pi) < \eta_\tau\} - \mathbb{I}\{R_1^*(\pi) + R_2^*(\pi) < \eta_\tau\}|H_1] \right\|_{P,2}^2 \\
&\quad + \left\| \frac{\pi_1(A_{1,i}|H_{1,i})}{\widehat{b}_1(A_{1,i}|H_{1,i})} - \frac{\pi_1(A_{1,i}|H_{1,i})}{b_1(A_{1,i}|H_{1,i})} \right\|_{P,2} \cdot \left\| \widehat{\mathbb{E}}[(\mathbb{I}\{R_1^*(\pi) + R_2^*(\pi) < \eta_\tau\} - \tau)|H_1] \right\|_{P,2} \\
&\leq \frac{2}{\epsilon} \cdot \left\| \widehat{F}_{R^*(\pi)|H_1}(\eta_\tau; \hat{\alpha}|H_1) - F_{R^*(\pi)|H_1}(\eta_\tau|H_1) \right\|_{P,2} + \frac{2}{\epsilon^2} \left\| \widehat{b}_1(A_{1,i}|H_{1,i}) - b_1(A_{1,i}|H_{1,i}) \right\|_{P,2} = o(1).
\end{aligned}
\tag{69}
$$

Under Assumption (A1) and (A4), the last equality in Formula (69) holds because

$$
\left\| \widehat{F}_{R^*(\pi)|H_1}(\eta_\tau; \hat{\alpha}|H_1) - F_{R^*(\pi)|H_1}(\eta_\tau|H_1) \right\|_{P,2} \leq \left\| \delta\left( \widehat{F}_{R^*(\pi)|H_1}, F_{R^*(\pi)|H_1} \right) \right\|_{P,2} = o(1).
$$

Similarly,

$$
\begin{aligned}
(\mathrm{Var}&[c(\eta_\tau, \hat{\alpha}) - c^*(\eta_\tau)])^{1/2} \leq \|c(\eta_\tau, \hat{\alpha}) - c^*(\eta_\tau)\|_{P,2} \\
&= \left\| a_2(\hat{\alpha}) \cdot \widehat{\mathbb{E}}[(\mathbb{I}\{R_1 + \widehat{R}_2^*(\pi) < \eta_\tau\} - \tau)|H_2] - a_2^* \cdot \widehat{\mathbb{E}}[(\mathbb{I}\{R_1 + R_2^*(\pi) < \eta_\tau\} - \tau)|H_2] \right\|_{P,2} \\
&\leq \left\| a_2(\hat{\alpha}) \cdot \left\{ \widehat{\mathbb{E}}[\mathbb{I}\{R_1 + \widehat{R}_2^*(\pi) < \eta_\tau\} - \mathbb{I}\{R_1 + R_2^*(\pi) < \eta_\tau\}|H_2] \right\} \right\|_{P,2} \\
&\quad + \left\| \left\{ a_2(\hat{\alpha}) - a_2^* \right\} \cdot \widehat{\mathbb{E}}[(\mathbb{I}\{R_1 + R_2^*(\pi) < \eta_\tau\} - \tau)|H_2] \right\|_{P,2} \\
&\leq \frac{2}{\epsilon^2} \cdot \left\| \widehat{\mathbb{E}}[\mathbb{I}\{R_1 + \widehat{R}_2^*(\pi) < \eta_\tau\} - \mathbb{I}\{R_1 + R_2^*(\pi) < \eta_\tau\}|H_2] \right\|_{P,2} \\
&\quad + \frac{4}{\epsilon^3} \cdot \left\{ \left\| \widehat{b}_1(A_1|H_1) - b_1(A_1|H_1) \right\|_{P,2} + \left\| \widehat{b}_2(A_2|H_2) - b_2(A_2|H_2) \right\|_{P,2} \right\} \\
&\leq \frac{2}{\epsilon^2} \cdot \left\| \widehat{F}_{R_2^*(\pi)|H_2}(\eta_\tau - R_1; \hat{\alpha}|H_2) - F_{R_2^*(\pi)|H_2}(\eta_\tau - R_1|H_2) \right\|_{P,2} + \frac{4}{\epsilon^3} \cdot o(1),
\end{aligned}
$$

which is thus bounded by

$$\frac{2}{\epsilon^2} \cdot \left\| \delta\left(\widehat{F}_{R_2^*(\pi)|H_2}, F_{R_2^*(\pi)|H_2}\right) \right\|_{P,2} + o(1) = o(1),$$

where we used Assumption (A1), (A4) and Positivity assumption (C3′).

The proof of Step 2.2 is thus complete.

**Step 3**: Prove that $\left\| \widehat{\eta}_\tau^{\mathrm{DR}} - \eta_\tau \right\| = O_p(n^{-1/2})$.

Similar to $\psi(W_i; \eta, \alpha)$, we split the original optimization problem $\Psi(W_i; \eta, \alpha)$ into three parts.

$$\Psi(W_i; \eta, \hat{\alpha}) := A(\eta, \hat{\alpha}) + B(\eta, \hat{\alpha}) + C(\eta, \hat{\alpha}), \tag{70}$$

where

$$
\begin{aligned}
A(\eta, \hat{\alpha}) &= \frac{\pi_1(A_1|H_1)\pi_2(A_2|H_2)}{\widehat{b}_1(A_1|H_1)\widehat{b}_2(A_2|H_2)} \rho_\tau(R_1 + R_2 - \eta), \\
B(\eta, \hat{\alpha}) &= a_1(\hat{\alpha}) \cdot \widehat{\mathbb{E}}[\rho_\tau(\widehat{R}_1 + \widehat{R}_2 - \eta)|X_1, (A_1, A_2) \sim \pi], \\
C(\eta, \hat{\alpha}) &= a_2(\hat{\alpha}) \cdot \widehat{\mathbb{E}}[\rho_\tau(R_1 + \widehat{R}_2 - \eta)|H_2, A_2 \sim \pi],
\end{aligned}
\tag{71}
$$

and $a_1(\hat{\alpha})$, $a_2(\hat{\alpha})$ are defined in formula (52). Specifically, when the nuisance parameters $\hat{\alpha}$ is replaced by the true values, we let $\Psi^*(W_i; \eta) := A^*(\eta) + B^*(\eta) + C^*(\eta)$ in the same manner.

Furthermore, we denote $M_n(\eta, \hat{\alpha}) = \mathbb{E}_n[\Psi(W; \eta, \hat{\alpha})]$ as the empirical average of our optimization function, and $M(\eta, \hat{\alpha}) = \mathbb{E}[\Psi(W; \eta, \hat{\alpha})]$. When the models with nuisance parameters are correctly specified, we define $M_n^*(\eta) = \mathbb{E}_n[\Psi^*(W; \eta)]$ and $M^*(\eta) = \mathbb{E}[\Psi^*(W; \eta)]$.

According to Theorem 3.2.5 of Van Der Vaart et al. (1996), to finish the proof of this step, it suffices to show that for every $n$ and a sufficiently small $\delta$, the following two conditions hold:

1. $\mathbb{E}\left[\sup_{\|\eta - \eta_\tau\| < \delta} |M_n(\eta, \hat{\alpha}) - M^*(\eta) - M_n(\eta_\tau, \hat{\alpha}) + M^*(\eta_\tau)|\right] \leq \frac{\delta}{\sqrt{n}}$;

2. There exists a constant $C_4$, such that $M^*(\eta) - M^*(\eta_\tau) \geq C_4 \cdot \|\eta - \eta_\tau\|^2$.

*Note: This is an extension to the original proof of Theorem 3.2.5. The result of Theorem 3.2.5 does not change after involving nuisance parameter $\hat{\alpha}$ in this case. Moreover, notice that the two conditions listed above hold for any $s \in \{1, \ldots, S\}$, and we omitted the subscript $s$ to avoid redundancy.

**Step 3.1**: Prove $\mathbb{E}\left[\sup_{\|\eta - \eta_\tau\| < \delta} |M_n(\eta, \hat{\alpha}) - M^*(\eta) - M_n(\eta_\tau, \hat{\alpha}) + M^*(\eta_\tau)|\right] \leq \frac{\delta}{\sqrt{n}}$.

Since

$$
\begin{aligned}
&\mathbb{E}\left[\sup_{\|\eta - \eta_\tau\| < \delta} |M_n(\eta, \hat{\alpha}) - M^*(\eta) - M_n(\eta_\tau, \hat{\alpha}) + M^*(\eta_\tau)|\right] \\
&\leq \mathbb{E}\left[\sup_{\|\eta - \eta_\tau\| < \delta} |M_n(\eta, \hat{\alpha}) - M(\eta, \hat{\alpha}) - M_n(\eta_\tau, \hat{\alpha}) + M(\eta_\tau, \hat{\alpha})|\right] \\
&\quad + \sup_{\|\eta - \eta_\tau\| < \delta} |(M(\eta, \hat{\alpha}) - M(\eta_\tau, \hat{\alpha})) - (M^*(\eta) - M^*(\eta_\tau))|,
\end{aligned}
\tag{72}
$$

to finish the proof of this step, it suffices to show that the first term on the RHS of Formula (72) is less than or equal to $\frac{\delta}{\sqrt{n}}$, and the second term is actually $o(n^{-\frac{1}{2}} \|\eta - \eta_\tau\|)$. We will prove them in Step 3.1.a and Step 3.1.b accordingly.

**Step 3.1.a**: Prove that $\mathbb{E}\left[\sup_{\|\eta - \eta_\tau\| < \delta} |M_n(\eta, \hat{\alpha}) - M(\eta, \hat{\alpha}) - M_n(\eta_\tau, \hat{\alpha}) + M(\eta_\tau, \hat{\alpha})|\right] \leq \frac{\delta}{\sqrt{n}}$.

Thus, it is equivalent to proving that

$$\mathbb{E}\left[\sup_\eta |M_n(\eta, \hat{\alpha}) - M(\eta, \hat{\alpha}) - M_n(\eta_\tau, \hat{\alpha}) + M(\eta_\tau, \hat{\alpha})|\right] \leq O(n^{-1/2} \|\widehat{\eta}_\tau^{\mathrm{DR}} - \eta_\tau\|). \tag{73}$$

Let $\mathcal{F}_\delta = \{\Psi(W; \eta, \hat{\alpha}) - \Psi(W; \eta_\tau, \hat{\alpha}) : \|\eta - \eta_\tau\| \leq \delta\}$, and $\mathcal{F}_\eta = \{\Psi(W; \eta, \hat{\alpha}) : \|\eta - \eta_\tau\| \leq \delta\}$.

We claim that for any $\eta_1$ and $\eta_2$ satisfying $\|\eta - \eta_\tau\| \leq \delta$, there exist a square-integrable function $M(W; \hat{\alpha})$, such that

$$|\Psi(W; \eta_1, \hat{\alpha}) - \Psi(W; \eta_2, \hat{\alpha})| \leq M(W; \hat{\alpha}) \|\eta_1 - \eta_2\|. \tag{74}$$

To prove this claim, we split formula (74) into three parts:

$$\begin{aligned}
|\Psi(W; \eta_1, \hat{\alpha}) - \Psi(W; \eta_2, \hat{\alpha})| &\leq |A(W; \eta_1, \hat{\alpha}) - A(W; \eta_2, \hat{\alpha})| \\
&\quad + |B(W; \eta_1, \hat{\alpha}) - B(W; \eta_2, \hat{\alpha})| + |C(W; \eta_1, \hat{\alpha}) - C(W; \eta_2, \hat{\alpha})|.
\end{aligned} \tag{75}$$

Without the loss of generality, we assume that $\eta_1 < \eta_2$. Recall that $R^*(\pi)$ is the conditional cumulative reward given $H_1$. Since

$$(R^*(\pi) - \eta_2)\mathbb{I}\{R^*(\pi) < \eta_2\} - (R^*(\pi) - \eta_1)\mathbb{I}\{R^*(\pi) < \eta_1\} = \begin{cases} \eta_1 - \eta_2, & R^*(\pi) < \eta_1 \\ R^*(\pi) - \eta_2, & \eta_1 < R^*(\pi) < \eta_2 \\ 0, & R^*(\pi) > \eta_2 \end{cases} \tag{76}$$

we have $|(R^*(\pi) - \eta_2)\mathbb{I}\{R^*(\pi) < \eta_2\} - (R^*(\pi) - \eta_1)\mathbb{I}\{R^*(\pi) < \eta_1\}| \leq \|\eta_2 - \eta_1\|$.

Thus,

$$\begin{aligned}
&|A(W; \eta_1, \hat{\alpha}) - A(W; \eta_2, \hat{\alpha})| \\
&= \left| \frac{\pi_1(A_1|H_1)\pi_2(A_2|H_2)}{\widehat{b}_1(A_1|H_1)\widehat{b}_2(A_2|H_2)} (\rho_\tau(R_1 + R_2 - \eta_1) - \rho_\tau(R_1 + R_2 - \eta_2)) \right| \\
&\leq \frac{1}{\epsilon^2} \cdot |\rho_\tau(R^*(\pi) - \eta_1) - \rho_\tau(R^*(\pi) - \eta_2)| \\
&\leq \frac{1}{\epsilon^2} \cdot \left\{ |(R^*(\pi) - \eta_2)\mathbb{I}\{R^*(\pi) < \eta_2\} - (R^*(\pi) - \eta_1)\mathbb{I}\{R^*(\pi) < \eta_1\}| + \tau \|\eta_2 - \eta_1\| \right\} \\
&\leq \frac{\tau + 1}{\epsilon^2} \cdot \|\eta_2 - \eta_1\|.
\end{aligned} \tag{77}$$

Similarly,

$$\begin{aligned}
&|B(W; \eta_1, \hat{\alpha}) - B(W; \eta_2, \hat{\alpha})| \\
&= \left| a_1(\hat{\alpha})\widehat{\mathbb{E}}\left[ \rho_\tau(\widehat{R}_1 + \widehat{R}_2 - \eta_1) - \rho_\tau(\widehat{R}_1 + \widehat{R}_2 - \eta_2)\big|H_1 \right] \right| \\
&\leq \frac{2}{\epsilon} \cdot \widehat{\mathbb{E}}\left[ \left|\rho_\tau(\widehat{R}_1 + \widehat{R}_2 - \eta_1) - \rho_\tau(\widehat{R}_1 + \widehat{R}_2 - \eta_2)\right| \big|H_1 \right] \leq \frac{2(\tau + 1)}{\epsilon} \cdot \|\eta_2 - \eta_1\|.
\end{aligned} \tag{78}$$

and

$$\begin{aligned}
&|C(W; \eta_1, \hat{\alpha}) - C(W; \eta_2, \hat{\alpha})| \\
&= \left| a_2(\hat{\alpha}) \cdot \widehat{\mathbb{E}}\left[ \rho_\tau(R_{1,i} + \widehat{R}_2 - \eta_1) - \rho_\tau(R_1 + \widehat{R}_2 - \eta_2)\big|H_2 \right] \right| \\
&\leq \frac{2}{\epsilon^2} \cdot \widehat{\mathbb{E}}\left[ \left|\rho_\tau(R_1 + \widehat{R}_2 - \eta_1) - \rho_\tau(R_1 + \widehat{R}_2 - \eta_2)\right| \big|H_2 \right] \leq \frac{2(\tau + 1)}{\epsilon^2} \cdot \|\eta_2 - \eta_1\|.
\end{aligned} \tag{79}$$

Combining the result of formula (77), (78) and (79), we can simply set $M(W; \hat{\alpha}) = 5(\tau + 1)/\epsilon^2$, so that the condition in Formula (74) is satisfied. More specifically, $\Psi(W; \eta, \hat{\alpha})$ is Lipschitz continuous of order 1. Therefore, $\mathcal{F}_\delta$ has envelope function $F := \delta \cdot M(W; \hat{\alpha})$. Based on Theorem 2.7.11 and Lemma 2.7.8 of Van Der Vaart et al. (1996), its bracketing number satisfies

$$N_{[\,]}(2\epsilon \|F\|, \mathcal{F}_\delta, \|\cdot\|) \leq N(\epsilon, \{\eta : \|\eta - \eta_\tau\| \leq \delta\}, \|\cdot\|) \leq \frac{C\delta}{\epsilon}. \tag{80}$$

Hence, according to the maximal inequality with bracketing numbers, we have

$$\mathbb{E}\left[ \left\| \sqrt{n}(M_n(\eta, \hat{\alpha}) - M(\eta, \hat{\alpha})) \right\|_{\mathcal{F}_\delta} \right] \preceq \int_0^{\|F\|} \sqrt{\log N_{[\,]}(\epsilon, \mathcal{F}_\delta, \|\cdot\|)} \, d\epsilon \preceq \delta. \tag{81}$$

Plugging in the definition of $\|\cdot\|_{\mathcal{F}_\delta}$, it shows that

$$\mathbb{E}\left[\sup_{\|\eta-\eta_\tau\|<\delta}|M_n(\eta,\hat{\alpha})-M(\eta,\hat{\alpha})-M_n(\eta_\tau,\hat{\alpha})+M(\eta_\tau,\hat{\alpha})|\right] \leq O(n^{-1/2}\|\hat{\eta}_\tau^{\mathrm{DR}}-\eta_\tau\|) \leq \frac{\delta}{\sqrt{n}}.$$

Thus, the claim of Step 3.1.a follows.

**Step 3.1.b:** Prove $\sup_{\|\eta-\eta_\tau\|<\delta}|(M(\eta,\hat{\alpha})-M(\eta_\tau,\hat{\alpha}))-(M^*(\eta)-M^*(\eta_\tau))| = o(n^{-\frac{1}{2}}\|\eta-\eta_\tau\|)$.

By using exactly the same strategy as deriving formula (41) in Lemma 2, we can obtain similar results for $\Psi(W;\eta,\hat{\alpha})$:

$$\begin{aligned}
&M(\eta,\hat{\alpha})-M^*(\eta) = \mathbb{E}[\Psi(W;\eta,\hat{\alpha})]-\mathbb{E}[\Psi^*(W;\eta)]\\
=&\mathbb{E}\left[\left(\frac{\pi_1(A_1|H_1)}{\widehat{b}_1(A_1|H_1)}\right)\left(\frac{\pi_2(A_2|H_2)}{\widehat{b}_2(A_2|H_2)}-\frac{\pi_2(A_2|H_2)}{b_2(A_2|H_2)}\right)\cdot\left(\widehat{\mathbb{E}}[\rho_\tau(R_1+R_2^*(\pi)-\eta)|H_2]-\widehat{\mathbb{E}}[\rho_\tau(R_1+\widehat{R}_2^*(\pi)-\eta)|H_2]\right)\right]\\
+&\mathbb{E}\left[\left(\frac{\pi_1(A_1|H_1)}{\widehat{b}_1(A_1|H_1)}-\frac{\pi_1(A_1|H_1)}{b_1(A_1|H_1)}\right)\cdot\left(\widehat{\mathbb{E}}[\rho_\tau(R_1^*(\pi)+R_2^*(\pi)-\eta)|H_1]-\widehat{\mathbb{E}}[\rho_\tau(\widehat{R}_1^*(\pi)+\widehat{R}_2^*(\pi)-\eta)|H_1]\right)\right].
\end{aligned}\tag{82}$$

Similarly, we can get the expression for $M(\eta_\tau,\hat{\alpha})-M^*(\eta_\tau)$ by substituting $\eta_\tau$ for $\eta$ in Formula (82).

Therefore,

$$\begin{aligned}
&|M(\eta,\hat{\alpha})-M^*(\eta)-(M(\eta_\tau,\hat{\alpha})-M^*(\eta_\tau))|\\
\leq&\left|\mathbb{E}\left[\left(\frac{\pi_1(A_1|H_1)}{\widehat{b}_1(A_1|H_1)}\right)\left(\frac{\pi_2(A_2|H_2)}{\widehat{b}_2(A_2|H_2)}-\frac{\pi_2(A_2|H_2)}{b_2(A_2|H_2)}\right)\cdot\widehat{\mathbb{E}}[\rho_\tau(R_1+R_2^*(\pi)-\eta)\right.\right.\\
&\left.\left.-\rho_\tau(R_1+\widehat{R}_2^*(\pi)-\eta)-\rho_\tau(R_1+R_2^*(\pi)-\eta_\tau)+\rho_\tau(R_1+\widehat{R}_2^*(\pi)-\eta_\tau)|H_2]\right]\right|\\
+&\left|\mathbb{E}\left[\left(\frac{\pi_1(A_1|H_1)}{\widehat{b}_1(A_1|H_1)}-\frac{\pi_1(A_1|H_1)}{b_1(A_1|H_1)}\right)\cdot\widehat{\mathbb{E}}[\rho_\tau(R_1^*(\pi)+R_2^*(\pi)-\eta)\right.\right.\\
&\left.\left.-\rho_\tau(\widehat{R}_1^*(\pi)+\widehat{R}_2^*(\pi)-\eta)-\rho_\tau(R_1^*(\pi)+R_2^*(\pi)-\eta_\tau)+\rho_\tau(\widehat{R}_1^*(\pi)+\widehat{R}_2^*(\pi)-\eta_\tau)|H_1]\right]\right|.
\end{aligned}\tag{83}$$

Let's first focus on bounding the last term in formula (83).

$$\begin{aligned}
&\left|\widehat{\mathbb{E}}\left[\rho_\tau(R_1^*(\pi)+R_2^*(\pi)-\eta)-\rho_\tau(\widehat{R}_1^*(\pi)+\widehat{R}_2^*(\pi)-\eta)\right.\right.\\
&\left.\left.-\rho_\tau(R_1^*(\pi)+R_2^*(\pi)-\eta_\tau)+\rho_\tau(\widehat{R}_1^*(\pi)+\widehat{R}_2^*(\pi)-\eta_\tau)|H_1\right]\right|\\
=&\left|\int_{r\in\mathbb{R}}(\rho_\tau(r-\eta_\tau)-\rho_\tau(r-\eta))\cdot\left(\widehat{f}_{R^*(\pi)|H_1}(r|H_1)-f_{R^*(\pi)|H_1}(r|H_1)\right)dr\right|\\
\leq&\int_{r\in\mathbb{R}}|\rho_\tau(r-\eta_\tau)-\rho_\tau(r-\eta)|\cdot\left|\widehat{f}_{R^*(\pi)|H_1}(r|H_1)-f_{R^*(\pi)|H_1}(r|H_1)\right|dr.
\end{aligned}\tag{84}$$

By mean value theorem, for any fixed $r$, there exists a constant $\eta_r \in (\min\{\eta,\eta_\tau\},\max\{\eta,\eta_\tau\})$, such that

$$\rho_\tau(r-\eta_\tau)-\rho_\tau(r-\eta) = (\tau-\mathbb{I}\{r<\eta_r\})\cdot(\eta-\eta_\tau).\tag{85}$$

Therefore, for any $r$, it holds that

$$|\rho_\tau(r-\eta_\tau)-\rho_\tau(r-\eta)| = |(\tau-\mathbb{I}\{r<\eta_r\})\cdot\|\eta-\eta_\tau\| \leq \|\eta-\eta_\tau\|.\tag{86}$$

Under mild conditions and Assumption (A1),

$$
\begin{aligned}
\Big| \widehat{\mathbb{E}} \Big[ &\rho_\tau(R_1^*(\pi) + R_2^*(\pi) - \eta) - \rho_\tau(\widehat{R}_1^*(\pi) + \widehat{R}_2^*(\pi) - \eta) \\
&- \rho_\tau(R_1^*(\pi) + R_2^*(\pi) - \eta_\tau) + \rho_\tau(\widehat{R}_1^*(\pi) + \widehat{R}_2^*(\pi) - \eta_\tau) \big| H_1 \Big] \Big| \\
&\leq \int_{r \in \mathbb{R}} |\rho_\tau(r - \eta_\tau) - \rho_\tau(r - \eta)| \cdot \Big| \widehat{f}_{R^*(\pi)|H_1}(r|H_1) - f_{R^*(\pi)|H_1}(r|H_1) \Big| \, dr, \\
&\leq \|\eta - \eta_\tau\| \cdot \int_{r \in \mathbb{R}} \Big| \widehat{f}_{R^*(\pi)|H_1}(r|H_1) - f_{R^*(\pi)|H_1}(r|H_1) \Big| \, dr \\
&= \|\eta - \eta_\tau\| \cdot 2\delta \left( \widehat{F}_{R^*(\pi)|H_1} - F_{R^*(\pi)|H_1} \right).
\end{aligned}
\tag{87}
$$

Similarly, we can prove by Assumption (A1) that

$$
\begin{aligned}
\Big| \widehat{\mathbb{E}} [ &\rho_\tau(R_1 + R_2^*(\pi) - \eta) - \rho_\tau(R_1 + \widehat{R}_2^*(\pi) - \eta) - \rho_\tau(R_1 + R_2^*(\pi) - \eta_\tau) + \rho_\tau(R_1 + \widehat{R}_2^*(\pi) - \eta_\tau) \big| H_2 ] \Big| \\
&\leq \int_{r \in \mathbb{R}} |\rho_\tau(R_1 + r - \eta_\tau) - \rho_\tau(R_1 + r - \eta)| \cdot \Big| \widehat{f}_{R_2^*(\pi)|H_2}(r|H_2) - f_{R_2^*(\pi)|H_2}(r|H_2) \Big| \, dr \\
&\leq \|\eta - \eta_\tau\| \cdot \int_{r \in \mathbb{R}} \Big| \widehat{f}_{R_2^*(\pi)|H_2}(r|H_2) - f_{R_2^*(\pi)|H_2}(r|H_2) \Big| \, dr \\
&= \|\eta - \eta_\tau\| \cdot 2\delta \left( \widehat{F}_{R_2^*(\pi)|H_2} - F_{R_2^*(\pi)|H_2} \right).
\end{aligned}
\tag{88}
$$

By summarizing the result of Formula (83), (87) and (88), we have

$$
\begin{aligned}
|M(&\eta, \hat{\alpha}) - M^*(\eta) - (M(\eta_\tau, \hat{\alpha}) - M^*(\eta_\tau))| \\
&\leq \left\| \left( \frac{\pi_1(A_1|H_1)}{\widehat{b}_1(A_1|H_1)} \right) \left( \frac{\pi_2(A_2|H_2)}{\widehat{b}_2(A_2|H_2)} - \frac{\pi_2(A_2|H_2)}{b_2(A_2|H_2)} \right) \right\|_{P,2} \cdot 2 \|\eta - \eta_\tau\| \cdot \left\| \delta \left( \widehat{F}_{R_2^*(\pi)|H_2}, F_{R_2^*(\pi)|H_2} \right) \right\|_{P,2} \\
&\quad + \left\| \frac{\pi_1(A_1|H_1)}{\widehat{b}_1(A_1|H_1)} - \frac{\pi_1(A_1|H_1)}{b_1(A_1|H_1)} \right\|_{P,2} \cdot 2 \|\eta - \eta_\tau\| \cdot \left\| \delta \left( \widehat{F}_{R^*(\pi)|H_1}, F_{R^*(\pi)|H_1} \right) \right\|_{P,2} \\
&\leq \frac{1}{\epsilon^3} \left\| \widehat{b}_2(A_2|H_2) - b_2(A_2|H_2) \right\|_{P,2} \cdot 2 \|\eta - \eta_\tau\| \cdot \left\| \delta \left( \widehat{F}_{R_2^*(\pi)|H_2}, F_{R_2^*(\pi)|H_2} \right) \right\|_{P,2} \\
&\quad + \frac{1}{\epsilon^2} \left\| \widehat{b}_1(A_1|H_1) - b_1(A_1|H_1) \right\|_{P,2} \cdot 2 \|\eta - \eta_\tau\| \cdot \left\| \delta \left( \widehat{F}_{R^*(\pi)|H_1}, F_{R^*(\pi)|H_1} \right) \right\|_{P,2} = o(n^{-\frac{1}{2}} \|\eta - \eta_\tau\|),
\end{aligned}
\tag{89}
$$

where the last inequality holds according to Assumption (A1).

**Step 3.2**: Prove that $M^*(\eta) - M^*(\eta_\tau) \geq C_4 \cdot \|\eta - \eta_\tau\|^2$.

According to Taylor series expansion, we have

$$
M^*(\eta) = M^*(\eta_\tau) + \mathbb{E}[\psi^*(W; \eta_\tau)](\eta - \eta_\tau) + \frac{1}{2} \partial_\eta \mathbb{E}[\psi^*(W; \eta_\tau)](\eta - \eta_\tau)^2 + o(\|\eta - \eta_\tau\|^2),
\tag{90}
$$

in which $\mathbb{E}[\psi^*(W; \eta_\tau)] = 0$, and

$$
\begin{aligned}
\partial_\eta \mathbb{E}[\psi^*(W_i; \eta)] &= \partial_\eta \mathbb{E} \left[ \frac{\pi_1(A_1|H_1)\pi_2(A_2|H_2)}{\widehat{b}_1(A_1|H_1)\widehat{b}_2(A_2|H_2)} (\mathbb{I}\{R_1 + R_2 < \eta\} - \tau) \right] \\
&\quad + \partial_\eta \mathbb{E} \left[ \mathbb{E} \left[ a_1^* | H_1 \right] \cdot \widehat{\mathbb{E}} [(\mathbb{I}\{R_1^*(\pi) + R_2^*(\pi) < \eta\} - \tau) | H_1] \right] \\
&\quad + \partial_\eta \mathbb{E} \left[ \mathbb{E} \left[ a_2^* | H_2 \right] \cdot \widehat{\mathbb{E}} [(\mathbb{I}\{R_1 + R_2^*(\pi) < \eta\} - \tau) | H_2] \right] \\
&= \mathbb{E}_{H_1 \sim \mathbb{G}} \left[ f_{R^*(\pi)|H_1}(\eta|H_1) \right] + 0 + 0 = f_{R^*(\pi)}(\eta) \geq 0
\end{aligned}
\tag{91}
$$

where $f_{R^*(\pi)|H_1}(\eta|H_1)$ is the pdf of random variable $R_1^*(\pi_1) + R_2^*(\pi)$ given baseline information $H_1$, and $f_{R^*(\pi)}(\eta)$ is the marginal pdf of $R_1^*(\pi_1) + R_2^*(\pi)$ given baseline distribution $H_1 \sim \mathbb{G}$.

Therefore, by Assumption (A3),

$$M^*(\eta) - M^*(\eta_\tau) = \frac{1}{2} \cdot f_{R^*(\pi)}(\eta)(\eta - \eta_\tau)^2 + o(\|\eta - \eta_\tau\|^2) \geq C_4 \cdot \|\eta - \eta_\tau\|^2, \tag{92}$$

where $C_4$ can be defined as $C_1/2 - \delta$ with a small enough $\delta$ such that $C_4 > 0$.

Therefore, the proof of Step 3 is complete.

**Step 4**: Prove that $\sup_{\|\eta-\eta_\tau\|\leq\delta} |\mathbb{E}[\psi(W;\eta,\hat{\alpha}) - \psi^*(W;\eta)]| = o(n^{-1/2})$.

According to the derivation we did in formula (41),

$$
\begin{aligned}
&\sup_{\|\eta-\eta_\tau\|\leq\delta} \cdot |\mathbb{E}[\psi(W;\eta,\hat{\alpha}) - \psi^*(W;\eta)]| \\
&= \sup_{\|\eta-\eta_\tau\|\leq\delta} \cdot \left| \mathbb{E}\left[ \left(\frac{\pi_1(A_{1,i}|H_{1,i})}{\widehat{b}_1(A_{1,i}|H_{1,i})}\right) \left(\frac{\pi_2(A_{2,i}|H_{2,i})}{\widehat{b}_2(A_{2,i}|H_{2,i})} - \frac{\pi_2(A_{2,i}|H_{2,i})}{b_2(A_{2,i}|H_{2,i})}\right) \right. \right. \\
&\qquad \left. \cdot \left(\mathbb{E}[\mathbb{I}\{R_{1,i} + R_2^*(\pi) < \eta\}|H_{2,i}] - \widehat{\mathbb{E}}[\mathbb{I}\{R_{1,i} + \widehat{R}_2^*(\pi) < \eta\}|H_{2,i}]\right) \right] \\
&\qquad + \mathbb{E}\left[ \left(\frac{\pi_1(A_{1,i}|H_{1,i})}{\widehat{b}_1(A_{1,i}|H_{1,i})} - \frac{\pi_1(A_{1,i}|H_{1,i})}{b_1(A_{1,i}|H_{1,i})}\right) \right. \\
&\qquad \left. \left. \cdot \left(\widehat{\mathbb{E}}[\mathbb{I}\{R_1^*(\pi) + R_2^*(\pi) < \eta\}|H_{1,i}] - \widehat{\mathbb{E}}[\mathbb{I}\{\widehat{R}_1^*(\pi) + \widehat{R}_2^*(\pi) < \eta\}|H_{1,i}]\right) \right] \right| \\
&\leq \frac{1}{\epsilon^3} \cdot \left\|\widehat{b}_2(A_2|H_2) - b_2(A_2|H_2)\right\|_{P,2} \cdot \left\|\delta\left(\widehat{F}_{R_2^*(\pi)|H_2}, F_{R_2^*(\pi)|H_2}\right)\right\|_{P,2} \\
&\quad + \frac{1}{\epsilon^2} \cdot \left\|\widehat{b}_1(A_1|H_1) - b_1(A_1|H_1)\right\|_{P,2} \cdot \left\|\delta\left(\widehat{F}_{R^*(\pi)|H_1}, F_{R^*(\pi)|H_1}\right)\right\|_{P,2} = o(n^{-1/2}),
\end{aligned}
\tag{93}
$$

where the order in the last equality comes from Assumption (A1).

**Step 5**: Prove that $H_0$ is bounded.

In Formula (91), we obtained that $\partial_\eta \mathbb{E}[\psi^*(W_i;\eta)] = \mathbb{E}_{H_1 \sim \mathbb{G}}\left[f_{R^*(\pi)}(\eta|H_1)\right] = f_{R^*(\pi)}(\eta)$. Therefore,

$$H_0 = \partial_\eta^2 \mathbb{E}[\psi^*(W_i;\eta)]\Big|_{\eta=\eta_\tau} = \partial_\eta f_{R^*(\pi)}(\eta)\Big|_{\eta=\eta_\tau}. \tag{94}$$

This condition holds naturally according to Assumption (A5).

By combining the results in Step 2-5, one can follow the structure in Step 1 and the whole proof of Theorem 1 is thus complete. $\qquad\square$

## C.5 Proof of Semi-Parametric Efficiency for $\eta_\tau$

In this subsection, we proof the semi-parametric efficiency of the DR estimator $\widehat{\eta}_\tau^{\mathrm{DR}}$ proposed in Equation 11. There are several ways of finding the efficient influence function (EIF) (see e.g., Kennedy, 2022; Hines et al., 2022; Levy, 2019). In our proof, we focus on deriving the explicit expression of the Gateaux derivative, and the EIF is thus the unique mean zero function whose inner product with the score function equals the pathwise derivative of $\eta_\tau(\alpha)$.

Recall that $W = \{(X_k, A_k, R_k)\}_{k=1}^K$ is the full data with baseline information $X_1 \sim \mathbb{G}$. For the brevity of notation, we define a function $U(\eta, \alpha)$ as $U(\eta, \alpha) = \mathbb{E}_W[\psi(W, \eta, \alpha)]$, i.e.

$$
\begin{aligned}
U(\eta, \alpha) = F_{R^*(\pi)}(\eta) - \tau = \sum_{\{a_k\}_{k=1}^K} \int_{\{r_k\}, \{h_k\}} \Bigg( \mathbb{I}\Big\{ \sum_{k=1}^K r_k < \eta \Big\} - \tau \Bigg) \cdot \prod_{k=1}^K \Bigg\{ \pi_k(a_k|h_k) \times \\
f_{R_k|\bar{A}_k, H_k}(r_k|\bar{a}_k, h_k; \alpha) \cdot f_{H_k|\bar{A}_{k-1}, H_{k-1}}(h_k|\bar{a}_{k-1}, h_{k-1}; \alpha) \Bigg\} d\{r_k\} d\{h_k\}.
\end{aligned}
\tag{95}
$$

Notice that when $k = 1$, we manually set $\{\bar{A}_{k-1}, H_{k-1}\} = \phi$, yielding $f_{H_1|\bar{A}_0, H_0} = f(X_1) \sim \mathbb{G}$. Therefore, by definition, $U(\eta_\tau, \alpha^*) = \mathbb{E}_W[\psi(W, \eta_\tau, \alpha^*)] = 0$. The efficient influence function (EIF) of $\eta_\tau$ should satisfy

$$
\frac{\partial}{\partial \alpha} \eta_\tau(\alpha) \Big|_{\alpha = \alpha^*} = \mathbb{E}\big[ V(W; \eta_\tau, \alpha^*) \cdot s(W) \big],
\tag{96}
$$

where $s(W)$ is the score function of the entire data sequence $W$, given by

$$
s(W) = \frac{\partial}{\partial \alpha} l(W) = \frac{\partial}{\partial \alpha} \left[ \sum_{k=1}^K \Big\{ \log(f_{R_k|\bar{A}_k, H_k; \alpha}) + \log(b(A_k|H_k; \alpha)) + \log(f_{H_k|\bar{A}_{k-1}, H_{k-1}; \alpha}) \Big\} \right].
$$

Our objective is to derive the explicit expression for $V(W; \eta_\tau, \alpha^*)$, which is the EIF of $\eta_\tau$.

To do so, we start from the LHS of Equation 96. For simplicity, we will omit the subscript $\tau$ in $\eta_\tau$ when necessary, and denote $\eta^*$ as the true quantile value under the given target policy $\pi$. Since $U(\eta^*, \alpha^*) = 0$, we have

$$
\frac{\partial}{\partial \alpha} \eta_\tau(\alpha) \Big|_{\alpha = \alpha^*} = - \left[ \frac{\partial}{\partial \eta} U(\eta, \alpha^*) \Big|_{\eta = \eta^*} \right]^{-1} \left[ \frac{\partial}{\partial \alpha} U(\eta^*, \alpha) \Big|_{\alpha = \alpha^*} \right].
\tag{97}
$$

On the RHS of Equation 97, the first term $\frac{\partial}{\partial \eta} U(\eta, \alpha^*) \Big|_{\eta = \eta^*}$ can be easily simplified as

$$
\frac{\partial}{\partial \eta} U(\eta, \alpha^*) \Big|_{\eta = \eta^*} = \frac{\partial}{\partial \eta} \big\{ F_{R^*(\pi)}(\eta) - \tau \big\} \Big|_{\eta = \eta^*} = f_{R^*(\pi)}(\eta^*),
$$

which is a positive constant for any given quantile $\eta^*$ (Notice that the positiveness is guaranteed by Assumption (A3)). Next, we focus on deriving the second term of Equation 97, i.e. $\frac{\partial}{\partial \alpha} U(\eta^*, \alpha) \Big|_{\alpha = \alpha^*}$. Taking the derivative with respect to $\alpha$, we can decompose $\frac{\partial}{\partial \alpha} U(\eta^*, \alpha) \Big|_{\alpha = \alpha^*}$ into two components:

$$
\frac{\partial}{\partial \alpha} U(\eta^*, \alpha) \Big|_{\alpha = \alpha^*} = \sum_{k=1}^K T_1^{(k)} + \sum_{k=1}^K T_2^{(k)},
\tag{98}
$$

where

$$
\begin{aligned}
T_1^{(k)} = \sum_{\{a_{k'}\}_{k'=1}^K} \int_{\{r_{k'}\}, \{h_{k'}\}} \prod_{k'=1}^K \pi_k(a_{k'}|h_{k'}) \cdot \Big( \mathbb{I}\Big\{ \sum_{k'=1}^K r_{k'} < \eta \Big\} - \tau \Big) \cdot \frac{\partial}{\partial \alpha} f_{R_k|\bar{A}_k, H_k}(r_k|\bar{a}_k, h_k; \alpha) \times \\
\prod_{\substack{k'=1 \\ k' \neq k}}^K \Big\{ f_{R_{k'}|\bar{A}_{k'}, H_{k'}}(r_{k'}|\bar{a}_{k'}, h_{k'}; \alpha) \Big\} \cdot \prod_{k'=1}^K \Big\{ f_{H_{k'}|\bar{A}_{k'-1}, H_{k'-1}}(h_{k'}|\bar{a}_{k'-1}, h_{k'-1}; \alpha) \Big\} d\{r_{k'}\} d\{h_{k'}\};
\end{aligned}
$$

$$
\begin{aligned}
T_2^{(k)} = \sum_{\{a_{k'}\}_{k'=1}^K} \int_{\{r_{k'}\}, \{h_{k'}\}} \prod_{k'=1}^K \pi_k(a_{k'}|h_{k'}) \cdot \Big( \mathbb{I}\Big\{ \sum_{k'=1}^K r_{k'} < \eta \Big\} - \tau \Big) \cdot \frac{\partial}{\partial \alpha} f_{H_k|\bar{A}_{k-1}, H_{k-1}}(h_k|\bar{a}_{k-1}, h_{k-1}; \alpha) \times \\
\prod_{k'=1}^K \Big\{ f_{R_{k'}|\bar{A}_{k'}, H_{k'}}(r_{k'}|\bar{a}_{k'}, h_{k'}; \alpha) \Big\} \cdot \prod_{\substack{k'=1 \\ k' \neq k}}^K \Big\{ f_{H_{k'}|\bar{A}_{k'-1}, H_{k'-1}}(h_{k'}|\bar{a}_{k'-1}, h_{k'-1}; \alpha) \Big\} d\{r_{k'}\} d\{h_{k'}\}.
\end{aligned}
$$

**Step 1:** For $T_1^{(k)}$, we can further derive

$$T_1^{(k)} = \mathbb{E}_{H_k, \bar{A}_k, \bar{R}_{k-1}} \left[ \prod_{k'=1}^{k-1} \frac{\pi_{k'}(A_{k'}|H_{k'})}{b_{k'}(A_{k'}|H_{k'})} \sum_{\{a_{k'}\}_{k'=k}^K} \int_{\{r_{k'}\}, \{h_{k'}\}}^{k' \geq k} \prod_{k'=k}^K \pi_k(a_{k'}|h_{k'}) \cdot \frac{\partial}{\partial \alpha} f_{R_k|\bar{A}_k, H_k} \times \right.$$
$$\left. \left( \mathbb{I}\left\{ \sum_{k'=1}^K r_{k'} < \eta \right\} - \tau \right) \cdot \prod_{k'=k+1}^K \left\{ f_{R_{k'}|\bar{A}_{k'}, H_{k'}} \right\} \cdot \prod_{k'=k}^K \left\{ f_{H_{k'}|\bar{A}_{k'-1}, H_{k'-1}} \right\} d\{r_{k'}\} d\{h_{k'}\} \right].$$

Since

$$\frac{\partial}{\partial \alpha} f_{R_k|\bar{A}_k, H_k} = \frac{\partial}{\partial \alpha} \left( \log f_{R_k|\bar{A}_k, H_k} \right) \cdot f_{R_k|\bar{A}_k, H_k}$$
$$= \left\{ \mathbb{E}[s(W)|H_k, \bar{A}_k, \bar{R}_k] - \mathbb{E}[s(W)|H_k, \bar{A}_k, \bar{R}_{k-1}] \right\} \cdot f_{R_k|\bar{A}_k, H_k},$$

we can plug in the result above to the expression of $T_1^{(k)}$, which yields

$$T_1^{(k)} = \mathbb{E}\left[ \prod_{k'=1}^k \frac{\pi_{k'}(A_{k'}|H_{k'})}{b_{k'}(A_{k'}|H_{k'})} \sum_{\{a_{k'}\}_{k'=k+1}^K} \int_{\{r_{k'}\}, \{h_{k'}\}}^{k' \geq k+1} \prod_{k'=k+1}^K \pi_k(a_{k'}|h_{k'}) \cdot \left( \mathbb{I}\left\{ \sum_{k'=1}^K r_{k'} < \eta \right\} - \tau \right) \cdot \prod_{k'=k+1}^K \left\{ \right.\right.$$
$$\left. f_{R_{k'}|\bar{A}_{k'}, H_{k'}} \cdot f_{H_{k'}|\bar{A}_{k'-1}, H_{k'-1}} \right\} \left\{ \mathbb{E}[s(W)|H_k, \bar{A}_k, \bar{R}_k] - \mathbb{E}[s(W)|H_k, \bar{A}_k, \bar{R}_{k-1}] \right\} d\{r_{k'}\} d\{h_{k'}\} \right]$$
$$= \mathbb{E}\left[ \prod_{k'=1}^k \frac{\pi_{k'}(A_{k'}|H_{k'})}{b_{k'}(A_{k'}|H_{k'})} \mathbb{E}_{A_{k'} \sim \pi, H_{k'}, R_{k'}, k' \geq k+1} \left\{ \left( \mathbb{I}\left\{ \sum_{k'=1}^K R_{k'} < \eta \right\} - \tau \right) \cdot \mathbb{E}[s(W)|H_k, \bar{A}_k, \bar{R}_k] \right\} \right.$$
$$\left. - \prod_{k'=1}^k \frac{\pi_{k'}(A_{k'}|H_{k'})}{b_{k'}(A_{k'}|H_{k'})} \mathbb{E}_{A_{k'} \sim \pi, H_{k'}, R_{k'}, k' \geq k+1} \left\{ \left( \mathbb{I}\left\{ \sum_{k'=1}^K R_{k'} < \eta \right\} - \tau \right) \cdot \mathbb{E}[s(W)|H_k, \bar{A}_k, \bar{R}_{k-1}] \right\} \right]$$
$$= \mathbb{E}\left[ \prod_{k'=1}^k \frac{\pi_{k'}(A_{k'}|H_{k'})}{b_{k'}(A_{k'}|H_{k'})} \mathbb{E}\left\{ \left( \mathbb{I}\left\{ \sum_{k'=1}^K R_{k'} < \eta \right\} - \tau \right) \Big| H_k, \bar{A}_k, \bar{R}_k \right\} s(W) \right. \qquad - \prod_{k'=1}^k \frac{\pi_{k'}(A_{k'}|}{b_{k'}(A_{k'}|}$$
$$= \mathbb{E}\left[ \prod_{k'=1}^k \frac{\pi_{k'}(A_{k'}|H_{k'})}{b_{k'}(A_{k'}|H_{k'})} \cdot \left\{ F_{R^*(\pi)|H_k, \bar{A}_k, \bar{R}_k}(\eta) - F_{R^*(\pi)|H_k, \bar{A}_k, \bar{R}_{k-1}}(\eta) \right\} \cdot s(W) \right],$$

where the first expectation after all the equality signs above is taken over $(H_k, \bar{A}_k, \bar{R}_{k-1})$.

The derivation above works for all $k \in \{1, \ldots, K-1\}$. When $k = K$, as a special case,

$$T_1^{(K)} = \mathbb{E}\left[ \prod_{k'=1}^K \frac{\pi_{k'}(A_{k'}|H_{k'})}{b_{k'}(A_{k'}|H_{k'})} \left( \mathbb{I}\left\{ \sum_{k'=1}^K R_{k'} < \eta \right\} - \tau \right) s(W) \right.$$
$$\left. - \prod_{k'=1}^K \frac{\pi_{k'}(A_{k'}|H_{k'})}{b_{k'}(A_{k'}|H_{k'})} \mathbb{E}\left\{ \left( \mathbb{I}\left\{ \sum_{k'=1}^K R_{k'} < \eta \right\} - \tau \right) \Big| H_k, \bar{A}_k, \bar{R}_{k-1} \right\} s(W) \right] \qquad (99)$$
$$= \mathbb{E}\left[ \prod_{k'=1}^K \frac{\pi_{k'}(A_{k'}|H_{k'})}{b_{k'}(A_{k'}|H_{k'})} \cdot \left\{ \left( \mathbb{I}\left\{ \sum_{k'=1}^K R_{k'} < \eta \right\} - \tau \right) - F_{R^*(\pi)|H_k, \bar{A}_k, \bar{R}_{k-1}}(\eta) \right\} \cdot s(W) \right].$$

Note that in the first line of Equation 99, the expectation outside of the indicator function vanishes because all observational data $(H_k, \bar{A}_k, \bar{R}_k)$ were given when $k = K$.

**Step 2:** Similarly, for $T_2^{(k)}$, we have

$$T_2^{(k)} = \mathbb{E}_{H_{k-1}, \bar{A}_{k-1}, \bar{R}_{k-1}} \left[ \prod_{k'=1}^{k-1} \frac{\pi_{k'}(A_{k'}|H_{k'})}{b_{k'}(A_{k'}|H_{k'})} \sum_{\{a_{k'}\}_{k'=k}^{K}} \int_{\{r_{k'}\}, \{h_{k'}\}}^{k' \geq k} \prod_{k'=k}^{K} \pi_k(a_{k'}|h_{k'}) \cdot \frac{\partial}{\partial \alpha} f_{H_k|\bar{A}_{k-1}, H_{k-1}} \times \right.$$
$$\left. \left( \mathbb{I}\left\{ \sum_{k'=1}^{K} r_{k'} < \eta \right\} - \tau \right) \cdot \prod_{k'=k}^{K} \left\{ f_{R_{k'}|\bar{A}_{k'}, H_{k'}} \right\} \cdot \prod_{k'=k+1}^{K} \left\{ f_{H_{k'}|\bar{A}_{k'-1}, H_{k'-1}} \right\} d\{r_{k'}\} d\{h_{k'}\} \right]$$

Since

$$\frac{\partial}{\partial \alpha} f_{H_k|\bar{A}_{k-1}, H_{k-1}} = \frac{\partial}{\partial \alpha} \left( \log f_{H_k|\bar{A}_{k-1}, H_{k-1}} \right) \cdot f_{H_k|\bar{A}_{k-1}, H_{k-1}}$$
$$= \left\{ \mathbb{E}[s(W)|H_k, \bar{A}_{k-1}, \bar{R}_{k-1}] - \mathbb{E}[s(W)|H_{k-1}, \bar{A}_{k-1}, \bar{R}_{k-1}] \right\} \cdot f_{H_k|\bar{A}_{k-1}, H_{k-1}},$$

we can plug in the result above to the expression of $T_2^{(k)}$, which yields

$$T_2^{(k)} = \mathbb{E} \left[ \prod_{k'=1}^{k-1} \frac{\pi_{k'}(A_{k'}|H_{k'})}{b_{k'}(A_{k'}|H_{k'})} \sum_{\{a_{k'}\}_{k'=k}^{K}} \int_{\{r_{k'}\}, \{h_{k'}\}}^{k' \geq k} \prod_{k'=k}^{K} \pi_k(a_{k'}|h_{k'}) \cdot \left( \mathbb{I}\left\{ \sum_{k'=1}^{K} r_{k'} < \eta \right\} - \tau \right) \cdot \prod_{k'=k}^{K} \left\{ \right. \right.$$
$$\left. \left. f_{R_{k'}|\bar{A}_{k'}, H_{k'}} \cdot f_{H_{k'}|\bar{A}_{k'-1}, H_{k'-1}} \right\} \left\{ \mathbb{E}[s(W)|H_k, \bar{A}_{k-1}, \bar{R}_{k-1}] - \mathbb{E}[s(W)|H_{k-1}, \bar{A}_{k-1}, \bar{R}_{k-1}] \right\} d\{r_{k'}\} d\{h_{k'}\} \right]$$
$$= \mathbb{E} \left[ \prod_{k'=1}^{k-1} \frac{\pi_{k'}(A_{k'}|H_{k'})}{b_{k'}(A_{k'}|H_{k'})} \mathbb{E}_{A_{k'} \sim \pi, H_{k'}, R_{k'}, k' \geq k} \left\{ \left( \mathbb{I}\left\{ \sum_{k'=1}^{K} R_{k'} < \eta \right\} - \tau \right) \cdot \mathbb{E}[s(W)|H_k, \bar{A}_{k-1}, \bar{R}_{k-1}] \right\} \right.$$
$$\left. - \prod_{k'=1}^{k-1} \frac{\pi_{k'}(A_{k'}|H_{k'})}{b_{k'}(A_{k'}|H_{k'})} \mathbb{E}_{A_{k'} \sim \pi, H_{k'}, R_{k'}, k' \geq k} \left\{ \left( \mathbb{I}\left\{ \sum_{k'=1}^{K} R_{k'} < \eta \right\} - \tau \right) \cdot \mathbb{E}[s(W)|H_{k-1}, \bar{A}_{k-1}, \bar{R}_{k-1}] \right\} \right]$$
$$= \mathbb{E} \left[ \prod_{k'=1}^{k-1} \frac{\pi_{k'}(A_{k'}|H_{k'})}{b_{k'}(A_{k'}|H_{k'})} \mathbb{E} \left\{ \left( \mathbb{I}\left\{ \sum_{k'=1}^{K} R_{k'} < \eta \right\} - \tau \right) \Big| H_k, \bar{A}_{k-1}, \bar{R}_{k-1} \right\} s(W) \right.$$
$$\left. - \prod_{k'=1}^{k-1} \frac{\pi_{k'}(A_{k'}|H_{k'})}{b_{k'}(A_{k'}|H_{k'})} \mathbb{E} \left\{ \left( \mathbb{I}\left\{ \sum_{k'=1}^{K} R_{k'} < \eta \right\} - \tau \right) \Big| H_{k-1}, \bar{A}_{k-1}, \bar{R}_{k-1} \right\} s(W) \right]$$
$$= \mathbb{E} \left[ \prod_{k'=1}^{k-1} \frac{\pi_{k'}(A_{k'}|H_{k'})}{b_{k'}(A_{k'}|H_{k'})} \cdot \left\{ F_{R^*(\pi)|H_k, \bar{A}_{k-1}, \bar{R}_{k-1}}(\eta) - F_{R^*(\pi)|H_{k-1}, \bar{A}_{k-1}, \bar{R}_{k-1}}(\eta) \right\} \cdot s(W) \right],$$

where the first expectation after all the equality signs above is taken over $(H_{k-1}, \bar{A}_{k-1}, \bar{R}_{k-1})$.

Similar to $T_1^{(k)}$, there is a slight exception when $k = 1$. This happens when the target policy is inconsistent with the behavior policy from the initial stage, which corresponds to the DM estimator component in the final expression of the EIF:

$$T_2^{(1)} = \mathbb{E} \left[ \mathbb{E} \left\{ \left( \mathbb{I}\left\{ \sum_{k'=1}^{K} R_{k'} < \eta \right\} - \tau \right) \Big| H_1 \right\} s(W) \right] = \mathbb{E} \left[ \left\{ F_{R^*(\pi)|H_1}(\eta) - \tau \right\} \cdot s(W) \right].$$

**Step 3:** By combining the expression for $T_1^{(k)}$ and $T_2^{(k)}$ derived in the previous two steps with Equation 98, we have

$$\frac{\partial}{\partial \alpha} U(\eta^*, \alpha)\Big|_{\alpha=\alpha^*} = \sum_{k=1}^{K} T_1^{(k)} + \sum_{k=1}^{K} T_2^{(k)} = T_1^{(K)} + \sum_{k=1}^{K-1} \left\{ T_1^{(k)} + T_2^{(k+2)} \right\} + T_2^{(1)}$$

$$= \mathbb{E}\left[ s(W) \cdot \prod_{k'=1}^{K} \frac{\pi_{k'}(A_{k'}|H_{k'})}{b_{k'}(A_{k'}|H_{k'})} \cdot \left\{ \left( \mathbb{I}\left\{ \sum_{k'=1}^{K} R_{k'} < \eta \right\} - \tau \right) - F_{R^*(\pi)|H_k, \bar{A}_k, \bar{R}_{k-1}}(\eta) \right\} \right.$$

$$+ \sum_{k=1}^{K-1} \prod_{k'=1}^{k} \frac{\pi_{k'}(A_{k'}|H_{k'})}{b_{k'}(A_{k'}|H_{k'})} \cdot \left\{ F_{R^*(\pi)|H_k, \bar{A}_k, \bar{R}_k}(\eta) - F_{R^*(\pi)|H_k, \bar{A}_k, \bar{R}_{k-1}} + \right.$$

$$\left. F_{R^*(\pi)|H_k, \bar{A}_{k-1}, \bar{R}_{k-1}}(\eta) - F_{R^*(\pi)|H_{k-1}, \bar{A}_{k-1}, \bar{R}_{k-1}}(\eta) \right\} + \left\{ F_{R^*(\pi)|H_1}(\eta) - \tau \right\} \right]$$

$$= \mathbb{E}\left[ s(W) \cdot \prod_{k'=1}^{K} \frac{\pi_{k'}(A_{k'}|H_{k'})}{b_{k'}(A_{k'}|H_{k'})} \cdot \left\{ \left( \mathbb{I}\left\{ \sum_{k'=1}^{K} R_{k'} < \eta \right\} - \tau \right) - F_{R^*(\pi)|H_k, \bar{A}_k, \bar{R}_{k-1}}(\eta) \right\} \right.$$

$$+ \sum_{k=1}^{K-1} \prod_{k'=1}^{k} \frac{\pi_{k'}(A_{k'}|H_{k'})}{b_{k'}(A_{k'}|H_{k'})} \cdot \left\{ F_{R^*(\pi)|H_k, \bar{A}_k, \bar{R}_k}(\eta) - F_{R^*(\pi)|H_{k-1}, \bar{A}_{k-1}, \bar{R}_{k-1}}(\eta) \right\}$$

$$\left. + \left\{ F_{R^*(\pi)|H_1}(\eta) - \tau \right\} \right]$$

where the last equation holds since $F_{R^*(\pi)|H_k, \bar{A}_k, \bar{R}_{k-1}} = F_{R^*(\pi)|H_k, \bar{A}_{k-1}, \bar{R}_{k-1}}(\eta)$, given that $\pi_k$ is known. Define $C(\eta^*) = -f_{R^*(\pi)}(\eta^*)$ as a constant independent of $\eta$. According to Equation 97, the EIF of $\eta$ is given by

$$\text{EIF}_\eta = C(\eta^*) \cdot \prod_{k=1}^{K} \frac{\pi_k(A_k|H_k)}{b_k(A_k|H_k)} \cdot \left( \mathbb{I}\left\{ \sum_{k=1}^{K} R_k < \eta \right\} - F_{R^*(\pi)|H_{K-1}, \bar{A}_{K-1}}(\eta) \right)$$

$$+ \sum_{k=2}^{K-1} \prod_{k'=1}^{k} \frac{\pi_{k'}(A_{k'}|H_{k'})}{b_{k'}(A_{k'}|H_{k'})} \cdot \left( F_{R^*(\pi)|H_k, \bar{A}_k}(\eta) - F_{R^*(\pi)|H_{k-1}, \bar{A}_{k-1}}(\eta) \right) \qquad (100)$$

$$+ \frac{\pi_1(A_1|H_1)}{b_1(A_1|H_1)} \cdot \left( F_{R^*(\pi)|H_1, \bar{A}_1}(\eta) - \tau \right).$$

By breaking the bracket and re-arranging terms via propensity scores, this further gives us

$$\text{EIF}_\eta = C(\eta^*) \cdot \mathbb{E}\left[ \frac{\prod_{k'=1}^{K} \pi_{k'}(A_{k'}|H_{k'})}{\prod_{k'=1}^{K} b_{k'}(A_{k'}|H_{k'})} \left( \mathbb{I}\left\{ \sum_{k=1}^{K} R_k < \eta \right\} - \tau \right) \right.$$

$$\left. + \sum_{k=1}^{K} \left( \prod_{k'=1}^{k-1} \frac{\pi_{k'}(A_{k'}|H_{k'})}{b_{k'}(A_{k'}|H_{k'})} \right) \left( 1 - \frac{\pi_k(A_k|H_k)}{b_k(A_k|H_k)} \right) F_{R^*(\pi)|H_k, \bar{A}_k}(\eta) \right]. \qquad (101)$$

The expression above aligns perfectly with the quantile DR estimator we proposed in Equation 11 after taking the derivative with a constant-wise difference. As the optimization result remains the same after multiplying a constant, the EIF we derived above demonstrates the semi-parametric efficiency of our DR estimator.

