# OpenReview forum: "Doubly Robust Uncertainty Quantification for Quantile Treatment Effects in Sequential Decision Making"
_TMLR — Accepted by TMLR_

### Review · Reviewer_ULxt · 2025-05-20

**Summary Of Contributions:**

The authors present the development of a doubly robust estimator for quantile treatment effect, for the case of a single intervention, and for the multi-stage sequential setting. They provide strong theoretical proofs, showing the robustness properties, the convergence, and the semi-parametric efficiency of the proposed estimators. Further, they evaluate the proposed method on simulated and real data.

**Audience:**

Yes

**Broader Impact Concerns:**

I don't have concerns about the broader impact.

**Claims And Evidence:**

Yes

**Requested Changes:**

Regarding the conduct of the experiments
- Regarding the comparison of the Rquantile and Rmean, maybe an additional baseline to consider would be Rmean, but with simple transformation of the outcome, like log? Also, the reported metric is the MSE, is it the MSE of the outcome, or the MSE of the estimated effect? In those tables 1 and 2, the standard deviation could also be reported, in addition to the mean, to give some insight about the significance of the difference between the two methods.
- For the results presented in Figure 3, it would be interesting to represent the same result for other levels of heavy-tailed, only the worst case of the Rmean approach is represented, which does not fully account for all the results in tables 1 and 2. Additionnally, the results are presented only for N=2500, and the number of sample for Figure 3 is not specified.
- The histogram of figure 4 is very badly presented, why this range of the x-axis, while it does not allow to visualize well the data distribution? there is no axes labels on this plot, and the legend is incomplete
- figure 7, the x-axis label is R1, while the text says it represents R*.
- it would be very interesting to provide more simulations with cases where the DM approach works and not the IPW
- the DM name is surprising, does it correspond to a form of g-computation or plug-in g-formula?
- the plots in Figure 1 don't have axes labels.
- regarding the GDBT approach for estimation, what is the used loss? the authors mention that it should be proper, is it?

Overall, to reach a good level the authors should provide more experiments, more baselines, a better figure presentation, and caption so that they are easy to understand, and clarify the state of the art and previous methods. It is also very important for the impact and potential to reuse the method that the code for the estimators is shared, and the code to reproduce all the experiments.

**Strengths And Weaknesses:**

The article is theoretically sound, and clearly written providing an interesting new method for practical applications.

However, the experiments are very insufficiently developed and presented. Although the multi-stage seems to be an entirely new contribution, other methods have been proposed for the single stage quantile estimation, and the authors should review prior works on the topic, and clarify exactly their contribution compared to this prior work. In particular, the work of Firpo, Sergio. "Efficient semiparametric estimation of quantile treatment effects." Econometrica 75.1 (2007): 259-276, seems very relevant, and to have similarities in the proposed solution and its properties. The presentation of the results on simulated and real data should include this baseline, as well as other baselines (some results are shown in supplementary methods, but not all). There are also a number of presentation issues (see the requested changes section).

---

> ### Author Response · Authors · 2025-06-29
> **Response to Reviewer ULxt (Part 1)**
>
> We thank the reviewer for all of your thoughtful comments. Below, we provide our point-by-point responses:
>
> **Strengths And Weaknesses - Comparison against Firpo (2007)**
>
> Thanks for your suggestion of adding the baseline method by Firpo (2007). As their method is only applicable in the single-stage setting, we have conducted comparisons using our single-stage experiments and report the corresponding results at
> [this anonymous GitHub repository](https://anonymous.4open.science/r/QOPE_Rebuttal_TMLR-6378/Comparison_Firpo2007.md).
>
> The results show that our proposed estimator achieves substantially smaller MSE and improved coverage. This improvement is largely due to our use of modern machine learning tools for estimating nuisance functions. In contrast, Firpo (2007) rely on a fixed set of polynomial basis functions. It can be seen that their proposed confidence intervals fails to cover the ground truth in many cases.
>
> We also would also like to emphasize that, in contrast to Firpo (2007), our work considers the more general and challenging multi-stage setting, which is the main focus of our paper.
>
> **Request to Change 1**
>
> **First**, following your suggestion, we have updated our simulation studies to compare *Rquantile*, *Rmean*, and *Rmean\_log* -- a variant of *Rmean* that applies a logarithmic transformation to the reward, computes the average, and then exponentiates the resulting estimator. Since the outcome follows a $t$-distribution, which can take both positive and negative values, we add a sufficiently large constant $C$ to all observed outcomes to ensure positivity before applying the logarithmic transformation, and subtract this constant after exponentiation.
>
> The updated figure is available in [this anonymous GitHub link](https://anonymous.4open.science/r/QOPE_Rebuttal_TMLR-6378/Comparison_Rmean_log.md). It can be seen that *Rmean\_log* achieves a much larger MSE than both *Rmean* and our proposed *Rquantile*. This is expected, since *Rmean\_log* is inherently biased: we have $\exp\\{\mathbb{E}[\log (R+C)]\\} < \mathbb{E}[R]-C$ according to Jensen's inequality. Also, the significantly larger variance of *Rmean_log* arises from the added constant and logarithmic transformation, which smooth out important information and make the model harder to learn.
>
> **Second**, the reported metric in Tables 1 and 2 is the MSE of the estimated quantile of the cumulative reward under the target policy. If we understand the reviewer’s question correctly -- where "outcome" refers to the quantile itself and "treatment effect" refers to the difference in quantiles between treatment and control -- then reported in Tables 1 and 2 are MSEs of the outcome, not the treatment effect.
>
> **Third**, we have included the standard deviation of the MSEs in both Tables 1 and 2 in the updated paper (`marked in blue`). As expected, *Rmean* yields not only higher MSEs over 100 replications but also substantially larger standard deviations. This further supports our claim that *Rmean* is much more unstable compared to *Rquantile* with heavy-tailed outcomes.
>
> **Request to Change 2**
>
> We appreciate the reviewer’s suggestion to include results where the reward distribution is not extremely heavy-tailed, as this provides a more comprehensive comparison between *Rquantile* and *Rmean*. In response, we have added all the plots in [this anonymous GitHub repository](https://anonymous.4open.science/r/QOPE_Rebuttal_TMLR-6378/Rquantile_Rmean_Comparison_All.md). As expected, *Rquantile* becomes less advantageous compared to *Rmean* as the level of heavy-tailedness decreases -- or equivalently, as the degrees of freedom in the $t$-distribution increase -- ultimately leading to comparable performance between the two methods.
>
> **Request to Change 3**
>
> Figure 4 in our original submission (now Figure 1 in the revised version, to better highlight our motivation in the introduction) consists of three plots, with the leftmost one being the histogram you referred to. The $x$-axis range in that plot was automatically determined based on the support of the outcome variable. Due to the heavy-tailed distribution, the extreme values are present but are hard to see in the histogram because of their low density. To improve clarity, we have removed the histogram. For the remaining two plots, we have added a legend and updated the caption to enhance clarity (see Page 2, Figure 1 in the revised paper). We hope these changes address your concerns.
>
> **Request to Change 4**
>
> Thanks for pointing it out. It has been corrected in the updated version.

---

> ### Author Response · Authors · 2025-06-29
> **Response to Reviewer ULxt (Part 2)**
>
> **Request to Change 5**
>
> First, we remark that in general, IPW tends to outperform DM when the behavior policy is close to the target policy. In our original simulation setting, the behavior policy was designed to be similar to the target policy, which naturally favored the performance of IPW.
>
> Second, to address your comment, we have added an additional set of experiments where the behavior and target policies differ largely. The updated results are available at the following [anonymous link](https://anonymous.4open.science/r/QOPE_Rebuttal_TMLR-6378/Comparison_DM_IPW_DR.md). These results show that both IPW and DM have their own strengths. For example, IPW performs better when the target quantile lies in the mid-range, while DM outperforms IPW for higher quantiles. Nonetheless, DR, which combines both approaches, consistently achieves the best overall performance.
>
>
> **Request to Change 6**
>
> The direct method (DM) is indeed conceptually equivalent to the g-computation formula in causal inference. We use the term "direct method" instead of "g-computation" due to its wide use in the machine learning community, particularly in the context of off-policy evaluation in bandits and reinforcement learning (see e.g., Dudík et al., 2014; Farajtabar et al., 2018; Su et al., 2019; Kallus & Uehara,
> 2020). We have added a brief discussion in the revised paper to point out the connection (Page 4, Section 2.2.1, `marked in blue`).
>
> **Request to Change 7**
>
> Thank you for pointing this out. We have added axis labels to Figure 1 (now Figure 2 in the revised main paper).
>
> **Request to Change 8**
>
> We use the standard softmax cross-entropy loss for implementing GBDT classification. By "proper", we mean that the algorithm should perform well to ensure that the resulting estimator converges to the ground truth to satisfy Assumption 1. We have clarified this point in the revised paper (see Section 2.2.2 and Remark 1, both `marked in blue`).
>
> **Lastly**, we have included the source code for our method in the [anonymous GitHub repository](https://anonymous.4open.science/r/QOPE_Rebuttal_TMLR-6378/DR_QuantileOPE.py), under*DR\_QuantileOPE.py*. We truly appreciate the reviewers' comments and are happy to address any further questions.
>
> Sincerely, Authors of paper 4251

---

> ### Author Response · Authors · 2025-06-29
> **References**
>
> 1. Miroslav Dudík, Dumitru Erhan, John Langford, and Lihong Li. Doubly robust policy evaluation and optimization. Statistical Science, 29(4):485–511, 2014.
> 2. Mehrdad Farajtabar, Yinlam Chow, and Mohammad Ghavamzadeh. More robust doubly robust off-policy evaluation. In International Conference on Machine Learning, pp. 1447–1456. PMLR, 2018.
> 3. Sergio Firpo. Efficient semiparametric estimation of quantile treatment effects. Econometrica, 75(1):259–276, 2007.
> 4. Nathan Kallus and Masatoshi Uehara. Double reinforcement learning for efficient off-policy evaluation in markov decision processes. J. Mach. Learn. Res., 21:167–1, 2020.
> 5. Yi Su, Lequn Wang, Michele Santacatterina, and Thorsten Joachims. Cab: Continuous adaptive blending for policy evaluation and learning. In International Conference on Machine Learning, pp. 6005–6014. PMLR, 2019.

---

### Review · Reviewer_QX8u · 2025-05-21

**Summary Of Contributions:**

This paper studies quantile treatment effect estimation using doubly robust estimators, potentially in multi-stage settings. It first defines the quantity of interest as the quantile of the potential outcome under a given policy, based on observational data. The paper then proposes an estimation procedure for this quantity and compares inverse probability weighting (IPW) and doubly robust (DR) methods, initially under the assumption of a single-stage setting. It subsequently extends the approach to the multi-stage case and establishes a corresponding estimation framework. Theoretical guarantees are provided by proving both asymptotic convergence to the standard distribution and bounds on sample complexity. Finally, the paper validates the proposed methods through simulation and real-world experiments.

**Audience:**

Yes

**Claims And Evidence:**

Yes

**Requested Changes:**

Paper Organization
- The paper could benefit from adding more motivating examples or scenarios where the quantile treatment effect is of genuine interest.

- The single-stage and multi-stage settings appear to have significant overlap. A more concise and unified presentation of these two parts would improve clarity and readability.

**Strengths And Weaknesses:**

Strength
- The novelty of the paper is clearly highlighted.

- The paper is comprehensive—covering a clear problem formulation, a well-defined solution framework, theoretical guarantees, and both synthetic and real-world experiments.

- The claims are well supported by a range of empirical and theoretical evidence.


Weakness
- The paper could benefit from including more motivating examples or practical scenarios where the quantile treatment effect is of genuine interest.

- In the real-world experiments, the paper defines the quantity of interest as the mean of the 0.999 quantile. A natural question arises: why use this definition instead of directly estimating quantiles?

- The extension from one stage to multiple stages does not appear to introduce significant technical challenges—or perhaps this aspect is under-explained. Clarifying the difficulty and novelty of this extension would strengthen the contribution.

- Although Assumption 1 is justified in the paper, it seems relatively strong. For instance, if Gradient Boosting Decision Trees (GBDT) are used—as claimed in the experiments—does the assumption still hold? Clarification would be helpful.

- The assumption of i.i.d. triplets in the multi-stage setting also feels strong and may limit practical applicability. For example, when it comes to sequential decision making, there could be correlations between subsequent steps, e.g. Markovian.

- The algorithm follows a plug-in approach, and the core methodological novelty remains somewhat unclear.

---

> ### Author Response · Authors · 2025-06-29
> **Response to Reviewer QX8u (Part 1)**
>
> Thank you for your insightful comments. Below, we provide our point-by-point responses:
>
> **Weakness 1**
>
> Thank you for the suggestion. We have revised the introduction and added several new paragraphs (specifically, the second to fourth paragraphs, `marked in blue`) to strengthen the motivation for studying the quantile treatment effect. In particular:
>
> *  In the second paragraph, we use our real data example from a world-leading short video platform to demonstrate the need for studying quantile treatment effects.
> *  In the third paragraph, we highlight various other scenarios where quantiles are of primary interest, including welfare programs, gender discrimination studies, microbiome research, and e-commerce applications.
> *  In the fourth paragraph, we note that even when the mean reward is the target, heavy-tailed rewards can lead to poor estimators. In contrast, estimating quantiles first and then averaging can yield more accurate mean estimators. This offers an alternative motivation for studying quantiles. We also use our real data example to illustrate the prevalence of heavy-tailed outcomes in practical applications.
>
> **Weakness 2**
>
> *"In the real-world experiments, the paper defines the quantity of interest as the mean of the 0.999 quantile. A natural question arises: why use this definition instead of directly estimating quantiles?"*
>
> We make some clarifications. The goal of our data analysis is two-fold:
> 1. to estimate and conduct statistical inference on the policy effect at any given quantile (as highlighted in the second paragraph of the revised introduction, we are particularly interested in the lower 25th percentile); and
> 2. to estimate the mean outcome.
>
> We acknowledge that the original submission may have given the impression that our main objective was #2. However, we have conducted both types of analyses and have revised the corresponding discussion (see Section 6, paragraph 2-3, `marked in blue`) for clarity.
>
> Additionally, even when focusing on #2, as discussed in the fourth paragraph of the revised introduction, traditional mean-based estimators can be suboptimal in the presence of heavy-tailed outcome distributions. In such cases, estimating quantiles first and then averaging can yield more accurate mean estimators. This is empirically verified in Tables 1, 2, and 4, which further reinforces our motivation for estimating quantiles.

---

> ### Author Response · Authors · 2025-06-29
> **Response to Reviewer QX8u (Part 2)**
>
> **Weakness 3 \& 6**
>
> We would like to clarify both the challenges we addressed and the novelty of our algorithm from three key perspectives:
>
> 1. **Single-stage estimation.** You are right that, once the doubly robust estimating equation is derived, our algorithm adopts a plug-in approach which estimates nuisance functions and substitutes them into the estimating equation to solve for the quantiles. However, the key challenge lies in the plug-in step itself, due to the parameter-dependent nature of the conditional mean function: it depends on the quantile value being estimated. This substiantially complicates the solution of the estimating equation, as it requires to estimate the conditional mean for a range of parameter values, which can be computationally intensive and inefficient (Kallus et al., 2019). A key contribution of our work is to address this challenge using deep conditional generative modeling. By learning the full conditional distribution, our method allows for efficient computation of the conditional mean across all target quantiles via Monte Carlo sampling, making the plug-in procedure more scalable and practical.
> 2. **Doubly robust inference.** Another novelty of our proposal lies in the development of a doubly robust estimator for the variance of the quantile estimator, which is essential for implementing our proposed **doubly robust inference** procedure used for uncertainty quantification. Specifically, our proposed confidence interval for the target quantile is itself doubly robust. While existing literature has primarily focused on **doubly robust estimation** of the quantile value (see e.g., Firpo, 2007), it does not provide similarly robust methods for uncertainty quantification.
> 3. **Multi-stage extension.** To the best of our knowledge, our work is among the first to extend the estimation of quantile treatment effects to the multi-stage setting. Although the estimation procedure may be similar to that of the single-stage setting, the theoretical development is much more complex -- due to challenges such as accounting for the propagation of estimation errors over stages. Moreover, while existing literature has established semiparametric efficiency bounds for quantile estimators in single-stage settings (see, e.g., Firpo, 2007), our work extends these results to the multi-stage setting, as formally presented in Theorem 2.
>
> **Weakness 4**
>
> Thank you for your comment. In response, we first clarify that Assumption 1 is not overly restrictive. We then justify this assumption for estimators constructed similarly to our implementation. Finally, we argue that even if Assumption 1 does not hold exactly, it can be relaxed to much weaker conditions without compromising the main results.
>
> **First**, Assumption 1 requires the product of the convergence rates of the propensity score and the conditional density function estimators to be at least $o(n^{-1/2})$, implying that each one of them must be $o(n^{-1/4})$ if both share the same convergence rate. This is a mild requirement, and similar assumptions are commonly imposed in the literature (see, e.g., Chernozhukov et al., 2018; Farrell et al., 2021; Kallus & Uehara, 2022). For example, when lookup tables are used to estimate the propensity score and conditional density, each estimator typically will converge at a rate of $O(n^{-1/2})$, resulting in a product rate of $O(n^{-1})$, which clearly satisfies the $o(n^{-1/2})$ condition.
>
> **Second**, while convergence guarantees for general gradient boosted decision tree (GBDT) algorithms are not well established in the literature, we note that certain variants do satisfy the required rate. In particular, Schuler et al. (2022) showed that a "lassoed" GBDT algorithm with early stopping can attain a convergence rate of approximately $o(n^{-1/3})$, which is sufficient for satisfying the $o(n^{-1/4})$ requirement.
>
> **Finally**, even in cases where the product rate of $o(n^{-1/2})$ is not attainable, Assumption 1 can be further relaxed by requiring that the propensity score and conditional density estimators converge at a rate of $O(n^{-\ell})$ for any $\ell > 0$. This can be achieved by leveraging high-order influence functions to construct the OPE estimator (see, e.g., Liu et al., 2017; Shi et al., 2021). Notice that the relaxed assumption is substantially weaker, since the exponent $\ell$ can be chosen arbitrarily small.
>
> We have added the related discussions in the revised paper (Page 12, Remark 1, `marked in blue`).

---

> ### Author Response · Authors · 2025-06-29
> **Response to Reviewer QX8u (Part 3)**
>
> **Weakness 5**
>
> We clarify that the i.i.d. assumption is imposed on the entire **data trajectories rather than on individual triplets**. Specifically, within a single trajectory $(X_1, A_1, R_1, X_2, A_2, R_2, \dots, X_K, A_K, R_K)$, we do not assume that the triplets $(X_t, A_t, R_t)$ are i.i.d. over time $t$ -- as you noted, this would be unrealistic. Instead, we consider settings with multiple trajectories $(X_{1,i}, A_{1,i}, R_{1,i}, X_{2,i}, A_{2,i}, R_{2,i}, \dots, X_{K,i}, A_{K,i}, R_{K,i})$, where $i$ indexes the $i$-th trajectory, and these trajectories are assumed to be independent and follow the same distribution function. This form of i.i.d. assumption is standard in the literature (see, e.g., Sutton et al., 1998; Kallus & Uehara, 2020; Shi et al., 2022). We have clarified this point in the revised manuscript (see the beginning of Page 9, `marked in blue`).
>
> **Requested Changes - Paper Organization:**
>
> We begin with the single-stage setting to ease readers into the more complex multi-stage notation. However, we acknowledge that some content in the multi-stage section, particularly the estimation details in Section 3.3, overlapped with earlier parts. We have revised this section to condense and remove redundant content for greater conciseness. Thank you for your suggestion, and we are happy to make further adjustments if there are still sections that feel overly lengthy.
>
>
> Again, thank you for your time and effort in reviewing our paper. We would be glad to address any further questions or suggestions you may have.
>
> Sincerely, Authors of paper 4251

---

> ### Author Response · Authors · 2025-06-29
> **References**
>
> 1. Victor Chernozhukov, Denis Chetverikov, Mert Demirer, Esther Duflo, Christian Hansen, Whitney Newey, and James Robins. Double/debiased machine learning for treatment and structural parameters, 2018.
> 2. Max H Farrell, Tengyuan Liang, and Sanjog Misra. Deep neural networks for estimation and inference. Econometrica, 89(1):181–213, 2021.
> 3. Sergio Firpo. Efficient semiparametric estimation of quantile treatment effects. Econometrica, 75(1):259–276, 2007.
> 4. Nathan Kallus and Masatoshi Uehara. Double reinforcement learning for efficient off-policy evaluation in markov decision processes. J. Mach. Learn. Res., 21:167–1, 2020.
> 5. Nathan Kallus and Masatoshi Uehara. Efficiently breaking the curse of horizon in off-policy evaluation with double reinforcement learning. Operations Research, 2022.
> 6. Nathan Kallus, Xiaojie Mao, and Masatoshi Uehara. Localized debiased machine learning: Efficient inference on quantile treatment effects and beyond. arXiv preprint arXiv:1912.12945, 2019.
> 7. Lin Liu, Rajarshi Mukherjee, Whitney K Newey, and James M Robins. Semiparametric efficient empirical higher order influence function estimators. arXiv preprint arXiv:1705.07577, 2017.
> 8. Alejandro Schuler, Yi Li, and Mark van der Laan. Lassoed tree boosting. arXiv preprint arXiv:2205.10697, 2022.
> 9. Chengchun Shi, Runzhe Wan, Victor Chernozhukov, and Rui Song. Deeply-debiased off-policy interval estimation. In International Conference on Machine Learning, pp. 9580–9591. PMLR, 2021.
> 10. Chengchun Shi, Sheng Zhang, Wenbin Lu, and Rui Song. Statistical inference of the value function for reinforcement learning in infinite-horizon settings. Journal of the Royal Statistical Society Series B: Statistical Methodology, 84(3):765–793, 2022.
> 11. Richard S Sutton, Andrew G Barto, et al. Reinforcement learning: An introduction, volume 1. MIT press Cambridge, 1998

---

### Review · Reviewer_3nDJ · 2025-06-15

**Summary Of Contributions:**

This paper proposes a double robust estimator for the quantile loss function, which they apply to get the quantiles of the reward of a policy in decision-making settings. They extend this to the sequential decision making setting, where the goal is to get the quantiles of the reward of a policy for multiple rounds of decisions simultaneously. Here again they propose a doubly robust estimator of the quantile loss function, which now depends on the estimation of the outcome and propensity for each round. They provide extensive theoretical analysis with normal doubly robust style results and experimentally demonstrate improvement over baselines.

**Audience:**

Yes

**Claims And Evidence:**

Yes

**Requested Changes:**

I request the authors include a more detailed related work section, focused around causal estimation on distributional qualities. As an example of some starting points please consider the following three papers and references therein:

Inference on Counterfactual Distributions. Victor Chernozhukov, Ivan Fernandez-Val, Blaise Melly

Counterfactual Mean Embeddings. Krikamol Muandet, Motonobu Kanagawa, Sorawit Saengkyongam, Sanparith Marukatat

Quantifying Aleatoric Uncertainty of the Treatment Effect: A Novel Orthogonal Learner. Valentyn Melnychuk, Stefan Feuerriegel, Mihaela van der Schaar.

**Strengths And Weaknesses:**

Strengths:
- I think the paper is very well motivated and the extension to sequential decision making problems is a good one. It is easy to see such a method could be usefully applied.
- The theoretical and experimental results are extensive.
- The paper is in general clear, easy to follow, and well written.

Weaknesses:
 -  Currently the related work section is very small (only a paragraph) which makes it very hard to situate the work within the broader literature.

---

> ### Author Response · Authors · 2025-06-29
> **Response to Reviewer 3nDJ**
>
> Many thanks for your thoughtful comments and for taking the time to review our paper. In response to your suggestion, we have added more references -- including those you recommended and several others relevant to our study. These related works are organized into three paragraphs:
>
> (i) policy evaluation in single-stage decision-making;
>
> (ii) policy learning in single-stage decision-making; and
>
> (iii) policy learning and evaluation in RL or MDPs.
>
> Please refer to our newly added related work section (Section 1.1 of the introduction), where the changes are `marked in blue`.
>
> Thank you again for your time and effort in reviewing our paper. We would be happy to address any further questions or suggestions you may have.
>
> Sincerely,
> Authors of Paper 4251

---

### Author Response · Authors · 2025-07-10

Dear Reviewers,

Thank you for taking the time to read our paper and for providing valuable feedback.

As the author-reviewer discussion period approaches this weekend (July 13), we just wanted to send a friendly reminder in case you have any additional questions or points you’d like to discuss. We would be more than happy to engage in further conversation to clarify or elaborate on any aspect of our submission.

Thank you again, and we look forward to your response.

Sincerely,
The Authors of Submission 4251

---

### Decision · Action_Editor_FmqX · 2025-08-13

**Recommendation:** Accept as is

**Audience:**

Yes

**Audience Explanation:**

The authors address a challenging but commonly occurring task of sequential decision-making and within this scope propose an estimator that goes beyond the previously available methods, providing robust and reliable estimates for the quantiles of the outcome distribution in addition to just the mean. That is, they provide a fully developed solution for a task for which no previous methods exist, while providing also a full theoretical validation for it. This is definitely interesting for everyone working on such problems and the proposed method is also likely to be useful in practice, and the extremely detailed proof of the robustness of the estimator may be found useful for people working on the theory of causal inference more broadly. The main outcome, the estimator itself, is likely to be useful also for industry practitioners.

**Claims And Evidence:**

Yes

**Claims Explanation:**

The authors consider the task of estimating quantiles of the outcome distribution in a multi-stage sequential decision-making setup. The authors observe that the distributions in typical setups have heavy tails that need to be accounted for and claim that previous methods focus primarily on estimating mean effects, ignoring uncertainty altogether. They explain why a doubly robust estimator is needed and provide such an estimator with a theoretically sound basis.

The work includes substantial theoretical and empirical evidence supporting the specific claims made in the introduction of the work. In addition to the main result of establishing the robustness of the estimator, the authors e.g. the characterize asymptotic properties of the estimator, demonstrate empirically that typical data sets include heavy tails, and evaluate the estimator on artificial data with known ground truths as well as on a real sequential decision-making problems. Full proofs for all theoretical claims are provided in a detailed appendix. The empirical experimentation was considered somewhat narrow, but sufficient for validating the claims and during the revision the authors improved the transparency of the experimentation and e.g. provided a code for re-producing the experiments.